# The 3d *A*-model and generalised symmetries, part I: Bosonic Chern-Simons theories

Cyril Closset[⋆], Elias Furrer[†], Adam Keyes[‡] and Osama Khlaif[°]

School of Mathematics, University of Birmingham,
Watson Building, Edgbaston, Birmingham B15 2TT, United Kingdom

[⋆] c.closset@bham.ac.uk , [†] e.r.furrer@bham.ac.uk ,
[‡] axk1352@student.bham.ac.uk , [°] okk183@student.bham.ac.uk

## Abstract

The 3d *A*-model is a two-dimensional approach to the computation of supersymmetric observables of three-dimensional $\mathcal{N} = 2$ supersymmetric gauge theories. In principle, it allows us to compute half-BPS partition functions on any compact Seifert three-manifold (as well as of expectation values of half-BPS lines thereon), but previous results focussed on the case where the gauge group $\widetilde{G}$ is a product of simply-connected and/or unitary gauge groups. We are interested in the more general case of a compact gauge group $G = \widetilde{G}/\Gamma$, which is obtained from the $\widetilde{G}$ theory by gauging a discrete one-form symmetry. In this paper, we discuss in detail the case of pure $\mathcal{N} = 2$ Chern–Simons theories (without matter) for simple groups $G$. When $G = \widetilde{G}$ is simply-connected, we demonstrate the exact matching between the supersymmetric approach in terms of Seifert fibering operators and the 3d TQFT approach based on topological surgery in the infrared Chern–Simons theory $\widetilde{G}_k$, including through the identification of subtle counterterms that relate the two approaches. We then extend this discussion to the case where the Chern–Simons theory $G_k$ can be obtained from $\widetilde{G}_k$ by the condensation of abelian anyons which are bosonic. Along the way, we revisit the 3d *A*-model formalism by emphasising its 2d TQFT underpinning.

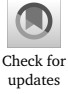

# 1 Introduction

To the theoretical high-energy physicist in 2025, the main claim to fame of supersymmetry may be its uncanny ability to deliver exact results in what are otherwise strongly-interacting quantum field theories (QFTs) [1–4]. In recent years, independently of supersymmetry, a very large body of work explored higher-form symmetries and other generalised symmetries [5] – see *e.g.* [6–58] for a very partial list of references and [59–64] for some lectures and reviews; additional recent works include [65–89]. Our ever-evolving and ever more capacious notion of global symmetry has led to many new constraints on the dynamics of quantum systems. In this context, it is natural to ask whether supersymmetric methods can shed new light on the study of generalised symmetries, and vice versa.

Three-dimensional $\mathcal{N} = 2$ supersymmetric QFT provides an ideal laboratory to explore this question. Our knowledge of supersymmetric observables in such theories is very well developed, and those 3d theories can also admit an intricate set of generalised symmetries; see *e.g.* [9, 19, 44, 90, 91]. In this work, building on previous insights [92–95], we hope to initiate a systematic study of generalised symmetries in 3d $\mathcal{N} = 2$ gauge theories through the computation of their half-BPS observables. We will here focus on theories with one-form symmetries. The general context of our endeavour is the so-called 3d *A*-model [96], which allows us to compute half-BPS observables on Euclidean space-times that take the shape of oriented Seifert three-manifolds $\mathcal{M}$. These Seifert manifolds are circle fibrations over a two-dimensional orbifold $\Sigma_{g,\mathbb{N}}$,

$$S^1_A \longrightarrow \mathcal{M} \longrightarrow \Sigma_{g,\mathbb{N}}. \tag{1}$$

They form the most general set of half-BPS geometries on which to define 3d $\mathcal{N} = 2$ field theories [97].[1] One can then compute the supersymmetric partition functions $Z_{\mathcal{M}}$ and, more generally, correlation functions of arbitrary half-BPS lines wrapping the $S^1_A$ fibre direction:

$$\langle \mathscr{L}_\mu \mathscr{L}_\nu \cdots \rangle_{\mathcal{M}}, \qquad Z_{\mathcal{M}} \equiv \langle \mathbf{1} \rangle_{\mathcal{M}}. \tag{2}$$

In principle, one can compute the supersymmetric path integrals for the observables (2) using supersymmetric localisation [103–106], yet this is a practical course of action only for $\mathcal{M}$ a Seifert manifold of relatively simple topology, such as for the 'topologically twisted index' on $\mathcal{M} = \Sigma_g \times S^1$ [107–110] or for $\mathcal{M} = L(p,q)_b$ a 'squashed' lens space [111–115], including the squashed three-sphere $S^3_b = L(1,1)_b$ [116–119] which plays a central role in understanding 3d RG flows [120–123]. A distinct and more powerful approach involves viewing the 3d $\mathcal{N} = 2$ theory $\mathcal{T}$ on $\mathbb{R}^2 \times S^1_A$ as a 2d $\mathcal{N} = (2,2)$ theory $D_{S^1_A}\mathcal{T}$ of Kaluza-Klein (KK) type [96, 107, 124], using the fact that the supersymmetric background on the fibration (1) is literally a pull-back of the topological *A*-twist [125] on the base $\Sigma_{g,\mathbb{N}}$ [97, 126]. This is the framework of the *3d A-model*, which is really a 2d topological quantum field theory (TQFT) approach – the 3d theory is not fully topological on $\mathcal{M}$, but it is topological along the base $\Sigma_{g,\mathbb{N}}$. The *A*-model formalism allows us to define geometry-changing operators which correspond to changing the details of the Seifert fibration. Indeed, the partition function on $\mathcal{M}$ can always be computed by inserting a *Seifert-fibering operator* $\mathcal{G}_{\mathcal{M}}$ along the circle factor of the product manifold $\Sigma_g \times S^1_A$ [126]:

$$Z_{\mathcal{M}} = \langle \mathcal{G}_{\mathcal{M}} \rangle_{\Sigma_g \times S^1}. \tag{3}$$

This approach is most fully realised in the case where $\mathcal{T}$ is a 3d $\mathcal{N} = 2$ supersymmetric gauge theory – that is, a super-Yang–Mills-Chern–Simons-matter theory [127, 128] – for some compact gauge group $G$, which gives us a UV-free description of the physics. Indeed, in this case the *A*-model for $D_{S^1_A}\mathcal{T}$ is most efficiently written down as an effective field theory for the 2d $\mathcal{N} = (2,2)$ abelian vector multiplets along the 2d Coulomb branch. The vacua of this effective field theory in 2d are known as Bethe vacua, which are essentially obtained as critical points $u = \hat{u}$ of the effective twisted superpotential $\mathcal{W}(u)$ of the *A*-model. Here $u$ is a complex scalar valued in the complexified Cartan subalgebra $\mathfrak{h}_{\mathbb{C}} \subset \mathfrak{g}_{\mathbb{C}}$, $\mathfrak{g} = \mathrm{Lie}(G)$. In the simplest cases, to be discussed momentarily, the 2d vacuum equations take the form:

$$\exp\left(2\pi i \frac{\partial \mathcal{W}(\hat{u})}{\partial u}\right) = 1, \tag{4}$$

and they are often closely related to Bethe ansatz equations [124].[2] Equivalently, they correspond to the supersymmetric ground states of the 3d theory quantised on $\mathbb{R} \times T^2$. Then, the

---

[1]Up to one important exception, the so-called superconformal index background $S^2_q \times S^1$ [98,99]; see *e.g.* [100]. And not including 3d orbifolds recently discussed *e.g.* in [101, 102].

[2]This latter fact will be of no interest to us in this work, except for the fact that it explains this by-now standard terminology of 'Bethe vacua' to denote the 2d vacua of the *A*-model on a cylinder.

half-BPS observables (2) can always be computed as traces over Bethe vacua:

$$\langle \mathscr{L} \rangle_{\mathcal{M}} = \sum_{\hat{u} \in \mathcal{S}_{\mathrm{BE}}} \mathcal{H}(\hat{u})^{g-1} \mathcal{G}_{\mathcal{M}}(\hat{u}) \mathscr{L}(\hat{u}), \tag{5}$$

as we will review in some detail. In particular, the on-shell Seifert fibering operator takes the form:

$$\mathcal{G}_{\mathcal{M}}(\hat{u}) = \prod_i^{\mathbb{N}} \mathcal{G}_{q_i,p_i}(\hat{u}), \tag{6}$$

which is a product of Seifert-fibering operators $\mathcal{G}_{q,p}$ localised at the orbifold points of $\Sigma_{g,\mathbb{N}}$, each introducing an exceptional fibre of the Seifert fibration $\mathcal{M} \cong [0; g; (q_i, p_i)]$, where the coprime integers $(q, p)$ are called the Seifert invariants of the exceptional fiber [129].

The 3d $A$-model formalism was fully fleshed out in [126] under an interesting and slightly restrictive assumption: the gauge group $G = \widetilde{G}$ was assumed to be a product of simply-connected and unitary factors. Equivalently, it was assumed that the fundamental group of $\widetilde{G}$ was a freely generated abelian group,

$$\pi_1(\widetilde{G}) \cong \mathbb{Z}^{n_T}. \tag{7}$$

Then (4) gives us the correct vacuum equations. For simplicity of presentation, in the following, we can assume that $\pi_1(\widetilde{G}) = 0$. One of our goals is to relax this assumption on the compact gauge group $G$. Recall that, if $\widetilde{G}$ denotes the unique simply-connected group with Lie algebra $\mathfrak{g}$, all the other possible compact gauge groups take the form:

$$G = \widetilde{G}/\Gamma, \qquad \Gamma \subseteq Z(\widetilde{G}), \tag{8}$$

where $Z(\widetilde{G})$ denotes the centre of $\widetilde{G}$. Going from $\widetilde{G}$ to $G$ corresponds physically to gauging a discrete one-form symmetry $\Gamma_{\mathrm{3d}}^{(1)} \cong \Gamma$. Such a gauging is possible only if the matter fields in chiral multiplets sit in representations of $G$ and if the 't Hooft anomaly of $\Gamma_{\mathrm{3d}}^{(1)}$ is trivial.

In this paper, the first of a series, we explore the gauging of one-form symmetries in $\mathcal{N} = 2$ supersymmetric Chern–Simons (CS) theories $\widetilde{G}_K$, where $K$ is the supersymmetric CS level for a simple simply-connected group $\widetilde{G}$, in the absence of matter fields. These theories are in many ways 'too simple', since they flow to ordinary ($\mathcal{N} = 0$) Chern–Simons theories $\widetilde{G}_k$ in the infrared (with $k = K - h^\vee$, for $h^\vee$ the dual Coxeter number of $\mathfrak{g}$). Yet they already contain all the essential ingredients involved in gauging $\Gamma_{\mathrm{3d}}^{(1)}$; in particular, the 't Hooft anomaly of the one-form symmetry only depends on the CS levels [5]. Thus, studying Chern–Simons theories allows us to focus on the most essential conceptual aspects of the one-form gauging in 3d gauge theories. In particular, our main result will be to derive the explicit formula for the Seifert-fibering operator (6) of the $G_K$ theory in terms of one for $\widetilde{G}_K$ given in [126]. (The addition of matter chiral multiplets in this discussion is straightforward, but will be discussed elsewhere.)

Moreover, since the infrared CS theory is actually a 3d TQFT whose observables can be computed by topological surgery [130], it is very instructive to compare the Seifert-fibering operator formalism to standard 3d TQFT results [131–139].[3] We find that the two approaches exactly agree up to a counterterm. This counterterm is proportional to the central charge $c(\hat{\mathfrak{g}}_k)$ of the 2d WZW[$G_k$] model that arises at the boundary of the CS theory [130, 155], and it relates our supersymmetric scheme, wherein the partition function depends mildly on the Riemannian metric on $\mathcal{M}$ [156], to the topological scheme of Witten wherein the partition function is metric-independent but depends on a choice of framing [130]. For instance, for the so-called

---

[3]See also [140–154] for related work on CS theories on Seifert manifolds.

squashed three-sphere (with squashing parameter $b^2 \in \mathbb{Q}$ in the present context [126]), one finds:

$$Z^{\text{SUSY}}_{S^3_b}[G_K] = \exp\left(-\frac{\pi i\, c(\hat{\mathfrak{g}}_k)}{12}\left(b^2 + b^{-2} - 2\right)\right) Z^{\text{TQFT}}_{S^3}[G_k], \tag{9}$$

where the squashing-dependence of the supersymmetric partition function of the $\mathcal{N} = 2$ CS theory only appears through this prefactor.[4] Our analysis provides a very detailed consistency check of the results of [126], including various subtle phase factors that arise from the quantisation of the gauginos.[5]

The content of this paper can be summarised by the following diagram:

$$
\begin{array}{ccc}
\widetilde{G}_K \text{ on } \Sigma_g \times S^1 & \xrightarrow{\quad[95]\quad} & G_K \text{ on } \Sigma_g \times S^1 \\
{\scriptstyle[126]}\big\downarrow{\scriptstyle\mathcal{G}^{\widetilde{G}}_{q,p}} & & {\scriptstyle 3.2.3}\big\downarrow{\scriptstyle\mathcal{G}^{G}_{q,p}} \\
\widetilde{G}_K \text{ on Seifert } \mathcal{M} & \xrightarrow{\quad 3.2.4\quad} & G_K \text{ on Seifert } \mathcal{M} \\
\\
\widetilde{G}_k \text{ on } \Sigma_g \times S^1 & \xrightarrow[\quad 3.1\quad]{\text{anyon condensation}} & G_k \text{ on } \Sigma_g \times S^1 \\
{\scriptstyle 2.3.3}\big\downarrow{\scriptstyle\text{top. surgery}} & & \big\downarrow{\scriptstyle 3.1.3} \\
\widetilde{G}_k \text{ on Seifert } \mathcal{M} & \xrightarrow{\quad 3.1.3\quad} & G_k \text{ on Seifert } \mathcal{M}\,.
\end{array} \tag{10}
$$

As the upper commuting diagram in (10) implies, the present paper aims to combine the recent analysis of three of the authors [95] about gauging $\Gamma^{(1)}_{3d}$ on $\Sigma_g \times S^1$ (see also [92, 94]) with the introduction of non-trivial Seifert fibering operators as discussed in [126]. The lower commuting diagram in (10) discusses the infrared perspective, where we can use the full power of the 3d TQFT formalism. In particular, the gauging of the one-form symmetry in the 3d TQFT has been long understood [131] – it is often called 'anyon condensation' in the (condensed-matter) literature [19, 157, 158], and it corresponds to extensions by simple currents in the WZW model that lives at the boundary of space-time [159–161]. We shall review in some detail how the gauging procedure in the 3d $A$-model perspective is equivalent to the anyon condensation process; see also [162, 163] for some closely related discussion that significantly influenced our work.

An important limitation of the present work is that it only addresses gauging processes of $\Gamma^{(1)}_{3d}$ that correspond to condensing abelian anyons that are bosonic. This means that all the Bethe vacua of the $G_K$ theory are bosonic, just like the ones of the $\widetilde{G}_K$ theory. Consequently, the 3d TQFT $G_k$ is bosonic. This need not be the case in general, as some of the abelian anyons one can condense can be fermionic, which leads to $G_k$ theories that are fermionic, that is, they are spin-TQFTs [164] and not only bosonic TQFTs.[6] The more general case, where the one-form

---

[4]Incidentally, assuming this result is also valid for $b \in \mathbb{C}$ and using the formula

$$\tau_{rr} = -\frac{2}{\pi^2}\text{Re}\left[\frac{\partial^2}{\partial b^2}\log Z^{\text{SUSY}}_{S^3_b}\Big|_{b=1}\right],$$

from [97] for the two-point function of the energy-momentum tensor, we then find $\tau_{rr} = 0$, in agreement with the fact that the infrared theory is a 3d TQFT.

[5]These consistency checks for $\widetilde{G}_K$ were already known to the authors of [126] but have not been spelled out in the literature until now, despite the promise made in [126]. Better late than never.

[6]The list of all possible CS theories $G_k$, bosonic or fermionic, for $G$ simple, is reviewed in Appendix B.

symmetry gauging may lead to fermionic ground states, will be discussed in future work [165]. Finally, one can also consider non-invertible symmetries and their explicit realisation in the 3d *A*-model; we will study some instances of categorical symmetries in future work as well [166]. This paper is organised as follows. In section 2, we give a detailed account of the 2d TQFT approach to 3d $\mathcal{N} = 2$ theories and we establish the precise relationship between the Seifert fibering operator and the 3d TQFT formalism for the $\widetilde{G}_K$ CS theories. In section 3, we study one-form symmetries and their gauging on Seifert three-manifolds, focussing on the CS theories $G_K = (\widetilde{G}/\Gamma)_K$ and with the assumption that the $\Gamma$ symmetry lines are bosonic. Our group-theory conventions are summarised in appendix A, and appendix B explains the classification of $G_k$ Chern–Simons theories (bosonic or fermionic) for $\mathfrak{g}$ a simple Lie algebra.

## 2 The 3d *A*-model and Chern–Simons theory

Let us begin by reviewing the 3d *A*-model formalism, emphasising those more elementary aspects that will allow us to best explain the intimate relationship between the *A*-model approach to 3d $\mathcal{N} = 2$ gauge theories and the 3d TQFT approach to Chern–Simons theory on Seifert manifolds. While we closely follow the approach of [96, 107, 126], some of the discussion presented here appears to be new.[7] We refer to the review [100] for further details on 3d $\mathcal{N} = 2$ supersymmetry on curved space in this approach.

In the following, $\Sigma$ denotes any two-manifold (or, later, two-dimensional orbifold) on which the *A*-model is defined. The three-manifold is always taken to be the Seifert manifold $\mathcal{M} = \Sigma \times_f S_A^1$, wherein the $S_A^1$ factor may be non-trivially fibred.

### 2.1 Elementary aspects of 2d TQFTs

Consider a 2d TQFT $\mathcal{T}_{2d}$ with a finitely-generated Hilbert space on the circle, $\mathscr{H}_{S^1}$. This theory has a finitely-generated ring $\mathcal{R}$ of topological operators, denoted by $\mathscr{L}$. Let the label $\mu$ index the operators generating $\mathcal{R}$. The ring structure is given by:

$$\mathscr{L}_\mu \mathscr{L}_\nu = \mathcal{N}_{\mu\nu}{}^\lambda \mathscr{L}_\lambda \,, \tag{11}$$

where the sum over repeated indices is understood. These are local operators in 2d, while in the 3d *A*-model, they are twisted chiral operators that arise as half-BPS lines wrapped along $S_A^1$, hence the notation. The 2d TQFT structure is fully determined by the 2-point and 3-point functions of topological operators on the 2-sphere, which are denoted by:

$$\eta_{\mu\nu} = \left\langle \mathscr{L}_\mu \mathscr{L}_\nu \right\rangle_{S^2} \,, \qquad \mathcal{N}_{\mu\nu\rho} = \left\langle \mathscr{L}_\mu \mathscr{L}_\nu \mathscr{L}_\rho \right\rangle_{S^2} \,. \tag{12}$$

From the path integral point of view and using the topological invariance to move operators around $\Sigma = S^2$, it is clear that these quantities are fully symmetric in the indices. In a slightly more formal language, this gives us a Frobenius algebra structure – that is, $\eta$, also called the topological metric, gives us the Frobenius pairing, with $\eta(\mathscr{L}_\mu, \mathscr{L}_\nu) = \eta_{\mu\nu}$, which is assumed to be non-degenerate. We also assume that there exists a unique unit operator $\mathscr{L}_0 \equiv \mathbf{1}$, indexed by $\mu = 0$, and we then have:

$$\mathcal{N}_{\mu\nu 0} = \eta_{\mu\nu} \,, \qquad \mathcal{N}_{\mu\nu}{}^\lambda = \mathcal{N}_{\mu\nu\rho} \eta^{\rho\lambda} \,, \tag{13}$$

where $\eta^{\mu\nu}$ denotes the inverse of the topological metric. In the 3d *A*-model, the fusion coefficients $\mathcal{N}_{\mu\nu}{}^\lambda$ will take value in $\mathbb{K} = \mathbb{Z}(y)$, the field of fractions of flavour parameters (denoted

---

[7]Though most of it is likely known to experts. See also [167] for a related discussion of the modular group action on the states of the 3d $\mathcal{N} = 2$ theory on $T^2$.

by $y$) of the 3d $A$-model. The integrality of these coefficients follows from the fact that the correlators (12) are also 3d $\mathcal{N} = 2$ supersymmetric path integrals on $S^2 \times S^1$, which have an obvious interpretation as 3d topologically twisted indices in the presence of half-BPS lines [108]. In this paper, since we focus on pure Chern–Simons theories, the fusion coefficients will be integers and the chiral ring is then known as the Verlinde algebra.

**Handle-gluing operator in the twisted chiral-operator basis.**    By the operator-state correspondence, there exists a *twisted chiral-operator basis* $\{|\mu\rangle\}$ of the 2d TQFT Hilbert space $\mathcal{H}_{S^1}$, were $\mathscr{L}_\mu$ is inserted at the origin of the disk:

$$\mu \; \bullet \!\!\!\!\! \bigodiv \; = |\mu\rangle = \mathscr{L}_\mu|0\rangle \,. \tag{14}$$

By topological invariance, this is equivalent to inserting $\mathscr{L}_\mu$ at the tip of a long cigar, with the supersymmetric ground state on $S^1$ obtained by evolving the resulting state for a long time [168]:

$$\bigodiv \quad \rightsquigarrow \quad \bigcirc\!\!\!\!\!\!\!\!\!\!\!\!\!\!\!\!\!\!\!\!\!\!\!\! \bigodiv \;. \tag{15}$$

We similarly define the 'out' state $\langle\mu|$ using a cigar going in the opposite direction:

$$\bigodiv \, \bullet \; \mu \; = \langle\mu| \,. \tag{16}$$

In this quantum-mechanical language, the topological metric is the overlap of states, $\eta_{\mu\nu} = \langle\mu|\nu\rangle$. The product $\mathcal{H}_{S^1} \otimes \mathcal{H}_{S^1} \to \mathcal{H}_{S^1}$ is represented by the pair of pants:

$$\quad = \sum_{\mu,\nu,\rho} |\mu\rangle \mathcal{N}^{\mu\nu\rho} \langle\nu|\langle\rho| \,, \tag{17}$$

and the coproduct by the opposite cobordism:

$$\quad = \sum_{\mu,\nu,\rho} |\nu\rangle|\mu\rangle \mathcal{N}^{\mu\nu\rho} \langle\rho| \,. \tag{18}$$

Note also that we have a useful resolution of the identity represented by an empty cylinder:

$$\bigodiv \!\!\!\!\!\!\!\!\!\!\!\!\!\!\! \bigodiv \quad = \sum_{\mu,\nu} |\mu\rangle \eta^{\mu\nu} \langle\nu| \,, \tag{19}$$

where the indices $\mu, \nu, \cdots$ are always raised and lowered with the topological metric. One can build any 2d TQFT observable from these building blocks. In particular, we are interested in the handle-gluing operator, which is simply obtained by gluing two pairs of pants together:

$$\mathcal{H} = \left(\vcenter{\hbox{(diagram)}}\right) = \sum_{\mu, \nu, \rho, \lambda} \mathcal{N}^{\mu \nu \rho} \mathcal{N}_{\mu \nu}{}^{\lambda} |\rho\rangle\langle\lambda| . \tag{20}$$

This allows us to compute observables on a closed genus-$g$ Riemann surface $\Sigma_g$:

$$\left\langle \mathcal{L}_\mu \mathcal{L}_\nu \cdots \right\rangle_{\Sigma_g} = \left\langle \mathcal{H}^{g-1} \mathcal{L}_\mu \mathcal{L}_\nu \cdots \right\rangle_{\Sigma_1} = \mathrm{Tr}_{S^1}\left( \mathcal{H}^{g-1} \mathcal{L}_\mu \mathcal{L}_\nu \cdots \right) , \tag{21}$$

where the torus partition function precisely gives the trace over $\mathcal{H}_{S^1}$. It is important to note that we can think of $\mathcal{H}$ as a local operator on $\Sigma$, which corresponds to deforming a whole handle (including its contribution to the curvature) into a singular point [107].

paragraphBethe vacua basis and diagonalised fusion rules. In the 3d model, as in any so-called semi-simple 2d TQFT (see e.g. [169]), we have a distinguished basis of states, $\{|\alpha\rangle\}$, which diagonalise the fusion rules. Here, in some $A$-model of twisted chiral multiplets, these states are indexed by the so-called Bethe vacua, namely the (gauge-invariant) solutions $u = \hat{u}_\alpha$ to the 2d vacuum equations, also known as the Bethe equations:

$$\Pi(\hat{u}) \equiv \exp\left( 2\pi i \frac{\partial \mathcal{W}(\hat{u}_\alpha)}{\partial u} \right) = 1 , \tag{22}$$

schematically,[8] where $u$ denote the fundamental twisted chiral fields, and $\Pi(u)$ is called the gauge flux operator. The field $u \in \mathfrak{h}_\mathbb{C}$ is built out of the holonomy of the abelianised 3d gauge field along $S_A^1$ together with the 3d real scalar $\sigma$:

$$u = i\beta(\sigma + i a_0), \qquad a_0 = \frac{1}{2\pi}\int_{S_A^1} A, \tag{23}$$

with $\beta$ the radius of $S_A^1$ [100]. Many 3d $A$-model quantities, including the gauge flux operators and the handle-gluing operator, can be written as rational functions of the single-valued variables $x = e^{2\pi i u} \in T \subset \widetilde{G}$ (valued in a maximal torus $T$ of $\widetilde{G}$).[9] We denote the set of Bethe vacua by:

$$\mathcal{S}_{\mathrm{BE}} \equiv \left\{ \hat{u} \in \mathfrak{h}_\mathbb{C}/\Lambda_{\mathrm{mw}}^{\widetilde{G}}, \,\middle|\, \Pi(\hat{u})^{\mathfrak{m}} = 1 , \forall \mathfrak{m} \in \Lambda_{\mathrm{mw}}^{\widetilde{G}}, \quad \text{and } w \cdot \hat{u} \neq \hat{u} , \forall w \in W_{\widetilde{G}} \right\} \Big/ W_{\widetilde{G}} . \tag{24}$$

Here $\Lambda_{\mathrm{mw}}^{\widetilde{G}}$ is the magnetic weight lattice of $\widetilde{G}$. The Bethe vacua of the $\widetilde{G}$ gauge theory correspond to the solutions to (22) that form complete orbits of the Weyl group $W_{\widetilde{G}}$. The *Bethe states* $|\alpha\rangle \equiv |\hat{u}_\alpha\rangle$ can be constructed as the path integral on a cigar with the boundary condition set by the Bethe vacua $\hat{u}_\alpha \in \mathcal{S}_{\mathrm{BE}}$ on the right-hand-side boundary. Then, two things are true:

1. The Bethe states are orthonormal:

$$\langle \alpha | \beta \rangle = \delta_{\alpha\beta} . \tag{25}$$

---

[8]This holds for 3d $\mathcal{N} = 2$ gauge theories with gauge group $\widetilde{G}$ that are 'simply connected' in the sense of (7).

[9]See *e.g.* [170] for a recent discussion of the algebraic properties of the handle-gluing operators and of the Bethe equations themselves.

The states are orthogonal in the topologically twisted 2d theory because the $A$-twisted theory does not admit any solitons that would interpolate between distinct vacua,[10] and we normalise them to be of unit norm.

2. The Bethe states diagonalise all the twisted chiral operators:

$$\mathscr{L}_\mu|\alpha\rangle = \mathscr{L}_\mu(\alpha)|\alpha\rangle\,, \qquad (26)$$

where $\mathscr{L}(\alpha)$ denote the eigenvalues of the operator $\mathscr{L}$. This is simply because, in the $A$-model effective description, $\mathscr{L}$ is built out of the fundamental fields $u$ and clearly $u|\alpha\rangle = \hat{u}_\alpha|\alpha\rangle$ by definition.

Let $S$ denote the matrix that encodes the change of basis between the chiral-ring basis and the Bethe-state basis:[11]

$$|\mu\rangle = \sum_\alpha S_\mu{}^\alpha|\alpha\rangle\,. \qquad (27)$$

Note that there is no particularly distinguished state amongst the Bethe vacua. Instead, the unique vacuum $|0\rangle \equiv |\mu = 0\rangle$, which corresponds to the empty cigar, has a non-trivial decomposition:

$$|0\rangle = \sum_\alpha S_0{}^\alpha|\alpha\rangle\,. \qquad (28)$$

It directly follows that:

$$Z_{S^2} = \langle\mathbf{1}\rangle_{S^2} = \eta_{00} = \sum_\alpha (S_0{}^\alpha)^2\,. \qquad (29)$$

From the above consideration, we can easily show that the fusion rules are diagonalised. First, we note that:

$$S_{\mu\alpha} = \langle\alpha|\mu\rangle = \langle\alpha|\mathscr{L}_\mu|0\rangle = \sum_\beta \langle\alpha|\mathscr{L}_\mu S_0{}^\beta|\beta\rangle = \mathscr{L}_\mu(\alpha)S_{0\alpha}\,, \qquad (30)$$

hence:[12]

$$\mathscr{L}_\mu(\alpha) = \frac{S_{\mu\alpha}}{S_{0\alpha}}\,. \qquad (31)$$

Note that, since the Bethe states are orthonormal, we can raise and lower the $\alpha$ indices at no cost, unlike for the $\mu$ indices (hence $S_\mu{}^\alpha = S_{\mu\alpha}$). Now, given that:

$$\left(\mathscr{L}_\mu\mathscr{L}_\nu - \mathcal{N}_{\mu\nu}{}^\lambda\mathscr{L}_\lambda\right)|\alpha\rangle = 0\,, \qquad (32)$$

we see that (26) together with (31) imply that:

$$\frac{S_{\mu\alpha}S_{\nu\alpha}}{(S_{0\alpha})^2} = \frac{\mathcal{N}_{\mu\nu}{}^\lambda S_{\lambda\alpha}}{S_{0\alpha}}\,, \qquad (33)$$

where no sum over $\alpha$ is implied. This is equivalent to:

$$\mathcal{N}_{\mu\nu}{}^\lambda = \sum_\alpha \frac{S_{\mu\alpha}S_{\nu\alpha}(S^{-1})_\alpha{}^\lambda}{S_{0\alpha}}\,, \qquad (34)$$

---

[10]This is like for a $B$-twisted Landau-Ginzburg model of chiral multiplets [125,171]. Essentially, the vacuum equations only allow for constant values for $u$.

[11]Here we index the two bases by $\mu, \nu, \rho, \cdots$ and $\alpha, \beta, \gamma, \cdots$, respectively.

[12]Here we assume that $S_{0\alpha} \neq 0$. For Chern–Simons theories this will be true, as this is the statement that $d_\alpha = S_{0\alpha}/S_{00} \geq 1$ for the quantum dimensions of the chiral primaries of unitary RCFT (see *e.g.* [172]).

or, more symmetrically:

$$\mathcal{N}_{\mu\nu\rho} = \sum_\alpha \frac{S_{\mu\alpha}S_{\nu\alpha}S_{\rho\alpha}}{S_{0\alpha}}\,. \tag{35}$$

This is the statement that the Bethe vacua diagonalise the fusion rules, since we then have:


$$= \sum_\alpha \frac{1}{S_{0\alpha}}|\alpha\rangle\langle\alpha|\langle\alpha|\,. \tag{36}$$

The handle-gluing operator (17) is then also diagonal,

$$\mathcal{H} = \sum_\alpha S_{0\alpha}^{-2}|\alpha\rangle\langle\alpha|\,, \qquad \text{that is:} \quad \mathcal{H}(\alpha) = S_{0\alpha}^{-2}\,, \tag{37}$$

and the correlators (21) are given by the more familiar Bethe-vacua formula [107, 110]:

$$\left\langle \mathscr{L}_\mu \mathscr{L}_\nu \cdots \right\rangle_{\Sigma_g} = \left\langle \mathcal{H}^{g-1} \mathscr{L}_\mu \mathscr{L}_\nu \cdots \right\rangle_{\Sigma_1} = \sum_\alpha \left( S_{0\alpha}^{2-2g}\frac{S_{\mu\alpha}}{S_{0\alpha}}\frac{S_{\nu\alpha}}{S_{0\alpha}}\cdots \right), \tag{38}$$

which naturally generalises the $g = 0$ three-point functions (35).

## 2.2 Seifert fibering operators

In the 3d $A$-model on $\Sigma \times S^1$, an additional quasi-topological operation is allowed. Let us first consider $\Sigma$ the cylinder, so that the 3d space-time is

$$\mathcal{M} = \mathbb{R} \times T^2\,, \tag{39}$$

where $\mathbb{R}$ is the Euclidean time of the $A$-model. The supersymmetric ground states of the 3d $\mathcal{N} = 2$ theory quantised on $T^2$ are isomorphic to the set of Bethe vacua:

$$\mathcal{H}_{T^2}^{(3d)} \cong \mathcal{H}_{S^1}\,. \tag{40}$$

The insertion of a line $\mathscr{L}_\mu$ along $S_A^1$ at the tip of a long cigar gives us a state as before:


$$\mathscr{L}_\lambda \qquad = |\mathscr{L}_\lambda\rangle\,, \tag{41}$$

where we drew the $T^2$ boundary more explicitly.[13] Now, from the 3d perspective, we should expect some non-trivial action of the modular group $\text{SL}(2,\mathbb{Z})$ on this Hilbert space. Assuming that this modular action is understood, we can tentatively introduce a non-trivial fibration of the $S_A^1$ circle over the $\Sigma$ base through a Dehn surgery – we will discuss this in more detail in subsection 2.3.3 below.

---

[13] We will mostly suppress the extra circle from our notation, however.

**The Seifert fibering operator.** Given some modular action $\widetilde{U} \in \mathrm{SL}(2, \mathbb{Z})$, which we will write as the $2 \times 2$ matrix:

$$\widetilde{U} = \begin{pmatrix} s & -p \\ t & q \end{pmatrix} \in \mathrm{SL}(2, \mathbb{Z}), \tag{42}$$

one could, in principle, work out the modular action on the supersymmetric ground-states, which we denote by $\widetilde{\mathcal{U}}$. Then, in the 2d Hilbert space picture, we can tentatively define the *Seifert fibering operator* as:

$$\mathcal{G}_{q,p} = \quad \raisebox{-1em}{\includegraphics} \quad = \sum_{\mu,\nu,\rho} \langle 0|\widetilde{\mathcal{U}}|\mu\rangle |\nu\rangle \mathcal{N}^{\mu\nu\rho} \langle\rho| . \tag{43}$$

Here we simply glued an empty cap $\langle 0|$ onto a pair of pants with a non-trivial twist by $\widetilde{\mathcal{U}}$. Note that $\widetilde{U}$ is determined by the coprime integers $p$ and $q$, up to an ambiguity $(s, t) \to (s-p, t+q)$ which does not affect the supersymmetric geometry [126]. In the Bethe-vacua basis $\{|\alpha\rangle\}$, the operation (43) takes the simpler form:

$$\mathcal{G}_{q,p} = \sum_{\alpha} \frac{\mathcal{U}_{\alpha 0}}{S_{0\alpha}} |\alpha\rangle\langle\alpha| . \tag{44}$$

Here we defined the matrix element:

$$\mathcal{U}_{\alpha 0} = \langle 0|\widetilde{\mathcal{U}}|\alpha\rangle . \tag{45}$$

In particular, the Seifert fibering operator is also diagonalised by the Bethe vacua, just like the handle-gluing operator it is built from, with eigenvalues:

$$\mathcal{G}_{q,p}(\alpha) = \frac{\mathcal{U}_{\alpha 0}}{S_{0\alpha}} . \tag{46}$$

This description only makes sense if we can consistently perform the surgery in the cohomology of the supercharges $\mathcal{Q}_-$ and $\mathcal{Q}_+$ that survive the $A$-twists. Indeed, one can preserve precisely this half of the supersymmetry on any oriented Seifert-fibred three-manifold [97, 100]. The topological invariance along $\Sigma_g$ allows one to concentrate the non-trivial effects of the Seifert fibration to a finite number of points. In this way, we can obtain non-trivial $(q, p)$ Seifert fibers through the action of some twisted chiral operators $\mathcal{G}_{q,p}$ acting at the base points of the exceptional fibers. The neighbourhood of an exceptional Seifert fibre locally looks like a quotient of $\mathbb{C} \times S_A^1$ by the action:

$$(z , \psi) \sim \left( e^{\frac{2\pi i}{q}} z , \psi + \frac{2\pi p}{q} \right) , \tag{47}$$

for $z$ some local complex coordinate on $\Sigma$. In particular, the Seifert fibering operator $\mathcal{G}_{q,p}$ introduces a $\mathbb{Z}_q$ orbifold point at $z = 0$ on $\Sigma$. In practice, to compute the full modular action $\widetilde{\mathcal{U}}$ on the Bethe vacua remains an open problem in general 3d $\mathcal{N} = 2$ theories. Nonetheless, the Seifert fibering operators $\mathcal{G}_{q,p}(u)$ have been explicitly constructed off-shell for any 3d $\mathcal{N} = 2$ supersymmetric $\widetilde{G}$ gauge theory, using a mixture of supersymmetric localisation techniques and $A$-model arguments [126]. In any 3d $\mathcal{N} = 2$ gauge theory, the off-shell Seifert operator, like the off-shell handle-gluing operator, is a defect line operator wrapping the $S_A^1$ factor, which is realised on the 2d Coulomb branch as a particular holomorphic function $\mathcal{G}_{q,p}(u)$ of the

Coulomb-branch variables $u$; the on-shell operator is simply the value of that function on the Bethe vacua, which are located at particular points $u = \hat{u}_\alpha$ on the Coulomb branch — here, we use the shorthand notation $\mathcal{G}_{q,p}(\alpha) \equiv \mathcal{G}_{q,p}(\hat{u}_\alpha)$ for the off-shell operator in order to match our 2d TQFT notation.

Given the above data, we can write down any 3d $\mathcal{N} = 2$ half-BPS observable as a 3d $A$-model observable [126]. Let us denote by $\mathcal{G}_{\mathcal{M}}$ the Seifert-fibering operator (6) corresponding to all the exceptional Seifert fibers:[14]

$$\mathcal{G}_{\mathcal{M}}(\alpha) = \prod_{i=0}^{N} \mathcal{G}_{q_i,p_i}(\alpha) = \prod_{i=0}^{N} \frac{\mathcal{U}_{\alpha 0}^{(q_i,p_i)}}{S_{0\alpha}} \,. \tag{48}$$

Then, inserting $\mathcal{G}_{\mathcal{M}}$ inside the trace (38), we have:

$$\langle \mathscr{L}_\mu \mathscr{L}_\nu \cdots \rangle_{\mathcal{M}} = \sum_\alpha \left( S_{0\alpha}^{2-2g-N-1} \frac{S_{\mu\alpha}}{S_{0\alpha}} \frac{S_{\nu\alpha}}{S_{0\alpha}} \cdots \times \prod_{i=0}^{N} \mathcal{U}_{\alpha 0}^{(q_i,p_i)} \right) \,. \tag{49}$$

This gives us the correlation function of half-BPS line operators wrapping $S_A^1$ at generic fibers, generalising the $\mathcal{M} = \Sigma_g \times S_A^1$ case discussed above.

## 2.3 Supersymmetric Chern–Simons theory with simply-connected gauge group

Consider a 3d $\mathcal{N} = 2$ supersymmetric Chern–Simons (CS) theory with simply-connected simple gauge group $\widetilde{G}$ at level $K$, where we assume that $K \geq h^\vee$. Here $h^\vee$ is the dual Coxeter number of the Lie algebra $\mathfrak{g} = \mathrm{Lie}(\widetilde{G})$. The Chern–Simons level effectively acts like a mass term for the gaugino,[15] and by integrating them out we obtain a bosonic Chern–Simons theory for $\widetilde{G}$ at level

$$k \equiv K - h^\vee \,, \tag{50}$$

in the infrared. Hence, in this case, the supersymmetric $A$-model formalism should match up against the 3d TQFT formalism for the bosonic $\widetilde{G}_k$ theory [130]. This is what we show explicitly in the rest of this section.

### 2.3.1 Bethe vacua and integrable representations

**States generated by Wilson lines.** An interesting basis of states for the $\widetilde{G}_k$ bosonic CS theory on $T^2$ can be obtained by inserting Wilson lines along the circle at the centre of a solid torus [130, 155] – this prepares a specific state:

$$|W_\lambda\rangle \in \mathcal{H}_{T^2} \,. \tag{51}$$

The Wilson lines $W_\lambda$ are indexed by the highest-weights $\lambda \in \Lambda_{\mathrm{w}}^{\widetilde{G}}$, corresponding to irreducible representations of $\widetilde{G}$. Only the level-$k$ integrable representations are allowed, corresponding to $\lambda$ such that $(\lambda, \theta) \leq k$ for $\theta$ the highest root of $\mathfrak{g}$ (see Appendix A).

Consider putting our 3d $\mathcal{N} = 2$ CS theory on $\Sigma \times S_A^1$. It should be clear from the discussion of subsection 2.1 that the CS states (51) are precisely the states (14) generated by twisted-chiral operators in the 3d $A$-model, with $\mathscr{L}_\lambda = W_\lambda$ being a Wilson line wrapping $S_A^1$, which we denote by:

$$W_\lambda \left( \quad \right) = |W_\lambda(S_A^1)\rangle = W_\lambda |0\rangle \,. \tag{52}$$

---

[14]We will review the relevant Seifert geometry in section 2.3.3 below. Here we defined $(q_0, p_0) = (1, \mathrm{d})$.

[15]To see this, one can consider a super-Yang–Mills term as a UV regulator, in which case the gauginos have a mass proportional to $K g^2$, with $g^2$ the 3d Yang–Mills coupling.

Indeed, the supersymmetric Wilson line,

$$W_\lambda = \text{Tr}_\lambda P \exp\left( i \int_{S_A^1} (A - i\sigma d\psi) \right), \tag{53}$$

is given in the *A*-model by a Laurent polynomial in the variables $x_a = e^{2\pi i u_a}$, the character of the representation $\mathfrak{R}_\lambda$:[16]

$$W_\lambda(u) = \text{ch}_\lambda(e^{-2\pi i u}) = \sum_{\rho \in \mathfrak{R}_\lambda} e^{-2\pi i \rho(u)}. \tag{54}$$

Note that the supersymmetric Wilson line is equivalent to the ordinary Wilson line in the IR, because $\sigma = 0$ on-shell in the $\mathcal{N} = 2$ Chern–Simons theory. The relation between Wilson loops wrapping Seifert fibers and group characters is also well-established in pure CS theory [143].

**Solutions to the Bethe equations.** Let us now check explicitly that the Bethe vacuum equations reproduce the expected results from $\widetilde{G}_k$ Chern–Simons theory [155], assuming that $K \geq h^\vee$, as expected because the Bethe vacua give us the supersymmetric ground states of the $\widetilde{G}_K$ $\mathcal{N} = 2$ theory on a torus. The twisted superpotential of the $\mathcal{N} = 2$ CS theory on a circle reads:

$$\mathcal{W} = \frac{K}{2}(u, u) + \frac{K_g}{24}, \tag{55}$$

where $(u, u)$ denotes the Killing form (see Appendix A for our conventions). We will set the UV gravitational Chern–Simons terms $K_g$ [173] to

$$K_g = \dim(\mathfrak{g}), \tag{56}$$

which ensures that the gravitational CS level in the infrared bosonic CS description vanishes.[17] The gauge flux operator reads:

$$\Pi(u)^{\mathfrak{m}} \equiv \exp\left( 2\pi i \mathfrak{m} \frac{\partial \mathcal{W}}{\partial u} \right), \tag{57}$$

and the Bethe vacua are obtained as solutions $u = \hat{u}$ of the Bethe equations, $\Pi(\hat{u})^{\mathfrak{m}} = 1$, namely:

$$e^{2\pi i K(\mathfrak{m}, \hat{u})} = 1, \qquad \forall \mathfrak{m} \in \Lambda_{\text{mw}}^{\widetilde{G}}, \tag{58}$$

for $\mathfrak{m}$ any GNO-quantised magnetic flux for $\widetilde{G}$, which is equivalent to saying that $K(\cdot, u)$ is a weight. Taking $u = u_a \mathfrak{e}^a$ with $\{\mathfrak{e}^a\}$ our basis for $\Lambda_{\text{mw}}^{\widetilde{G}}$ dual to the fundamental-weight basis $\{\mathfrak{e}_a\}$, we have $(\mathfrak{m}, u) = \kappa^{ab} \mathfrak{m}_a \hat{u}_b$ in terms of the Killing-form matrix $\kappa^{ab}$ for $\mathfrak{g}$, and so:

$$\hat{u}_a = \frac{\kappa_{ab}^{-1}(\rho_W^b + \lambda^b)}{K}, \tag{59}$$

where $\hat{\lambda} \equiv \rho_W + \lambda$ is some weight. Admissible solutions to the Bethe equation need to be acted on freely by the Weyl group $W_{\widetilde{G}}$. The maximal orbits under $W_{\widetilde{G}}$ are in one-to-one correspondence with regular dominant weights, which are dominant weights $\lambda$ such that $\lambda^a \geq 1$, $\forall a$, in the fundamental-weight basis. Hence we can parametrise the Bethe vacua by $\hat{\lambda}$ a regular

---

[16]Here we picked the convention (53) for the Wilson loop in the representation $\lambda$, which would be the Wilson loop in the complex conjugate representation in the conventions of [100], which we otherwise follow. This is so as to match standard conventions for bosonic CS theories in what follows.

[17]Integrating out the gauginos has this effect. See *e.g.* Appendix A.2 of [126] for a more detailed explanation.

dominant weight. We write this down as $\hat{\lambda} = \rho_W + \lambda$ as in (59), with $\lambda$ any dominant weight and $(\rho_W^a) = (1, \cdots, 1)$. The latter happens to be the Weyl vector:

$$\rho_W = \sum_a e_a = \frac{1}{2} \sum_{\alpha \in \Delta^+} \alpha \,. \tag{60}$$

Finally we need to quotient by large-gauge transformations around $S_A^1$, $\hat{u} \sim \hat{u} + \mathfrak{m}$, $\forall \mathfrak{m} \in \Lambda_{\text{mw}}^{\widetilde{G}}$, which act on the weights as:

$$\lambda \sim \lambda + K \alpha^\vee \,, \tag{61}$$

for $\alpha^\vee \in \Lambda_{\text{cr}}$ the coroots. Hence the Bethe vacua take value in:

$$\lambda \in \frac{\Lambda_{\text{w}}}{W_{\widetilde{G}} \ltimes K \Lambda_{\text{cr}}} \,, \tag{62}$$

which are precisely the level-$k$ integrable representations of $\widetilde{G}_k$ (see *e.g.* [172]).

**Bethe states from vortex loops.** In the path integral formalism, the Bethe states $|\hat{u}_\lambda\rangle$ of the $\widetilde{G}$ theory correspond to half-BPS boundary conditions that set the $A$-model field $u$ to (59) on the right-boundary of a cap – see *e.g.* [168, 174]. Interestingly, the Bethe states can be generated by some twisted chiral operators $V_\lambda$:

$$V_\lambda \;\;\bigcirc\!\!\!\!) \;\; = V_\lambda |0\rangle = |\hat{u}_\lambda\rangle \,, \tag{63}$$

which are indexed by integrable representations just like the Wilson loops. In 3d, the $V_\lambda$ lines are half-BPS vortex loops [175, 176] wrapping $S_A^1$, which are disorder operators that impose a singular profile for the abelianised gauge field around the loop:

$$A_a \sim \frac{\hat{\lambda}_a}{K} d\varphi \,, \tag{64}$$

which is precisely the condition imposed by (59). These vortex operators are also known as monodromy defect operators or 't Hooft loop operators – see *e.g.* [131, 143, 177, 178]. Here $\varphi$ is the angular coordinate winding around the loop, and we use the shorthand notation $\hat{\lambda}_a = \kappa_{ab}^{-1} \hat{\lambda}^a$. The specification of such a defect is equivalent to inserting a (non-quantised) magnetic flux $\hat{\lambda}/K$ on the cigar. For any weight $\lambda$, the half-BPS vortex loop takes the explicit form:

$$V_\lambda = \exp\left(-i \int_{S_A^1} \hat{\lambda}(A - i\sigma d\psi)\right) \,, \tag{65}$$

where one conjugates the gauge field $A$ along $S_A^1$ to the Cartan subalgebra. Inserting this operator into the $\mathcal{N} = 2$ supersymmetric CS path integral modifies the equations of motion from $F = \sigma = D = 0$ to:

$$F_a = \frac{\hat{\lambda}_a}{K} \delta_V \,, \qquad D_a = -\frac{\hat{\lambda}_a}{K} \delta_V \,, \tag{66}$$

and $\sigma = 0$, with $\delta_V$ a Dirac $\delta$-function at the location of the vortex line which integrates to $2\pi$ on the cigar. Note that $F_a + D_a = 0$ due to supersymmetry.

### 2.3.2 Bethe states and modular transformations

The Bethe vacua are supersymmetric ground states of the 3d $\mathcal{N} = 2$ theory quantised on $T^2 = S^1 \times S_A^1$, where the first circle is the 'spatial direction' in the $A$-model on $\Sigma$. While the $A$-model supersymmetry treats the two circles differently in general (since the two supercharges that survive on a generic $\Sigma \times S_A^1$ anticommute to translation along $S_A^1$), when $\Sigma$ is a flat cylinder so that the 3-manifold is $\mathbb{R} \times T^2$, all four supercharges are preserved, and thus we expect a well-defined action of the modular group $\mathrm{SL}(2, \mathbb{Z})$ on the Bethe states, as already mentioned in section 2.2.

In the case of the $\mathcal{N} = 2$ Chern–Simons theory $\widetilde{G}_K$ without matter fields, the $\mathrm{SL}(2, \mathbb{Z})$ action is well understood in terms of the infrared 3d TQFT: the $S$ transformation directly maps the Wilson loop states to the Bethe states (also known as Verlinde states):

$$|W_\lambda\rangle = S_\lambda{}^\mu |\hat{u}_\mu\rangle. \tag{67}$$

The change of basis (27) is thus given by the $S$-matrix of the infrared Chern–Simons theory. This implies that, under a large diffeomorphism $S$ of the $T^2$ boundary of the solid torus, the Wilson line $W_\lambda$ along the central longitude $S_A^1$ is mapped to the vortex line $V_\lambda$ wrapping the same circle.

We can now check that the 3d $A$-model formalism reproduces the known results for modular matrices of the Chern–Simons theory. Firstly, we already noted that the insertion of any half-BPS line along $S_A^1$ is diagonalized by the Bethe vacua. Here, inserting $W_\lambda$ at a point on the cap $|\hat{u}_\mu\rangle$, we obtain:

$$V_\mu \, \vcenter{\hbox{$\bigcirc$}} \overset{W_\lambda}{} = \mathrm{ch}_\lambda(e^{-2\pi i \hat{u}_\mu})|\hat{u}_\mu\rangle. \tag{68}$$

Then, we see that:

$$S_{\lambda\mu} = \langle \hat{u}_\mu | W_\lambda \rangle = \langle \hat{u}_\mu | W_\lambda | 0 \rangle = \mathrm{ch}_\lambda(e^{-2\pi i \hat{u}_\mu}) S_{0\mu}. \tag{69}$$

Recall that $S_{0\mu}$ is related to the handle-gluing operator by (37), and that the latter is given by [107]:

$$\mathcal{H}(u) = K^{\mathrm{rank}(\mathfrak{g})} |\kappa| \prod_{\alpha \in \Delta} \left(1 - e^{2\pi i \alpha(u)}\right)^{-1}, \tag{70}$$

for this $\widetilde{G}_K$ $\mathcal{N} = 2$ CS theory, with $|\kappa|$ the determinant of the Killing form. Hence, we find:

$$S_{0\mu} = \mathcal{H}(\hat{u}_\mu)^{-\frac{1}{2}} = \mathrm{A}_S \, e^{-\frac{2\pi i}{K}(\rho_{\mathrm{W}}, \rho_{\mathrm{W}} + \mu)} \prod_{\alpha \in \Delta^+} \left(1 - e^{\frac{2\pi i}{K}(\alpha, \rho_{\mathrm{W}} + \mu)}\right), \tag{71}$$

where the product is over the positive roots, and with the normalisation:

$$\mathrm{A}_S \equiv \frac{i^{|\Delta^+|}}{K^{\frac{\mathrm{rank}(\mathfrak{g})}{2}} |\kappa|^{\frac{1}{2}}} = i^{|\Delta^+|} \left|\frac{\Lambda_{\mathrm{w}}}{K \Lambda_{\mathrm{cr}}}\right|^{-\frac{1}{2}}, \tag{72}$$

determined up to a sign (here we fixed the sign by comparing to the known CS results). We then find:

$$S_{\lambda\mu} = \mathrm{ch}_\lambda(e^{-2\pi i \hat{u}_\mu}) S_{0\mu} = \mathrm{A}_S \sum_{w \in W_{\mathfrak{g}}} \epsilon(w) \, e^{-\frac{2\pi i}{K}(w(\rho_{\mathrm{W}} + \lambda), \rho_{\mathrm{W}} + \mu)}, \tag{73}$$

where we used the Weyl character formula (A.19). Upon writing $K$ as $K = k + h^\vee$, this is the well-known formula for the $S$-matrix of the bosonic CS theory $\widetilde{G}_k$ [132, 179].

The 'ordinary' fibering operator introduces a principal circle fibration over $\Sigma$ [96]. For this $\mathcal{N} = 2$ supersymmetric CS theory, it is given by:

$$\mathcal{F}(u) = e^{2\pi i(\mathcal{W} - u\partial_u \mathcal{W})} = e^{2\pi i\left(-\frac{K}{2}(u,u) + \frac{\dim(\mathfrak{g})}{24}\right)}. \tag{74}$$

Evaluating this on the Bethe vacua and using the identity $(\rho_W, \rho_W) = h^\vee \dim(\mathfrak{g})/12$, one finds:

$$\mathcal{F}(\hat{u}_\mu) = e^{-2\pi i h_\mu + \frac{\pi i}{12} c(\widetilde{G}_k)}, \tag{75}$$

where $h_\mu$ and $c(\widetilde{G}_k)$ are given by:

$$h_\mu \equiv \frac{(\mu, \mu + 2\rho_W)}{2(k + h^\vee)} = \frac{(\mu, \mu + 2\rho_W)}{2K}, \qquad c(\widetilde{G}_k) \equiv \frac{k \dim(\mathfrak{g})}{k + h^\vee} = \left(1 - \frac{h^\vee}{K}\right)\dim(\mathfrak{g}). \tag{76}$$

These are the conformal spin and the central charge of the corresponding WZW$[\widetilde{G}_k]$ model. Recall that $\theta_\lambda \equiv e^{2\pi i h_\lambda}$ is the topological spin of the Wilson line $W_\lambda$. The fibering operator thus acts on the Bethe vacua exactly like the inverse of the modular $T$-matrix of the $\widetilde{G}_k$ CS theory:

$$T_{\lambda\mu} = \delta_{\lambda\mu} e^{2\pi i\left(h_\mu - \frac{c(\widetilde{G}_k)}{24}\right)} = \delta_{\lambda\mu} \mathcal{F}(\hat{u}_\mu)^{-1}. \tag{77}$$

These modular matrices satisfy the SL$(2, \mathbb{Z})$ relations:

$$S^2 = C, \qquad (ST)^2 = C, \qquad C^2 = \mathbf{1}, \tag{78}$$

where the central element $C$ acts as the charge conjugation matrix:

$$C_{\lambda\mu} = \delta_{\lambda\bar{\mu}} = \begin{cases} 1, & \text{if } \mathfrak{R}_\lambda = \overline{\mathfrak{R}}_\mu, \\ 0, & \text{otherwise.} \end{cases} \tag{79}$$

That is, $C_{\lambda\mu} = 1$ if $\lambda$ and $\mu$ are the highest weights of complex conjugate representations.[18] Recall that the usual SL$(2, \mathbb{Z})$ generators read:

$$S = \begin{pmatrix} 0 & -1 \\ 1 & 0 \end{pmatrix}, \qquad T = \begin{pmatrix} 1 & 1 \\ 0 & 1 \end{pmatrix}, \qquad C = \begin{pmatrix} -1 & 0 \\ 0 & -1 \end{pmatrix}. \tag{80}$$

We further note that the relations (37) and (77) between the $A$-model operators and the 3d TQFT modular matrices are well-known 'experimental facts' in the literature, and that they have often been used to identify TQFT phases of 3d supersymmetric theories [180–188]. The above discussion hopefully clarifies this relationship; we will also extend it to general Seifert fibering operators.

### 2.3.3 Seifert geometry and topological surgery

Let us briefly review a few relevant facts about Seifert geometry. (See [126] for a longer review.) From a TQFT perspective, we wish to view the introduction of a non-trivial circle fibration over $\Sigma$ as a sequence of topological surgery operations. The elementary surgery, generally called Dehn twist, is performed at a smooth point on $\Sigma$, where we locally have the trivial fibration $\mathcal{N} = \mathbb{D}^2 \times S_A^1$, where $\mathbb{D}^2$ is a small disk. Then $\mathcal{N}$ is a tubular neighbourhood

---

[18]Recall that the only compact real Lie algebras that admit intrinsically complex representations (as opposed to real or pseudoreal) are $\mathfrak{a}_{N-1} = \mathfrak{su}(N)$ (for $N > 2$) and $\mathfrak{e}_6$. In all the other cases we have $C = \mathbf{1}$.

of a regular fiber. For many purposes, it is useful to view $\mathcal{N}$ as the solid torus, $T(1,0)$, where we insert $\mathscr{L}_\mu$ along the *longitude* (*i.e.* the non-contractible 1-cycle) at the origin of the disk; the contractible 1-cycle (here, the boundary of the cigar) is called the *meridian*. Recall that the meridian $M$ of a solid torus is uniquely defined (up to orientation), while the longitude $L$ is only defined up to a shift:

$$L \to L + tM, \tag{81}$$

for $t \in \mathbb{Z}$. Picking local coordinates $(z, \psi)$ on $\mathbb{D}^2 \times S_A^1$, this ambiguity corresponds to the non-trivial automorphism:

$$(z, \psi) \to (e^{it\psi} z, \psi). \tag{82}$$

This will be important below. Then, introducing a non-trivial fibration at $z = 0$ corresponds to replacing $T(1,0)$ by a non-trivially fibred solid torus.

**Seifert fibre topology.** A Seifert 3-manifold $\mathcal{M}$ is a circle bundle over an orbifold:

$$S^1 \to \mathcal{M} \to \hat{\Sigma}_{g,\mathbb{N}}(q_1, \cdots, q_\mathbb{N}), \tag{83}$$

where $\hat{\Sigma}_{g,\mathbb{N}}$ is a genus-$g$ closed Riemann surface with $\mathbb{N}$ orbifold points. The Seifert fibration is fully determined by the following Seifert invariants:

$$\begin{aligned}
\mathcal{M} &\cong [\mathrm{d}; g; (q_1, p_1), \cdots, (q_\mathbb{N}, p_\mathbb{N})] \\
&\cong [0; g; (1, \mathrm{d}), (q_1, p_1), \cdots, (q_\mathbb{N}, p_\mathbb{N})],
\end{aligned} \tag{84}$$

with $\mathrm{d}$ the degree of the circle bundle and $(q_i, p_i)$ ($i = 1, \cdots, \mathbb{N}$) the so-called Seifert invariants of the exceptional fibers over the $\mathbb{Z}_{q_i}$-orbifold points. By convention, we have $q_i \geq 1$ (and $q_i > 1$ for a non-trivial orbifold point) and $p_i \in \mathbb{Z}$. Moreover, we generally treat the 'degree' $\mathrm{d}$ as a trivial orbifold point $(q_0, p_0) = (1, \mathrm{d})$. This corresponds to the identity

$$\mathcal{G}_{1,\mathrm{d}}(u) = \mathcal{F}(u)^\mathrm{d}, \tag{85}$$

between fibering operators, giving us a degree-$d$ fibration. The neighbourhood of each exceptional fibre is a solid fibred torus $T(q, t)$, with $pt = 1 \bmod q$. The class of geometries (84) is rather rich. Many explicit examples of supersymmetric backgrounds on Seifert-fibred three-manifolds were discussed in [126]. For instance, this class includes all lens spaces, various homology spheres such as the Poincaré homology sphere, and some torus bundles. We will discuss of few of these examples below.

Each $(q, p)$ exceptional Seifert fibre can be obtained by a Dehn surgery at the orbifold point, starting with a smooth point on the base $\Sigma$, removing a tubular neighbourhood $\mathcal{N} \cong T(1,0)$ of the regular fiber, and gluing back a solid torus with a prescribed $\mathrm{SL}(2, \mathbb{Z})$ twist. Let $(\psi, \varphi)$ denote the coordinates along the longitude and meridian of the boundary of $\mathcal{M} - \mathcal{N}$, and let $(\psi', \varphi')$ be the corresponding coordinates on the solid torus that we glue back in. The longitudes and meridians are related as:

$$L' = qL + tM, \qquad M' = pL - sM, \qquad pt + qs = 1. \tag{86}$$

In terms of the local coordinates, this corresponds to:

$$\begin{pmatrix} \psi \\ -\varphi \end{pmatrix} = \widetilde{U} \begin{pmatrix} \psi' \\ \varphi' \end{pmatrix}, \qquad \widetilde{U} = \begin{pmatrix} s & -p \\ t & q \end{pmatrix}, \tag{87}$$

which is the matrix anticipated in (42). Note that $p = 0$ (and hence $q = s = 1$) is the trivial gluing (the remaining parameter $t$ then gives us a trivial twist of the solid torus as in (82), and does not affect the topology of the 3-manifold).

In the following, it will be useful to define the matrix:

$$U = \begin{pmatrix} p & s \\ -q & t \end{pmatrix} = -\widetilde{U}S\,, \tag{88}$$

which we also denote by $U = U^{(q,p)}$ to emphasise the dependence on the Seifert invariants $(q, p)$. The SL$(2, \mathbb{Z})$ matrix (88) can be written in terms of the $S$ and $T$ matrices (80) as [132]:

$$U^{(q,p)} = \pm T^{m_\ell} S \cdots T^{m_1} S\,, \tag{89}$$

where the integers $(m_1, \cdots, m_\ell)$ give us the partial fraction decomposition of $-p/q$ as follows:

$$-\frac{p}{q} = m_\ell - \cfrac{1}{m_{\ell-1} - \cfrac{1}{\ddots - \frac{1}{m_1}}}\,, \tag{90}$$

and the overall sign in (89) is chosen so that $q > 0$. Note the ambiguity:

$$U^{(q,p)} = \begin{pmatrix} p & s \\ -q & t \end{pmatrix} \quad \longrightarrow \quad U^{(q,p)} T^{m_0} = \begin{pmatrix} p & s + m_0 p \\ -q & t - m_0 q \end{pmatrix}\,, \tag{91}$$

which does not affect the topology of $\mathcal{M}$.[19]

**Seifert fibering operator from topological surgery.** Given this explicit topological description of the $(q, p)$ fiber, we can revisit the description of the Seifert fibering operator given above (46). For the supersymmetric CS theory $\widetilde{G}_K$, since the $S_{\mu\alpha}$ matrix that expands Wilson lines into Bethe vacua is the modular $S$-matrix, we find the matrix elements:

$$\mathcal{U}_{\mu 0} = \langle 0 | \widetilde{\mathcal{U}} | \hat{u}_\mu \rangle = \sum_\rho S_{0\rho} \langle \hat{u}_\rho | \widetilde{\mathcal{U}} | \hat{u}_\mu \rangle = (\widetilde{\mathcal{U}}S)_{\mu 0}\,, \qquad \text{for} \quad \widetilde{\mathcal{U}}_{\mu\rho} \equiv \langle \hat{u}_\rho | \widetilde{\mathcal{U}} | \hat{u}_\mu \rangle\,, \tag{92}$$

where we used the fact that $S_{\lambda\mu}$ is symmetric. Note that the $S$ matrix is needed to expand the cap $\langle 0 |$ into Bethe vacua (28). More generally, we can consider the matrix element:

$$\mathcal{U}_{\mu\lambda} = \langle W_\lambda | \widetilde{\mathcal{U}} | \hat{u}_\mu \rangle = (\widetilde{\mathcal{U}}S)_{\mu\lambda}\,, \tag{93}$$

which corresponds to the insertion of a supersymmetric Wilson loop $W_\lambda^{-1}$ at the exceptional Seifert fiber.[20] Of course, the matrix $\mathcal{U}$ realises the SL$(2, \mathbb{Z})$ action (88) on the Bethe vacua. It can then be constructed as in (89) using the known $S$ and $T$ modular matrices:

$$\mathcal{U}_{\text{TQFT}} = T^{m_\ell} S \cdots T^{m_1} S C^{\frac{1 \mp 1}{2}}\,. \tag{94}$$

Here we denoted this matrix by $\mathcal{U}_{\text{TQFT}}$, the '3d TQFT' $\mathcal{U}$ matrix, to emphasise that the matrix constructed in this way depends on $U$ and not just on $(q, p)$, namely it depends on the specific choice of $t$ as well.[21] A completely explicit expression for this $\mathcal{U}_{\text{TQFT}}$ is known from the CS literature [132, 133, 139]. It reads:

$$(\mathcal{U}_{\text{TQFT}})_{\mu\lambda} = \frac{1}{q^{\frac{\text{rank}(\mathfrak{g})}{2}}} \sum_{\mathfrak{n} \in \Lambda_{\text{mw}}^{\widetilde{G}}(q)} (\mathcal{U}_{\text{TQFT}}^{(\mathfrak{n})})_{\mu\lambda}\,, \tag{95}$$

---

[19]The choice of $t$ does not affect the topology of the three-manifold but it does affect its framing.

[20]Here $W_\lambda^{-1} = W_{\bar{\lambda}}$ denotes the Wilson loop in the representation $\mathfrak{R}_{\bar{\lambda}} = \bar{\mathfrak{R}}_\lambda$ conjugate to $W_\lambda$, or equivalently the Wilson loop $W_\lambda$ wrapping the $S_A^1$ fibre with the opposite orientation with respect to (53).

[21]We can take (89) together with (90) as defining $t$, but we could also choose another $t$ by turning on the parameter $m_0$ in (91). Note also that, in order to compute the matrix element $\mathcal{U}_{\mu 0}$, the overall power of the central element $C$ does not matter since $C_{\mu 0} = \delta_{\mu 0}$. More generally, choosing a different sign in (89) would correspond to replacing $\mathcal{U}_{\mu\lambda}$ by $\mathcal{U}_{\mu\bar{\lambda}}$, or equivalently to flipping the orientation of the Wilson loop in (93).

where we sum over elements of the mod-$q$ reduction of the magnetic flux lattice for the simply-connected gauge group $\widetilde{G}$:

$$\Lambda_{\mathrm{mw}}^{\widetilde{G}}(q) \cong \frac{\Lambda_{\mathrm{mw}}^{\widetilde{G}}}{q\Lambda_{\mathrm{mw}}^{\widetilde{G}}}, \tag{96}$$

with the summands:

$$
\begin{aligned}
(\mathcal{U}_{\mathrm{TQFT}}^{(\mathfrak{n})})_{\mu\lambda} = {}& \mathrm{A}_S \, e^{-\frac{\pi i}{12}\Phi(U)\dim(\mathfrak{g})} \sum_{w\in W_{\mathfrak{g}}} \epsilon(w) \\
& \times \exp\Big[ -\frac{\pi i}{qK}\Big( p\|\rho_{\mathrm{W}}+\mu\|^2 - 2(\rho_{\mathrm{W}}+\mu, w(\rho_{\mathrm{W}}+\lambda)+2K\mathfrak{n}) \\
& + t\|w(\rho_{\mathrm{W}}+\lambda)+2K\mathfrak{n}\|^2 \Big)\Big],
\end{aligned}
\tag{97}
$$

and with the prefactor $\mathrm{A}_S$ defined in (72) and the notation $\|\mu\|^2 = (\mu,\mu)$ for the norm squared of $\mu$, using the (inverse) Killing form.[22] We also introduced the Rademacher function for the $SL(2,\mathbb{Z})$ matrix $U$ [132]:

$$\Phi(U) \equiv -\frac{p+t}{q} + 12\,\mathbf{s}(p,q), \qquad \mathbf{s}(p,q) \equiv \frac{1}{4q}\sum_{l=1}^{q-1}\cot\left(\frac{\pi l}{q}\right)\cot\left(\frac{\pi l p}{q}\right), \tag{98}$$

where $\mathbf{s}(p,q)$ is the so-called Dedekind sum. Using the Weyl character formula, we can massage the expression (95)-(97) into the more suggestive form:

$$(\mathcal{U}_{\mathrm{TQFT}})_{\mu\lambda} = q^{-\frac{\mathrm{rank}(\mathfrak{g})}{2}} \sum_{\mathfrak{n}\in\Lambda_{\mathrm{mw}}^{\widetilde{G}}(q)} e^{-\frac{2\pi i t}{q}h_\lambda}\mathrm{ch}_\lambda\left(e^{\frac{2\pi i}{q}(\hat{u}_\mu-t\mathfrak{n})}\right) \times (\mathcal{U}_{\mathrm{TQFT}}^{(\mathfrak{n})})_{\mu 0}, \tag{99}$$

with:

$$
\begin{aligned}
(\mathcal{U}_{\mathrm{TQFT}}^{(\mathfrak{n})})_{\mu 0} = {}& e^{\frac{\pi i t}{12q}c(\widetilde{G}_k)}\mathrm{A}_S\,(\mathcal{G}_{q,p}^{(0)})^{\dim(\widetilde{G})}\, e^{-\frac{\pi i K}{q}(p(\hat{u}_\mu,\hat{u}_\mu)-2(\mathfrak{n},\hat{u}_\mu)+t(\mathfrak{n},\mathfrak{n}))} \\
& \times e^{-\frac{2\pi i}{q}\rho_{\mathrm{W}}(\hat{u}_\mu-t\mathfrak{n})}\prod_{\alpha\in\Delta^+}\left(1-e^{\frac{2\pi i}{q}\alpha(\hat{u}_\mu-t\mathfrak{n})}\right),
\end{aligned}
\tag{100}
$$

in terms of the Bethe vacua $\hat{u}_\mu = (\rho_{\mathrm{W}}+\mu)/K$. Here we defined the phase factor:

$$\mathcal{G}_{q,p}^{(0)} = e^{\pi i\left(\frac{p}{12q}-\mathbf{s}(p,q)\right)} = e^{-\frac{\pi i}{12}\left(\Phi(U)+\frac{t}{q}\right)}, \tag{101}$$

which happens to be the contribution of a single gaugino to the Seifert fibering operator [126]. Let us now introduce the following modified $\mathcal{U}$ matrix:

$$(\mathcal{U}_{\mathrm{SUSY}})_{\mu\lambda} = e^{\frac{2\pi i t}{q}\left(h_\lambda-\frac{c(\widetilde{G}_k)}{24}\right)}(\mathcal{U}_{\mathrm{TQFT}})_{\mu\lambda}, \tag{102}$$

which is defined in such a way that the explicit $t$ dependence drops out. We will now check that $\mathcal{U}_{\mathrm{SUSY}}$ is indeed the correct supersymmetric result.

### 2.3.4 Seifert fibering operators in the supersymmetric CS theory

The Seifert fibering operator $\mathcal{G}_{q,p}$ is the natural generalisation of the ordinary fibering operator (with $\mathcal{F}=\mathcal{G}_{1,1}$). It allows us to introduce a $(q,p)$ Seifert fibre in a supersymmetric fashion [126]. Recall that the Seifert fibre is associated to the $SL(2,\mathbb{Z})$ matrix $U$ in (88) that we would use for 3d surgery. In our $\mathcal{N}=2$ CS theory, the off-shell Seifert fibering operator

---

[22]We also slightly abuse notation when we write down expressions like (97) which contain some weights $\lambda$ and some fluxes $\mathfrak{n}$, which live in dual spaces. It is then understood that $(\mu,\mathfrak{n}) \equiv \mu(\mathfrak{n})$.

is given as a sum over the 'orbifold fluxes' $\mathfrak{n}$ (called 'fractional fluxes' in [126]) taking value in (96):

$$\mathcal{G}_{q,p}(u) = q^{-\frac{\mathrm{rank}(\mathfrak{g})}{2}} \sum_{\mathfrak{n} \in \Lambda^{\widetilde{G}}_{\mathrm{mw}}(q)} \mathcal{G}_{q,p}(u)_{\mathfrak{n}}, \tag{103}$$

with

$$\mathcal{G}_{q,p}(u)_{\mathfrak{n}} = (\mathcal{G}^{(0)}_{q,p})^{\dim(\widetilde{G})} e^{-\frac{\pi i K}{q}(p(u,u) - 2(\mathfrak{n},u) + t(\mathfrak{n},\mathfrak{n}))} \frac{e^{-\frac{2\pi i}{q}\rho_{\mathrm{W}}(u - t\mathfrak{n})}}{e^{-2\pi i \rho_{\mathrm{W}}(u)}} \prod_{\alpha \in \Delta^+} \frac{1 - e^{\frac{2\pi i}{q}\alpha(u - t\mathfrak{n})}}{1 - e^{2\pi i \alpha(u)}}, \tag{104}$$

where the first exponential arises from the level-$K$ supersymmetric Chern–Simons action itself, and the product over roots arises from one-loop fluctuations of the vector multiplet. The expression (104) is independent of the choice of $t$ such that $pt = 1 \bmod q$.[23]

**Comparison to the 3d TQFT result.** According to general 3d TQFT arguments reviewed above, the Seifert fibering operator in a 3d TQFT (the bosonic CS theory $\widetilde{G}_k$) should act diagonally on the Bethe states as:

$$\mathcal{G}_{q,p}|\hat{u}_{\mu}\rangle = \frac{\mathcal{U}^{(q,p)}_{\mu 0}}{S_{0\mu}}|\hat{u}_{\mu}\rangle. \tag{105}$$

By direct comparison between the TQFT and supersymmetric result for the $\mathcal{N} = 2$ CS theory $\widetilde{G}_K$, we easily check that the on-shell fibering operator matches with what is expected from the TQFT only if we use what we called the $\mathcal{U}_{\mathrm{SUSY}}$ matrix elements defined in (102), namely:

$$\mathcal{G}_{q,p}(\hat{u}_{\mu}) = \frac{(\mathcal{U}^{(q,p)}_{\mathrm{SUSY}})_{\mu 0}}{S_{0\mu}} = e^{-\frac{\pi i t}{12q}c(\widetilde{G}_k)}\frac{\mathcal{U}^{(q,p)}_{\mu 0}}{S_{0\mu}}, \tag{106}$$

where, from now on, we denote by $\mathcal{U} = \mathcal{U}_{\mathrm{TQFT}}$ the 'proper' TQFT matrix (94). This relation can be extended to the insertion of a Wilson loop $W_{\lambda}^{-1}$ wrapping the exceptional fiber. Indeed, from supersymmetric localisation we know that the Wilson loop at that orbifold point gives us the character $\mathrm{ch}_{\lambda}(e^{2\pi i\frac{u - t\mathfrak{n}}{q}})$, where the argument is properly gauge invariant under $(u, \mathfrak{n}) \to (u + 1, \mathfrak{n} + p)$ [126]. We then have the on-shell insertion:

$$W_{\lambda}^{-1}(\hat{u}_{\mu})\mathcal{G}_{q,p}(\hat{u}_{\mu}) = \frac{(\mathcal{U}^{(q,p)}_{\mathrm{SUSY}})_{\mu\lambda}}{S_{0\mu}} = T^{\frac{t}{q}}_{\lambda\lambda}\frac{\mathcal{U}^{(q,p)}_{\mu\lambda}}{S_{0\mu}}, \tag{107}$$

where the second equality follows from (102).

In hindsight, the fact that the TQFT and supersymmetric computations give slightly different answers was to be expected. Indeed, in the supersymmetric case we consider a 'partially twisted' theory which is not exactly topological: the partition function can in principle depend on the 3d metric in a mild fashion, through an explicit dependence on the choice of transversely holomorphic foliation (THF), also known as a dependence on the 'squashing parameter' in the case of a lens space [97, 156]. On the other hand, the TQFT computation is truly topological – that is, metric-independent. Recall that, in the case of the bosonic Chern–Simons theory as discussed by Witten [130], one introduced a gravitational Chern–Simons counterterm proportional to $c_k(\widetilde{G})$ to cancel out an explicit metric dependence of the 'naive' quantum theory, at the price of introducing a dependence on the framing of $\mathcal{M}$. It is thus natural to interpret the relative phase in (106) as the contributions from that same counterterm with an opposite sign, thus removing the dependence on framing and reintroducing some mild metric dependence. A similar story should hold for the Wilson-loop insertions, which are unambiguously fixed in

---

[23]Indeed the shift $t \to t + q$ leaves the answer invariant because $(\mathfrak{n}, \mathfrak{n}) \in 2\mathbb{Z}$.

the supersymmetric context while they can depend on a choice of framing of the loops in the TQFT. We will give some non-trivial evidence for the correctness of this interpretation in the following.

## 2.4 Supersymmetric CS partition functions on Seifert manifolds

Consider the supersymmetric partition function on the Seifert 3-manifold (84). It is given by the insertion of the Seifert fibering operator (48) in the 3d $A$-model:

$$Z_{\mathcal{M}}^{\text{SUSY}}[\widetilde{G}_K] = \left\langle \mathcal{G}_{\mathcal{M}} \right\rangle_{\Sigma_g} = \sum_{\mu} \mathcal{H}(\hat{u}_{\mu})^{g-1} \prod_{i=0}^{N} \mathcal{G}_{q_i,p_i}(\hat{u}_{\mu}). \tag{108}$$

On the other hand, the TQFT computation using Dehn surgery at the exceptional fibers gives us:

$$Z_{\mathcal{M}}^{\text{TQFT}} = \sum_{\mu} S_{0\mu}^{2-2g-N-1} \prod_{i=0}^{N} \mathcal{U}_{\mu 0}^{(q_i,p_i)}, \tag{109}$$

as expected from (49). From the above discussion, we find that:

$$Z_{\mathcal{M}}^{\text{SUSY}} = e^{-S_{\text{ct}}} Z_{\mathcal{M}}^{\text{TQFT}}, \tag{110}$$

with the non-trivial counterterm:

$$e^{-S_{\text{ct}}} = \exp\left(-\frac{\pi i}{12} c(\widetilde{G}_k) \sum_{i=0}^{N} \frac{t_i}{q_i}\right). \tag{111}$$

As we just discussed, this counterterm should remove the framing anomaly of the TQFT partition function, giving us a supersymmetric partition function that depends mildly on the choice of metric. On general grounds, we expect that this counterterm can be written in terms of the supergravity background fields that define the half-BPS Seifert geometry, as an 'almost' local functional involving the gravitational Chern–Simons term as well as Chern–Simons terms for the $U(1)_R$ gauge field (or perhaps the $\eta$ invariant $\eta(\mathcal{M}, A^{(R)})$) and other auxiliary fields [173]. We know that this functional cannot be supersymmetric, since it allows us to go from a supersymmetric to a TQFT scheme, which makes it much harder to pin it down explicitly. We leave realising $S_{\text{ct}}$ as an explicit local functional in the background fields as a challenge for future work.

More generally, we can compute the correlation function of Wilson loops $W_{\lambda_k}$ wrapping generic fibers at distinct smooth points $z_k \in \hat{\Sigma}_{g,N}$, as well as Wilson loops $W_{\lambda_i}^{-1}$ wrapping the exceptional $(q_i, p_i)$ fibers, through the Bethe-vacua formula:

$$\left\langle \prod_k W_{\lambda_k} \times \prod_i W_{\lambda_i}^{-1} \right\rangle_{\mathcal{M}}^{\text{SUSY}} = \sum_{\mu} \mathcal{H}(\hat{u}_{\mu})^{g-1} \prod_k \text{ch}_{\lambda_k}(e^{-2\pi i u_{\mu}}) \times \prod_{i=0}^{N} \mathcal{G}_{q_i,p_i}[W_{\lambda_i}^{-1}](\hat{u}_{\mu}), \tag{112}$$

where we defined the Seifert fibering operator with a Wilson line inserted:

$$\mathcal{G}_{q,p}[W_{\lambda}^{-1}](u) = q^{-\frac{\text{rank}(\mathfrak{g})}{2}} \sum_{\mathfrak{n} \in \Lambda_{\text{mw}}^{\widetilde{G}}(q)} \text{ch}_{\lambda}(e^{\frac{2\pi i}{q}(u-t\mathfrak{n})}) \mathcal{G}_{q,p}(u)_{\mathfrak{n}}, \tag{113}$$

as an obvious generalisation of (103). The 3d TQFT result for the same observable reads:

$$\left\langle \prod_k W_{\lambda_k} \times \prod_i W_{\lambda_i}^{-1} \right\rangle_{\mathcal{M}}^{\text{TQFT}} = \sum_{\mu} S_{0\mu}^{2-2g-N-1} \prod_k \frac{S_{\lambda_k \mu}}{S_{0\mu}} \times \prod_{i=0}^{N} \mathcal{U}_{\mu\lambda_i}^{(q_i,p_i)}, \tag{114}$$

and we thus find:

$$\left\langle \prod_k W_{\lambda_k} \times \prod_i W_{\lambda_i}^{-1} \right\rangle_{\mathcal{M}}^{\text{SUSY}} = e^{-S_{\text{ct}}} e^{\sum_i \frac{2\pi i t_i}{q_i} h_{\lambda_i}} \left\langle \prod_k W_{\lambda_k} \times \prod_i W_{\lambda_i}^{-1} \right\rangle_{\mathcal{M}}^{\text{TQFT}}. \tag{115}$$

Note again that the insertion of the supersymmetric Wilson lines at an exceptional $(q,p)$ fibre differs from the insertion of ordinary Wilson lines in the bosonic CS theory by a non-trivial power of the topological spin $\theta_\lambda = e^{2\pi i h_\lambda}$:

$$W_\lambda^{-1}|_{\text{SUSY}} = \theta_\lambda^{\frac{t}{q}} W_\lambda^{-1}|_{\text{TQFT}}. \tag{116}$$

This is necessary in order to remove the dependence on the framing of the loop – indeed, the factor of $T^{\frac{t}{q}}$ is known to arise from the two-framing of a Wilson loop[24] wrapping the exceptional fiber [143]. The Wilson loops $W_{\lambda_k}$ wrapping regular fibers also have a specific framing, as written above, which is in a sense dictated by supersymmetry. We can always view the insertion of the Wilson loop as a trivial Dehn surgery ($p = 0$) with a loop inserted, and we have then the freedom of choosing $m_0 = -t_k$ in:

$$\mathcal{U}_{\mu\lambda_k}^{(1,0)} = (ST^{-t_k})_{\mu\lambda_k}, \tag{117}$$

which is the freedom to shift the two-framing of the loop. This multiplies (114) by a factor of $(T_{\lambda_k\lambda_k})^{-t_k}$ for each $W_{\lambda_k}$ [130].

To conclude this section, let us now discuss a few instructive special cases. We first discuss lens spaces, which are the Seifert fibrations over the two-sphere with at most two exceptional fibers, and in particular we briefly discuss Wilson loops on $S^3$. We then consider the equally interesting case of torus bundles over the circle which admit a Seifert structure.

### 2.4.1 Supersymmetric CS theory on lens spaces

Consider the supersymmetric lens space $L(p,q)_b$, where $b$ is a continuous squashing parameter. This supersymmetric background admits a Seifert fibration for:

$$b^2 = \frac{q_1}{q_2} \in \mathbb{Q}. \tag{118}$$

Indeed, any $g = 0$ Seifert manifold with at most two exceptional fibers is a lens space, and we have [126]:

$$L(p,q)_b \cong [0;0;(q_1,p_1),(q_2,p_2)], \qquad p = p_1 q_2 + p_2 q_1, \quad q = q_1 s_2 - p_1 t_2, \tag{119}$$

as well as:[25]

$$q' = q_2 s_1 - p_2 t_1, \qquad q q' = 1 \bmod p. \tag{120}$$

One can easily prove that:

$$\frac{t_1}{q_1} = \frac{b^{-2} - q'}{p}, \qquad \frac{t_2}{q_2} = \frac{b^2 - q}{p}, \tag{121}$$

from which one finds:

$$Z_{L(p,q)_b}^{\text{SUSY}} = \exp\left( -\frac{\pi i \, c(\widetilde{G}_k)}{12} \left( \frac{b^2 + b^{-2} - q - q'}{p} \right) \right) Z_{L(p,q)}^{\text{TQFT}}. \tag{122}$$

We see that, in this concrete example, the counterterm $S_{\text{ct}}$ depends on both the topology of the lens space and on the squashing parameter $b$, while the TQFT answer is of course independent of $b$.

---

[24]That is, a choice of two vector fields normal to the loop and respecting the Seifert fibration structure.

[25]Note that $L(p,q)$ and $L(p,q')$ are homeomorphic up to an orientation reversal.

**Wilson loops on the three-sphere.** As a simple special case, consider $L(1,1)_1 = S^3$, the round three-sphere ($b = 1$), which can be obtained using a single ordinary fibering operator. Thus, considering a single Seifert fibering operator with $(q,p) = (1,1)$, we have:

$$U = T^{-1}SCT^{-t} = \begin{pmatrix} 1 & m_0 \\ -1 & t \end{pmatrix}, \qquad \frac{\mathcal{U}_{\mu 0}}{S_{0\mu}} = T_{\mu\mu}^{-1}T_{00}^{-t}, \tag{123}$$

with $m_0 = -t$. Then, according to (114), the two-point function of Wilson loops wrapping the Hopf fibre (giving us a standard Hopf link) is given by:

$$\langle W_\nu W_\lambda \rangle_{S^3}^{\text{TQFT}} = \sum_\mu S_{\nu\mu}S_{\lambda\mu}\frac{\mathcal{U}_{\mu 0}}{S_{0\mu}} = T_{00}^{-t} T_{\nu\nu}S_{\nu\lambda}^{-1}T_{\lambda\lambda}, \tag{124}$$

where we used the SL(2, $\mathbb{Z}$) relations to obtain the final answer. Up to a choice of framing of $S^3$ and of the loops, this is of course the standard answer, $S_{\nu\lambda}^{-1}$, for a Hopf link [130].[26] The supersymmetric answer (115) then removes the explicit $t$-dependence, giving us $(TS^{-1}T)_{\nu\lambda}$.

Another interesting case of Wilson loops on the three-sphere is that of the torus knot. The $(\mathbf{p},\mathbf{q})$ torus knot $T_{\mathbf{p},\mathbf{q}}$ is a knot lying on the surface of an unknotted torus in $\mathbb{R}^3$, specified by two coprime integers $\mathbf{p}$ and $\mathbf{q}$ that describe its winding around the surface [189]. Like any knot, it has an associated Jones polynomial, given by (in a normalisation with $V_{\text{Unknot}}(t) = 1$)

$$V_{T_{\mathbf{p},\mathbf{q}}}(t) = t^{\frac{(\mathbf{p}-1)(\mathbf{q}-1)}{2}} \frac{1 - t^{\mathbf{p}+1} - t^{\mathbf{q}+1} + t^{\mathbf{q}+\mathbf{p}}}{1 - t^2}, \tag{125}$$

which should be reproduced by computing the one-point function of an appropriate fundamental Wilson loop in SU(2) Chern-Simons theory [130]. To see how to obtain this quantity using our formalism, we use the fact that the knot complement of $T_{\mathbf{p},\mathbf{q}}$ is the squashed three-sphere $S_b^3$, viewed as a Seifert fibration over the spindle $S^2(\mathbf{p},\mathbf{q})$. Therefore, to reproduce the above, we simply need to compute the one-point function of a Wilson loop wrapping a generic fibre on $S_b^3$, with Seifert invariants $q_1 = \mathbf{p}, q_2 = \mathbf{q}$, and $p_i$ chosen such that

$$q_1 p_2 + q_2 p_1 = 1 \tag{126}$$

(recalling that $S_b^3 = L(1,1)_b$). Again using (114), we obtain

$$\begin{aligned}
\langle W_1 \rangle_{S_b^3}^{\text{TQFT}} &= \sum_\mu \frac{S_{1\mu}}{S_{0\mu}} \mathcal{U}_{\mu 0}^{(q_1,p_1)} \mathcal{U}_{\mu 0}^{(q_2,p_2)} \\
&= \sum_\mu \frac{\sin(\frac{2\pi\mu}{K})}{\sin(\frac{\pi\mu}{K})} \prod_{j=1}^2 \frac{ie^{-\frac{i\pi}{4}}\Phi(U^{(q_j,p_j)})}{\sqrt{2K}} \\
&\quad \times \sum_{\mathfrak{n}_j=0}^{q_j-1} \sum_{\epsilon_j=\pm 1} \epsilon_j \, e^{\frac{\pi i}{2Kq_j}\left[p_j\mu^2 - 2(2K\mathfrak{n}_j - \epsilon_j)\mu + t_j(2K\mathfrak{n}_j - \epsilon_j)^2\right]},
\end{aligned} \tag{127}$$

which is precisely the result obtained in [137] for the same quantity (up to normalisation). This expression is simplified greatly in [143] to

$$\langle W_1 \rangle_{S_b^3}^{\text{TQFT}} = \frac{1}{2i}\sqrt{\frac{2}{K}}\, t^{-\frac{1}{2}(q_1 q_2 - q_1 - q_2 - 1)}\left[1 - t^{q_1+1} - t^{q_2+1} + t^{q_1+q_2}\right], \tag{128}$$

with $t = e^{-2\pi i/K}$. Upon normalising by the one-point function of the unknot $\langle W_1 \rangle_{S_1^3} = S_{10}$ (from above) and setting $q_1 = \mathbf{p}, q_2 = \mathbf{q}$, this precisely reproduces (125), as expected.

---

[26]That we get $S^{-1}$ and not $S$ is due to our choice of orientation of $S^3$. For $(q,p) = (1,-1)$, we do get $S_{\nu\lambda}$.

### 2.4.2 Supersymmetric CS theory on torus bundles

In order to give a non-trivial consistency check on the 3d *A*-model formalism in a case with more than two exceptional Seifert fibers, it is interesting to consider torus bundles which also admit a Seifert fibration [132, 190]. Torus bundles are non-trivial fibrations of $T^2$ over the circle:

$$\mathcal{M}^A = I \times T^2 / \sim_A, \qquad A \in \mathrm{SL}(2, \mathbb{Z}), \tag{129}$$

with $I = [0, 1]$ the unit interval and with the identification $\{0\} \times T^2 \sim_A \{1\} \times A(T^2)$, where the $T^2$ fibre is identified with itself up to a large diffeomorphism.[27] Identifying the interval $I$ with the time direction, the axioms of TQFT [130] give us the $\mathcal{M}^A$ partition function as a trace:

$$Z_{\mathcal{M}^A}^{\mathrm{TQFT}} = \mathrm{Tr}_{\mathcal{H}_{T^2}}(\mathcal{A}), \tag{130}$$

where $\mathcal{A}$ is the representation of $A$ on the torus Hilbert space, and hence the partition function is literally the trace of the matrix $\mathcal{A}$.

**Torus bundles from surgery on $T^3$.** There are exactly five torus bundles that admit a Seifert fibration, including the trivial fibration [190]:

$$
\begin{aligned}
\mathcal{M}^1 &= [1; 0; ] \cong T^3, \\
\mathcal{M}^C &= [0; 0; (2, 1), (2, 1), (2, -1), (2, -1)], \\
\mathcal{M}^{T^{-1}S} &= [0; 0; (3, 2), (3, -1), (3, -1)], \\
\mathcal{M}^S &= [0; 0; (2, 1), (4, -1), (4, -1)], \\
\mathcal{M}^{ST} &= [0; 0; (2, 1), (3, -1), (6, -1)].
\end{aligned}
\tag{131}
$$

The first one is the three-torus, whose partition function (130) counts the number of lines in the $\widetilde{G}_k$ CS theory, and which is also the Witten index of the 3d $\mathcal{N} = 2$ supersymmetric CS theory $\widetilde{G}_K$. For the pure $\mathrm{SU}(N)_k$ Chern–Simons theory, for instance, we have [191]:

$$Z_{T^3}[\mathrm{SU}(N)_k] = \binom{k + N - 1}{N - 1}. \tag{132}$$

More interestingly, we can consider the partition function on the *C*-twisted torus bundle. According to (130), it gives the trace of the charge conjugation matrix (79), which consequently counts the number of lines that are in self-conjugate representations.[28] Since the charge conjugation matrix is the identity for all compact simple Lie algebras that do not admit intrinsically complex representations, this determines many partition functions:

$$Z_{\mathcal{M}^C}[\widetilde{G}_k] = Z_{T^3}[\widetilde{G}_k], \quad \text{if} \quad \mathrm{Lie}(\widetilde{G}) \in \{\mathfrak{a}_1, \mathfrak{b}_n, \mathfrak{c}_n, \mathfrak{d}_n, \mathfrak{e}_7, \mathfrak{e}_8, \mathfrak{f}_4, \mathfrak{g}_2\}. \tag{133}$$

The only remaining simple algebras are $\mathfrak{a}_n \cong \mathfrak{su}(n+1)$ $(n > 1)$ and $\mathfrak{e}_6$. For $\mathrm{SU}(N)_k$, we count the self-conjugate integrable representations as follows. The conjugate of a Young tableaux is its transpose, which reverses the Dynkin labels as $(\lambda^1, \ldots, \lambda^{N-1}) \mapsto (\lambda^{N-1}, \ldots, \lambda^1)$. Counting the self-conjugate integrable representations for $\mathrm{SU}(N)_k$ thus amounts to counting the reversal

---

[27]The topology of $\mathcal{M}^A$ only depends on the conjugacy class of *A*, not on the specific *A* chosen [190].

[28]More precisely, representations which are real or pseudoreal.

invariant $(N-1)$-tuples whose sum is bounded by $k$. This gives:[29]

$$Z_{\mathcal{M}^C}[\mathrm{SU}(N)_k] = \begin{cases} \begin{pmatrix} \lfloor \frac{k}{2} \rfloor + \frac{N}{2} - 1 \\ \frac{N}{2} - 1 \end{pmatrix} \left(1 + k - 2\frac{N-2}{N}\lfloor\frac{k}{2}\rfloor\right), & N \text{ even}, \\ \begin{pmatrix} \lfloor \frac{k}{2} \rfloor + \frac{N-1}{2} \\ \frac{N-1}{2} \end{pmatrix}, & N \text{ odd}. \end{cases} \tag{134}$$

As anticipated, these values agree perfectly with the supersymmetric calculation described in section 2.4.[30] For the remaining three torus bundles twisted by $T^{-1}S$, $S$ and $ST$, the TQFT partition functions are not generally integers, since the corresponding TQFT matrices are complex. For $\mathrm{SU}(2)_k$, the partition functions for these three torus bundles can be compared with the explicit result [132], which is valid for all non-parabolic elements of $\mathrm{SL}(2,\mathbb{Z})$. Up to a different choice in orientation for the torus bundles, we find precise agreement:

$$Z_{\mathcal{M}^S}[\mathrm{SU}(2)_k] = \delta_{k \bmod 2, 0},$$
$$Z_{\mathcal{M}^{T^{-1}S}}[\mathrm{SU}(2)_k] = \delta_{k \bmod 3, 0} + e^{-\frac{\pi i}{3}}\delta_{k \bmod 3, 1}, \tag{135}$$
$$Z_{\mathcal{M}^{ST}}[\mathrm{SU}(2)_k] = \delta_{k \bmod 3, 0} + e^{+\frac{\pi i}{3}}\delta_{k \bmod 3, 1}.$$

Obtaining simple-looking formulas for all values of $k$ becomes increasingly difficult for larger rank. For instance, we find:

$$Z_{\mathcal{M}^S}[\mathrm{SU}(3)_k] = \delta_{k \bmod 4, 0} - i\delta_{k \bmod 4, 1},$$
$$Z_{\mathcal{M}^{T^{-1}S}}[\mathrm{SU}(3)_k] = \tfrac{1}{2}\left(2 + k - \sqrt{3}i\lfloor\tfrac{k+2}{3}\rfloor - \delta_{k \bmod 3, 2}\right), \tag{136}$$
$$Z_{\mathcal{M}^{ST}}[\mathrm{SU}(3)_k] = \delta_{k \bmod 6, 0} + e^{\frac{2\pi i}{3}}\delta_{k \bmod 6, 1} + e^{-\frac{\pi i}{3}}\delta_{k \bmod 6, 2} + e^{\frac{\pi i}{3}}\delta_{k \bmod 6, 3}.$$

In any case, we find good agreement between the two distinct TQFT computations, and hence with the supersymmetric computation.

# 3 Gauging one-form symmetries on Seifert manifolds

In this section, we generalise the discussion of the previous section to the case of $\mathcal{N} = 2$ supersymmetric Chern–Simons theories $G_K$, where $G$ is not necessarily simply-connected. Denoting by $\widetilde{G}$ the simply-connected group for the simple Lie algebra $\mathfrak{g}$, and by $Z(\widetilde{G})$ its centre, we consider:

$$G = \widetilde{G}/\Gamma, \qquad \Gamma \subset \widetilde{\Gamma} = Z(\widetilde{G}). \tag{137}$$

Conceptually, the easiest way to obtain all possible $G_K$ CS theories is by gauging the corresponding three-dimensional one-form symmetry $\Gamma^{(1)}_{\mathrm{3d}} \cong \Gamma$, which must be a non-anomalous subgroup of the full one-form symmetry $\widetilde{\Gamma}^{(1)}_{\mathrm{3d}}$ of the 3d gauge theory.

In this section, we wish to carry out this gauging in the language of the 3d $A$-model on $\mathcal{M}$, generalising recent discussions that focussed on $\mathcal{M} = \Sigma_g \times S^1_A$ [93–95]. First, however, it is useful to attack the problem from the 3d TQFT perspective. The standard TQFT approach

---

[29]For $N$ odd, this is equal to the number of $\frac{N-1}{2}$-tuples whose sum is bounded by $\lfloor\frac{k}{2}\rfloor$, since adjoining the reverse to those gives all suitable symmetric $(N-1)$-tuples. As a consequence, $Z_{\mathcal{M}^C}[\mathrm{SU}(N)_k] = Z_{T^3}[\mathrm{SU}(\frac{N+1}{2})_{\lfloor\frac{k}{2}\rfloor}]$. For $N$ even, the number of Dynkin labels is odd, and reversal invariant labels take the form $(\lambda^1, \dots, \lambda^{\frac{N}{2}-1}; \lambda^{\frac{N}{2}}; \lambda^{\frac{N}{2}-1}, \dots, \lambda^1)$. Thus we count the number of $(\frac{N}{2}-1)$-tuples with sum $j$, leaving $k+1-j$ choices for $\lambda^{\frac{N}{2}}$, and sum over $j = 0, \dots, \lfloor\frac{k}{2}\rfloor$. The claim follows after a binomial massage.

[30]We checked this numerically for a large number of cases.

to gauging one-form symmetries brings to light some important caveats relevant to the 3d *A*-model approach, which were not fully appreciated before.[31] The upshot is that, in this paper, we will limit ourselves to those 3d $\mathcal{N} = 2$ CS theories $G_K$ that flow to *bosonic* Chern–Simons theories in the infrared. The most general case involves spin-TQFTs as well, and will be treated in detail in a future work [165].

## 3.1 Anyon condensation in bosonic Chern–Simons theories

First of all, we would like to understand which is the most general $\mathcal{N} = 2$ supersymmetric $G_K$ Chern–Simons theory we can consider. Given the $\widetilde{G}_K$ Chern–Simons theory, which flows to the bosonic CS theory $\widetilde{G}_k$ in the infrared with

$$k = K - h^\vee \geq 0\,, \tag{138}$$

we can obtain the gauge group $G = \widetilde{G}/\Gamma$ by a process known as 'anyon condensation' [19, 131, 157, 158]. Let

$$\widetilde{\Gamma}_{3d}^{(1)} \cong Z(\widetilde{G})\,, \tag{139}$$

denote the full one-form symmetry of the $\widetilde{G}_k$ theory, and let $\gamma \in \widetilde{\Gamma}$ denote the group elements. This one-form symmetry is generated by the abelian anyons of the 3d TQFT. These are the Wilson lines $a_\gamma = W_{\lambda_\gamma}$ such that the fusion of $a_\gamma$ with all the other Wilson lines gives us a single Wilson line. Thus we can define the fusion action:

$$a_\gamma W_\mu \cong W_{\gamma(\mu)}\,. \tag{140}$$

In particular, the fusion of the abelian anyons reproduces the group law of the abelian group $\widetilde{\Gamma}_{3d}^{(1)}$, namely

$$a_\gamma a_{\gamma'} \cong a_{\gamma+\gamma'}\,. \tag{141}$$

Let $h[a_\gamma] = h_{\lambda_\gamma}$ denote the conformal spin of the abelian anyon $a_\gamma$, which is given by the conformal dimension (mod 1) of the corresponding WZW[$\widetilde{G}_k$] simple current. The one-form symmetry group $\Gamma_\gamma^{(1)} \subseteq \widetilde{\Gamma}_{3d}^{(1)}$ generated by $\gamma$ is non-anomalous if and only if $h[a_\gamma] \in \frac{1}{2}\mathbb{Z}$. In this paper, we consider the stronger condition:

$$h[a_\gamma] \bmod 1 = 0\,, \qquad \forall \gamma \in \Gamma\,, \tag{142}$$

so that the abelian anyons we condense are all bosonic. This ensures that the infrared Chern–Simons theory $(\widetilde{G}/\Gamma)_k$ remains bosonic. The possible gaugings of non-anomalous subgroups $\Gamma \subseteq \widetilde{\Gamma}$ for all simple simply-connected compact Lie groups are discussed in appendix B. All the possible *bosonic* CS theories for $G$ simple are listed in table 1.

### 3.1.1 Action of the one-form symmetry on the lines

Consider the states generated by Wilson lines and vortex lines of the $\widetilde{G}_k$ theory, as discussed in the previous section:

$$W_\mu \quad = |W_\mu\rangle\,, \qquad V_\mu \quad = |V_\mu\rangle = |\hat{u}_\mu\rangle\,. \tag{143}$$

---

[31]At least not by the present authors.

Table 1: List of all possible bosonic CS theories $G_k$ for $G$ simple. Here $n$ denotes the rank of the gauge group, $k \in \mathbb{Z}$ the CS level, and the conditions (142) for the existence of $G_k$ as a bosonic TQFT are given in the last column. The $(\mathrm{E}_8)_k$, $(\mathrm{F}_4)_k$ and $(\mathrm{G}_2)_k$ CS theories are uniquely determined by their level, and thus are not listed here. See appendix B for more details.

| $\mathfrak{g}$ | $\widetilde{G}_k$ | $\widetilde{\Gamma} = Z(\widetilde{G})$ | $\Gamma \subseteq \widetilde{\Gamma}$ | $G_k = (\widetilde{G}/\Gamma)_k$ | conditions |
|---|---|---|---|---|---|
| $\mathfrak{a}_n$ | $\mathrm{SU}(n+1)_k$ | $\mathbb{Z}_{n+1}$ | $\mathbb{Z}_r$ | $(\mathrm{SU}(n+1)/\mathbb{Z}_r)_k$ | $\frac{k(n+1)(r-1)}{2r^2} \in \mathbb{Z}$ |
| $\mathfrak{b}_n$ | $\mathrm{Spin}(2n+1)_k$ | $\mathbb{Z}_2$ | $\mathbb{Z}_2$ | $\mathrm{SO}(2n+1)_k$ | $\frac{k}{2} \in \mathbb{Z}$ |
| $\mathfrak{c}_n$ | $\mathrm{Sp}(2n)_k$ | $\mathbb{Z}_2$ | $\mathbb{Z}_2$ | $\mathrm{PSp}(2n)_k$ | $\frac{kn}{4} \in \mathbb{Z}$ |
| $\mathfrak{d}_{n=2l+1}$ | $\mathrm{Spin}(4l+2)_k$ | $\mathbb{Z}_4$ | $\mathbb{Z}_4$ | $\mathrm{PSO}(4l+2)_k$ | $\frac{kn}{8} \in \mathbb{Z}$ |
| | | | $\mathbb{Z}_2$ | $\mathrm{SO}(4l+2)_k$ | $\frac{k}{2} \in \mathbb{Z}$ |
| $\mathfrak{d}_{n=2l}$ | $\mathrm{Spin}(4l)_k$ | $\mathbb{Z}_2 \times \widetilde{\mathbb{Z}}_2$ | $\mathbb{Z}_2 \times \widetilde{\mathbb{Z}}_2$ | $\mathrm{PSO}(4l)_k$ | $\frac{k}{2} \in \mathbb{Z}$ & $\frac{kn}{8} \in \mathbb{Z}$ |
| | | | $\mathbb{Z}_2$ | $\mathrm{SO}_+(4l)_k$ | $\frac{kn}{8} \in \mathbb{Z}$ |
| | | | $\widetilde{\mathbb{Z}}_2$ | $\mathrm{SO}_-(4l)_k$ | $\frac{kn}{8} \in \mathbb{Z}$ |
| | | | $\mathbb{Z}_2^{\mathrm{diag}}$ | $\mathrm{SO}(4l)_k$ | $\frac{k}{2} \in \mathbb{Z}$ |
| $\mathfrak{e}_6$ | $(\mathrm{E}_6)_k$ | $\mathbb{Z}_3$ | $\mathbb{Z}_3$ | $(\mathrm{E}_6/\mathbb{Z}_3)_k$ | $\frac{k}{3} \in \mathbb{Z}$ |
| $\mathfrak{e}_7$ | $(\mathrm{E}_7)_k$ | $\mathbb{Z}_2$ | $\mathbb{Z}_2$ | $(\mathrm{E}_7/\mathbb{Z}_2)_k$ | $\frac{k}{4} \in \mathbb{Z}$ |

Let us denote the longitude (non-contractible 1-cycle $S_A^1$) and the meridian (contractible one-cycle) of this solid torus by $\mathcal{A}$ and $\mathcal{B}$, respectively. The abelian anyons act on lines in two ways, by braiding and by fusion. For any 3d TQFT, the braiding phase that one obtains when unlinking $a_\gamma$ and $W_\mu$ is given by [130]:[32]

$$B(a_\gamma, W_\mu) = \frac{S_{\lambda_\gamma \mu}^{-1}}{S_{0\mu}} = \frac{\theta(\gamma(\mu))}{\theta(\mu)\,\theta(\lambda_\gamma)} \equiv \Pi(\hat{u}_\mu)^\gamma\,, \tag{144}$$

where the last equality is just a convenient notation for this braiding phase, for now. We denote the operator obtained by acting with an abelian anyon $a_\gamma$ on the one-cycle $\mathcal{C}$ by $\mathcal{U}^\gamma(\mathcal{C})$. We then have:

$$\mathcal{U}^\gamma(\mathcal{A})|W_\mu\rangle = |W_{\gamma(\mu)}\rangle\,, \qquad \mathcal{U}^\gamma(\mathcal{B})|W_\mu\rangle = \Pi(\hat{u}_\mu)^\gamma |W_\mu\rangle\,, \tag{145}$$

from fusion and linking, respectively. Conversely, the action on vortex loops (and hence on the Bethe states) is:

$$\mathcal{U}^\gamma(\mathcal{A})|\hat{u}_\mu\rangle = \Pi(\hat{u}_\mu)^\gamma |\hat{u}_\mu\rangle\,, \qquad \mathcal{U}^\gamma(\mathcal{B})|\hat{u}_\mu\rangle = |\hat{u}_{\gamma(\mu)}\rangle\,. \tag{146}$$

Note that $\mathcal{U}^\gamma(\mathcal{A})$ is a twisted chiral operator in the $A$-model, which is thus diagonalised by the Bethe vacua, with $\Pi(\hat{u}_\mu)^\gamma$ being our notation for the eigenvalues. The actions (145) and (146) are compatible with (67) provided that:

$$S_{\gamma(\mu)\nu} = \Pi(\hat{u}_\mu)^\gamma S_{\mu\nu}\,. \tag{147}$$

---

[32]The second equality can be derived by considering $W_\mu$ and $a_\gamma$ linked and running along a twisted ribbon.

Indeed, we have the more general relation (in any 3d TQFT, see *e.g.* [159]):

$$S_{\gamma(\mu)\gamma'(\nu)} = B(a_\gamma, a_{\gamma'})B(a_\gamma, W_\mu)B(a_{\gamma'}, W_\nu)S_{\mu\nu}. \tag{148}$$

The self-braiding phases $B(a_\gamma, a_{\gamma'})$ capture the 't Hooft anomaly of $\widetilde{\Gamma}_{3d}^{(1)}$. One the $T^2$ Hilbert space, this anomaly controls the group commutator:

$$\left[ \mathcal{U}^\gamma(\mathcal{A}), \mathcal{U}^{\gamma'}(\mathcal{B}) \right] = \frac{B(a_{\gamma'}, a_{-\gamma}W_\mu)}{B(a_{\gamma'}, W_\mu)} = B(a_\gamma, a_{\gamma'})^{-1}, \tag{149}$$

where in the second equality we compute the braiding of $a_{\gamma'}$ with $a_{-\gamma}W_\mu$ by separating $a_{-\gamma}$ from $W_\mu$ and going through the two lines individually.

**Charge sectors, orbits and twisted-sector states.** The Wilson lines (or, similarly, the Bethe vacua) are organised into sectors and orbits by the action of $\widetilde{\Gamma}_{3d}^{(1)}$, or of any subgroup $\Gamma_{3d}^{(1)}$ thereof. Consider first the action on Wilson lines by linking $a_\gamma$. This defines the one-form symmetry charge $\vartheta_\mu$ of the line $W_\mu$ as an element of the Pontryagin dual group:

$$\vartheta_\mu \in \hat{\Gamma} \equiv \mathrm{Hom}(\Gamma, U(1)), \qquad \text{such that} \quad B(a_\gamma, W_\mu) = \vartheta_\mu(\gamma), \quad \forall \gamma \in \Gamma. \tag{150}$$

The same charge $\vartheta_\mu$ is assigned to the vortex line $V_\mu$ and to the Bethe state $|\hat{u}_\mu\rangle$, by fusing the vortex line with the abelian anyons. In either description, this splits the torus Hilbert space into charge sectors:

$$\mathscr{H}_{T^2} \cong \bigotimes_{\vartheta \in \hat{\Gamma}} \mathscr{H}_{T^2}^\vartheta. \tag{151}$$

It is important to note that this decomposition results from the action of $\mathcal{U}^\gamma(\mathcal{B})$ in the Wilson-line basis but from the action of $\mathcal{U}^\gamma(\mathcal{A})$ in the vortex-line (*i.e.* Bethe-state) basis. In the former description, the states in $\mathscr{H}_{T^2}^\vartheta$ are generated by the set of Wilson lines with fixed $\Gamma_{3d}^{(1)}$ charge:

$$\mathcal{S}_W^\vartheta = \left\{ W_\mu \,\middle|\, \Pi(\hat{u}_\mu)^\gamma = \vartheta_\mu(\gamma), \quad \forall \gamma \in \Gamma \right\}. \tag{152}$$

The set of Wilson lines with vanishing charge – that is, with $\vartheta = 1$ – will be of particular interest since they survive gauging.

Secondly, we consider the fusion of Wilson lines with the abelian anyons $a_\gamma$, or similarly the linking of a vortex line with the same $a_\gamma$. This defines orbits of integrable representations under the action of $\Gamma$:

$$\mu \to \gamma(\mu), \qquad \gamma \in \Gamma. \tag{153}$$

For $\widetilde{G}_k$ based on a simple Lie algebra $\mathfrak{g} = \mathrm{Lie}(\widetilde{G})$, this group action on the highest weights is well understood in terms of the action of the centre of $\widetilde{G}$ on the affine Dynkin diagram of $\mathfrak{g}$ – see *e.g.* [172]. Let us denote by $\hat{\omega}(\mu)$ the $\Gamma$ orbit of some $\mu$, and more generally any orbit by $\hat{\omega}$ (as the orbit is independent of the element $\mu \in \hat{\omega}$):

$$\hat{\omega} = \{\mu_1, \mu_2, \cdots, \mu_{|\hat{\omega}|}\}. \tag{154}$$

Here $|\hat{\omega}|$ denotes the length of the orbit. We denote by $\mathrm{Stab}(\hat{\omega}) \subseteq \Gamma$ the stabiliser of any element $\mu \in \hat{\omega}$, and we recall that $|\mathrm{Stab}(\hat{\omega})| = |\Gamma|/|\hat{\omega}|$ by the orbit-stabiliser theorem.[33] In the Wilson-line basis, we may define the $\Gamma$-invariant orbit states:

$$|W_{\hat{\omega}}\rangle = c_{\hat{\omega}} \sum_{\mu \in \hat{\omega}} |W_\mu\rangle, \tag{155}$$

---

[33] Here we use the fact that $\Gamma$ is abelian, hence $\mathrm{Stab}(\hat{\omega}) \equiv \mathrm{Stab}(\mu)$ is the exact same abelian group for any element $\mu \in \hat{\omega}$. This uneasy notation should not cause any confusion.

with $c_{\hat{\omega}}$ some unspecified normalisation factor. More useful for us will be the following orthonormal states, which are built simply by summing over orbits of Bethe states:

$$|\hat{\omega}\rangle = \frac{1}{\sqrt{|\hat{\omega}|}} \sum_{\mu \in \hat{\omega}} |\hat{u}_{\mu}\rangle, \qquad \mathcal{U}^{\gamma}(\mathcal{B})|\hat{\omega}\rangle = |\hat{\omega}\rangle. \tag{156}$$

Assuming the vanishing of the 't Hooft anomaly, the operators $\mathcal{U}^{\gamma}(\mathcal{A})$ and $\mathcal{U}^{\gamma}(\mathcal{B})$ can be diagonalised simultaneously. In particular, the orbits $\hat{\omega}$ and the states $|\hat{\omega}\rangle$ are then assigned a specific charge $\vartheta \in \hat{\Gamma}$. The states that trivialise all charge operators $\mathcal{U}^{\gamma}(\mathcal{C})$ on $T^2$ are thus the Bethe-state orbits with $\vartheta = 1$.

Finally, we need to consider the twisted-sector states as well. These are states obtained by inserting an abelian anyon $a_{\delta}$ along the time direction [131]:

$$W_{\mu} \; \raisebox{-1em}{\includegraphics{}} \; = |W_{\mu}; \delta\rangle \in \mathscr{H}_{T^2}^{(\delta)}, \tag{157}$$

where $\mathscr{H}_{T^2}^{(\delta)}$ denotes the $\delta$-twisted Hilbert space. The line $W_{\mu}$ can only participate in a $\delta$-twisted state if it is fixed by fusion with $a_{\delta}$, namely if $\delta(\mu) = \mu$.[34] We can similarly generate $\mathscr{H}_{T^2}^{(\delta)}$ using the $\delta$-twisted Bethe states, which are defined similarly in terms of vortex lines:

$$\mathscr{H}_{T^2}^{(\delta)} \cong \mathrm{Span}_{\mathbb{C}} \left\{ |\hat{u}_{\mu}; \delta\rangle \;\middle|\; \delta(\mu) = \mu \right\}, \qquad \delta \in \Gamma. \tag{158}$$

The one-form symmetry charges of the $\delta$-twisted sector states are independent of $\delta$, and thus the same as in the untwisted sector:

$$\mathcal{U}^{\gamma}(\mathcal{B})|W_{\mu}; \delta\rangle = \Pi(\hat{u}_{\mu})^{\gamma}|W_{\mu}; \delta\rangle, \qquad \mathcal{U}^{\gamma}(\mathcal{A})|\hat{u}_{\mu}; \delta\rangle = \Pi(\hat{u}_{\mu})^{\gamma}|\hat{u}_{\mu}; \delta\rangle. \tag{159}$$

The action of $\mathcal{U}^{\gamma}(\mathcal{A})$ on $|W_{\mu}; \delta\rangle$ is seemingly more complicated due to the trivalent junction [163, 192], yet one can argue that the naive fusion gives the correct result, namely:

$$\mathcal{U}^{\gamma}(\mathcal{A})|W_{\mu}; \delta\rangle = |W_{\gamma(\mu)}; \delta\rangle, \qquad \mathcal{U}^{\gamma}(\mathcal{B})|\hat{u}_{\mu}; \delta\rangle = |\hat{u}_{\gamma(\mu)}; \delta\rangle. \tag{160}$$

This is most easily shown in the *A*-model perspective [95]. Since the stabiliser group of $\mu$ is the same for every orbit element $\mu \in \hat{\omega}$, we find that the full orbit $\hat{\omega}$ has a twisted-sector copy for every $\delta \in \mathrm{Stab}(\hat{\omega})$. In particular, we can define the twisted-sector orbit states of Bethe vacua:

$$|\hat{\omega}; \delta\rangle = \frac{1}{\sqrt{|\hat{\omega}|}} \sum_{\mu \in \hat{\omega}} |\hat{u}_{\mu}; \delta\rangle, \qquad \mathcal{U}^{\gamma}(\mathcal{B})|\hat{\omega}; \delta\rangle = |\hat{\omega}; \delta\rangle, \tag{161}$$

generalising (156). These states are orthonormal: $\langle \hat{\omega}'; \delta' | \hat{\omega}; \delta \rangle = \delta_{\hat{\omega}\hat{\omega}'} \delta_{\delta\delta'}$.

### 3.1.2 Condensing the lines: The $G_k$ torus Hilbert space

From now on, consider $\Gamma$ a non-anomalous subgroup of the full one-form symmetry such that the condition (142) holds. On any 3-manifold, the condensation of these bosonic abelian anyons for $\Gamma$ is a three-step process [131]:

---

[34]We note that the trivalent vertex between the $W_{\mu}$ and $a_{\delta}$ lines is unique whenever the line $a_{\delta}$ is bosonic and non-anomalous (*i.e.* a condensable anyon). The trivalent vertex corresponds to a state in the Hilbert space of the sphere with three insertions $W_{\mu}$, $W_{\bar{\mu}}$ and $a_{\delta}$. For a condensable boson $\delta$, we have trivial braiding with all the lines, $B(\delta, \alpha) = 0$, by definition, hence $\frac{S_{\delta\alpha}}{S_{0\alpha}} = 1$ by (144). We then find that $\mathcal{N}_{\mu\nu\delta} = C_{\mu\nu}$ using the Verlinde formula (35).

1. Restrict the set of lines to the ones of vanishing one-form charge. In either the Wilson-line or the vortex line basis, those are the lines $\mathscr{L}_\mu$ indexed by $\mu$ such that:

$$\Pi(u_\mu)^\gamma = 1\,, \qquad \forall \gamma \in \Gamma\,. \tag{162}$$

In other words, one restricts to the $\vartheta = 1$ charge sector. For $\Gamma$ non-anomalous, this includes all its abelian anyons.

2. Identify the lines $\mathscr{L}_\mu$ that belong to the same orbit $\hat{\omega} = \hat{\omega}(\mu)$. In particular, the abelian anyons $a_\gamma$ are identified with the trivial line.

3. Every line $\mathscr{L}_\mu$ is replaced by a set of lines $\mathscr{L}_\mu^{(\delta)}$ where $\delta \in \text{Stab}(\mu)$. In the original $\widetilde{G}_k$ theory, these lines are lines $\mathscr{L}_\mu$ meeting with $a_\delta$ transversely, but those configurations become genuine lines after step 2.

On the solid torus in the Bethe-state basis, for definiteness, these three steps correspond to inserting all the possible abelian anyons along the $\mathcal{A}$, $\mathcal{B}$ and time direction, respectively. The torus Hilbert space of the $G_k = (\widetilde{G}/\Gamma)_k$ theory is obtained by projecting the extended Hilbert space (including all twisted sectors) of the $\widetilde{G}_k$ theory down to the states invariant under $\mathcal{U}^\gamma(\mathcal{C})$:

$$\begin{aligned} \mathcal{H}_{T^2}[G_k] &\cong \bigoplus_{\delta \in \Gamma} \mathcal{H}_{T^2}^{(\delta)}[\widetilde{G}_k] \Big/ \Gamma_\mathcal{A} \times \Gamma_\mathcal{B} \\ &\cong \text{Span}_{\mathbb{C}} \Big\{ |\hat{\omega}; \delta\rangle \,\Big|\, \Pi(\hat{\omega})^\gamma = 1\,,\, \delta \in \text{Stab}(\hat{\omega}) \Big\}\,. \end{aligned} \tag{163}$$

Here $\Gamma_\mathcal{C}$ denotes the $\Gamma$ action generated by the line operators $\mathcal{U}^\gamma(\mathcal{C})$. The $G_k$ theory enjoys a dual zero-form symmetry $\Gamma_{3d}^{(0)} \cong \hat{\Gamma}$ which acts by permutation on the 'twisted-sector' labels $\delta$. In many situations, it is more convenient to consider the following 'charge-$\chi$' states defined as discrete Fourier transforms:

$$|\hat{\omega}; \chi\rangle \equiv \frac{1}{\sqrt{|\Gamma_{\hat{\omega}}|}} \sum_{\delta \in \Gamma_{\hat{\omega}}} \chi(\delta) |\hat{\omega}; \delta\rangle\,, \qquad \text{where } \Gamma_{\hat{\omega}} \equiv \text{Stab}(\hat{\omega}) \text{ and } \chi \in \hat{\Gamma}_{\hat{\omega}}\,. \tag{164}$$

These states diagonalise $\Gamma_{3d}^{(0)}$, with $\chi$, viewed as a character of $\Gamma$, being their $\Gamma_{3d}^{(0)}$ charge; see [163] for a closely related discussion. The corresponding outgoing states are defined to be

$$\langle \hat{\omega}; \chi| \equiv \frac{1}{\sqrt{|\Gamma_{\hat{\omega}}|}} \sum_{\delta \in \Gamma_{\hat{\omega}}} \chi(\delta)^* \langle \hat{\omega}; \delta|\,, \tag{165}$$

so that

$$\langle \hat{\omega}'; \chi' | \hat{\omega}; \chi\rangle = \delta_{\hat{\omega}\hat{\omega}'} \delta_{\chi\chi'}\,, \tag{166}$$

by the orthogonality of characters.

### 3.1.3 Modular matrices of the gauged theory

Combining the condition (142) with (144), it is clear that we have $\theta(\gamma(\mu)) = \theta(\mu)$ for the Wilson lines of vanishing $\Gamma_{3d}^{(1)}$ charge. Hence the orbit states (155) have a definite topological spin. Correspondingly, the $T$-matrix of the $G_k$ theory is given in terms of the one for the $\widetilde{G}_k$ theory as:

$$T_{(\hat{\omega},\chi),(\hat{\omega}',\chi')} = \delta_{\chi\chi'} T_{\mu\nu}\,, \qquad \text{for} \quad \hat{\omega} = \hat{\omega}(\mu)\,, \quad \hat{\omega}' = \hat{\omega}'(\nu)\,, \tag{167}$$

where it is understood that the indices $(\hat{\omega}, \chi)$ run over the states (164). More precisely, we have:

$$T_{(\hat{\omega},\chi),(\hat{\omega}',\chi')} \equiv \langle \hat{\omega}'; \chi' | T | \hat{\omega}; \chi\rangle = \delta_{\chi\chi'} \delta_{\hat{\omega}\hat{\omega}'} T_{\mu\mu}\,, \qquad \text{for} \quad \hat{\omega} = \hat{\omega}(\mu)\,. \tag{168}$$

In particular, the $T$ matrix remains diagonal and encodes the topological spins of the $G_k$ states as expected.

While the $S$-matrix of the $G_k$ theory is a bit more involved, its general structure easily follows from the explicit form of the orthonormal states (164) written in terms of the $\widetilde{G}_k$ states. By definition, we have:

$$S_{(\hat{\omega};\chi),(\hat{\omega}',\chi')} \equiv \langle \hat{\omega}'; \chi'|S|\hat{\omega}; \chi \rangle. \tag{169}$$

Expanding it out, we find:

$$S_{(\hat{\omega},\chi),(\hat{\omega}',\chi')} = \frac{1}{|\Gamma|} \sum_{\delta \in \Gamma_{\hat{\omega}}} \sum_{\delta' \in \Gamma_{\hat{\omega}'}} \chi'(\delta')^* \chi(\delta) \sum_{\mu \in \hat{\omega}} \sum_{\mu' \in \hat{\omega}'} \langle u_{\mu'}; \delta'|S|u_\mu; \delta \rangle. \tag{170}$$

The overlap $\langle u_{\mu'}; \delta'|S|u_\mu; \delta \rangle$ vanishes unless $\delta = \delta'$. Let us thus define the matrix elements:

$$S_{\mu\nu}^\delta \equiv \langle \hat{u}_\nu; \delta|S|\hat{u}_\mu; \delta \rangle, \tag{171}$$

generalising the $S$-matrix $S_{\mu\nu} = S_{\mu\nu}^{\delta=0}$ of the $\widetilde{G}_k$ theory in the $\Gamma_{3d}^{(1)}$-neutral sector. Note that $S_{\gamma(\mu)\gamma'(\nu)} = S_{\mu\nu}$ for the states that survive step 1 and 2 of the anyon condensation process, and the same holds true for the $\delta$-twisted sectors. Hence we can pick any fixed $\mu$ to represent each orbit $\hat{\omega}(\mu)$ and sum over the identical copies. We then immediately find the expression:

$$S_{(\hat{\omega},\chi),(\hat{\omega}',\chi')} = \frac{|\Gamma|}{|\Gamma_{\hat{\omega}}||\Gamma_{\hat{\omega}'}|} \sum_{\delta \in \Gamma_{\hat{\omega}} \cap \Gamma_{\hat{\omega}'}} \chi'(\delta)^* \chi(\delta) S_{\mu\mu'}^\delta. \tag{172}$$

In any theory without twisted sectors (that is, where all the orbits $\hat{\omega}$ are of maximal dimension), this would give us $S_{(\hat{\omega},\chi),(\hat{\omega}',\chi')} = |\Gamma|S_{\mu\nu}$. More generally, using the fact that $\mu = 0$ always has trivial stabiliser (the orbit thereof being the set of $|\Gamma|$ abelian anyons), we see that:

$$S_{0,(\hat{\omega},\chi)} = |\hat{\omega}| S_{0\mu}, \tag{173}$$

in perfect agreement with the $A$-model result of [95]. In particular, since $S_{00}$ gives us the $S^3$ partition function of the 3d TQFT [130], we directly see that:[35]

$$Z_{S^3}[G_k] = |\Gamma| Z_{S^3}[\widetilde{G}_k]. \tag{174}$$

To access the full $S$ matrix, however, we need to compute explicitly the overlaps (171), which corresponds to a Hopf link of $W_\mu$ with $W_\nu$ where the two loops are also connected by the $a_\delta$ line. While it would be interesting to perform this computation explicitly, the final answer for any simple gauge group $G = \widetilde{G}/\Gamma$ has already been painstakingly computed in the context of WZW models [161], and we will simply use those known results. In particular, the matrix (171) can be built in terms of the $S$-matrix of a smaller 'orbit' Lie algebra. We will give the explicit expressions in the case of $\widetilde{G} = \mathrm{SU}(N)$ below.

Given these $S$ and $T$ matrices, we can now build any $\mathrm{SL}(2,\mathbb{Z})$ matrix $U$ exactly as in (94), and we can then construct any Seifert manifold partition function and observables for the $G_k$ bosonic Chern–Simons theories exactly as for the $\widetilde{G}_k$ theories.

### 3.1.4 Example: SU(N)$_k$ Chern–Simons theory

As a concrete example, let us discuss the gauging of centre subgroups for the pure $\mathrm{SU}(N)_k$ Chern–Simons theory. As discussed at length in [95], the $\mathrm{SU}(N)_k$ theory has a non-anomalous $\mathbb{Z}_r$ one-form symmetry if $kN/r^2 \in \mathbb{Z}$. If $r$ is even and $kN/r^2$ is odd, the topological spin of the line generating the $\mathbb{Z}_r$ symmetry is half-integer, and the $\mathbb{Z}_r$ gauging consequently results in a spin-TQFT. In this work, we focus on the bosonic cases, that is $(r-1)kN/r^2$ is even (see appendix B.1).

---

[35]Changing the framing does not affect this relation thanks to the simple relation between $T$ matrices.

**Action of $\mathbb{Z}_N$ on integrable representations.** The lines are then labelled by integrable representations $\lambda = (\lambda^1, \ldots, \lambda^{N-1})$ for $SU(N)_k$, where the integrability condition reads:

$$\sum_{a=1}^{N-1} \lambda^a \le k. \tag{175}$$

These correctly enumerate the vacua of the pure $SU(N)_k$ theory, whose number is given by (132). From these tuples, the Bethe vacua $\hat{u}$ are determined using (59). The fusion of abelian anyons with Wilson lines $W_\lambda$ gives rise to orbits under $\mathbb{Z}_N$, which is equivalent to the 2d 0-form symmetry permuting the Bethe vacua, $\hat{u} \to \hat{u} + \gamma_0$. As before, $\gamma_0$ is a generator of $\mathbb{Z}_N$, which in the fundamental weight basis is given by $\gamma_{0,a} = -\frac{a}{N}$. From this, we can work out the action on the integrable representations. For a general shift $\hat{u} \to \hat{u} + \gamma$ we have to find the corresponding integrable highest-weight after the shift by using large gauge transformations and Weyl invariance. For $\gamma = n\gamma_0$ with $n = 0, \ldots, N-1$, one finds:[36]

$$[\lambda^a] \to \gamma \cdot [\lambda^a] = [\lambda'^a], \qquad \lambda'^a = \begin{cases} \lambda^{N+a-n}, & a = 1, \cdots, n-1, \\ K - N - |\lambda|, & a = n, \\ \lambda^{a-n}, & a = n+1, \cdots, N-1, \end{cases} \tag{176}$$

where $|\lambda| \equiv \sum_{a=1}^{N-1} \lambda^a$. This action is easily pictured in terms of Young tableaux. The $\gamma_0$ shift corresponds to adding the Young tableau $[t_{\gamma_0}^a] = [K-N, 0, \cdots, 0]$ to the top of the Young tableau $[t^a]$ for the highest weight $\lambda = [\lambda^a]$, and simplifying the resulting $SU(N)$ tableau:[37]

$$t^a \to \gamma_0(t^a) = \begin{cases} K - N - t^{N-1}, & a = 1, \\ t^{a-1} - t^{N-1}, & a = 2, \cdots, N-1. \end{cases} \tag{177}$$

This representation also makes it clear that the action of $\gamma$ maps integrable representations to integrable representations, and also that the $\mathbb{Z}_N$ orbit of the trivial representation $\lambda = 0$ is always maximal, consisting of the representations:

$$[0, \cdots, 0], \ [k, 0, \cdots, 0], \ [0, k, 0, \cdots, 0], \ [0, \cdots, 0, k], \tag{178}$$

with $k = K - N$. These are the set of $N$ abelian anyons.

The $\mathbb{Z}_N^{(1)}$ charge of a Wilson line $W_\lambda$ is precisely the $N$-ality of the corresponding representation:[38]

$$\Pi(\hat{u}_\lambda)^{\gamma_0} = e^{-\frac{2\pi i}{N} n(\lambda)}, \tag{180}$$

where the $N$-ality is given by

$$n(\lambda) \equiv \sum_{a=1}^{N-1} a \lambda^a = \sum_{a=1}^{N-1} t^a, \tag{181}$$

with $t^a$ the labels of the corresponding Young tableau (that is, the $N$-ality is the number of boxes in the Young tableau).[39]

---

[36]Note that due to (175) we have $\lambda'^n \ge 0$, as required.

[37]The Dynkin labels $\lambda^a$ are related to Young tableaux $[t^1, \cdots, t^{N-1}]$ (with $t^1 \ge t^2 \ge \cdots \ge t^{N-1}$) as $\lambda^a = t^a - t^{a+1}$, with $t^N \equiv 0$.

[38]Using (177), we can study the action of the 0-form symmetry on the flux operator: The $N$-ality $n(\lambda)$ changes under $\gamma_0$ to $n(\gamma_0 \cdot \lambda) \equiv n(\lambda) + K \mod N$. Thus if $N|K$, the $N$-ality is preserved in each orbit, as required for the gauging procedure [95]. Otherwise, we have

$$\Pi(\hat{u}_\lambda + \gamma_0)^{\gamma_0} = e^{-2\pi i \frac{K}{N}} \Pi(\hat{u}_\lambda)^{\gamma_0}, \tag{179}$$

which is simply the mixed anomaly between $\mathbb{Z}_N^{(0)}$ and $\mathbb{Z}_N^{(1)}$ [95].

[39]The identification (180) fixes an ambiguity in the definition of the gauge flux operator $\Pi^\gamma$ (see equation (208) below), which for the $SU(N)$ theory is determined only up to an $N$-th root of unity. In the $A$-model, the ambiguity can be fixed by demanding 3d modularity for the expectations values of elementary topological lines on $T^3$, as demonstrated in [95].

**Anyon condensation.** It is now straightforward to perform the three-step gauging process, as discussed in section 3.1.2. Consider $\Gamma = \mathbb{Z}_r$, assuming the conditions mentioned above. In Step 1, we restrict ourselves to the $\vartheta = 1$ sector with $\Pi(u_\lambda)^\gamma = 1$, where $\gamma = \frac{N}{r}\gamma_0$ is a generator of $\mathbb{Z}_r$. Due to (180), this amounts to selecting the integrable representations $\lambda$ with

$$n(\lambda) = 0 \mod r. \tag{182}$$

In particular, if we want to gauge the full $\mathbb{Z}_N$ (assuming that $(N-1)k/N$ is even, in this case) then we restrict ourselves to Wilson lines of $N$-ality zero. In Step 2, we consider the $\mathbb{Z}_r$ orbits of Wilson lines, which are easily obtained by a repeated application of (176). This partitions the set of $\vartheta = 1$ lines into orbits whose dimensions are divisors of $r$. In Step 3, we create $r/l$ distinct copies for each orbit of length $l$.

The total number of lines in the resulting $(SU(N)/\mathbb{Z}_r)_k$ theory then agrees perfectly with the $A$-model calculation for the Witten index of the $(SU(N)/\mathbb{Z}_r)_K$ theory, with $K = k + N$ [95]:

$$\mathbf{I}_W[(SU(N)/\mathbb{Z}_r)_K] = \frac{1}{r^2} \sum_{d \mid \gcd(r,K)} J_3(d) \binom{\frac{K}{d}-1}{\frac{N}{d}-1}, \tag{183}$$

where $J_3$ is Jordan's totient function (see [95] for more details).

**Modular matrices.** Finally, we discuss the modular matrices of the gauged theory. As mentioned before, the gauging is understood at the level of orbit Lie algebras of the corresponding WZW model, with the $S$-matrix of the gauged theory being obtained as a sum over the $S$-matrices of the possible orbit Lie algebras under the one-form symmetry.

For simplicity, let us discuss the case of the full $\mathbb{Z}_N$ gauging. For $N$ an arbitrary integer, the stabiliser for the orbits can be any subgroup of $\mathbb{Z}_N$. Let $\mu$ and $\nu$ be two representations giving rise to states in the $SU(N)_k$ theory, with $\hat{\omega} = \hat{\omega}(\mu)$ and $\hat{\omega}' = \hat{\omega}'(\nu)$ their $\mathbb{Z}_N$ orbits, as before. The states in the $PSU(N)_k$ theory are then accordingly labelled by $(\hat{\omega}, \chi)$ and $(\hat{\omega}', \chi')$, see (164). Explicitly, we label the charge-$\chi$ states by some integers $j = 0, \ldots, |\Gamma_{\hat{\omega}}| - 1$, and analogously $j'$ for $\hat{\omega}'$. The intersection $\Gamma_{\hat{\omega}} \cap \Gamma_{\hat{\omega}'}$ is a subgroup of order $\gcd(|\Gamma_{\hat{\omega}}|, |\Gamma_{\hat{\omega}'}|)$, which is generated by $d_{\hat{\omega},\hat{\omega}'}\gamma_0$, with $d_{\hat{\omega},\hat{\omega}'} = N/\gcd(|\Gamma_{\hat{\omega}}|, |\Gamma_{\hat{\omega}'}|)$. Consequently, we parametrise the sum (172) over this intersection as the multiples $l$ of $d_{\hat{\omega},\hat{\omega}'}\gamma_0$, which we denote by $l \in \langle d_{\hat{\omega},\hat{\omega}'}\gamma_0 \rangle$. The orbit algebra for some element $l\gamma_0$ depends only on the subgroup it generates. This gives us [161]:

$$S^{PSU(N)_k}_{(\hat{\omega},\chi),(\hat{\omega}',\chi')} = \frac{N}{|\Gamma_{\hat{\omega}}||\Gamma_{\hat{\omega}'}|} \sum_{l \in \langle d_{\hat{\omega},\hat{\omega}'}\gamma_0 \rangle} e^{\frac{2\pi i(j-j')l}{N}} e^{-\frac{i\pi k(N^2-\gcd(l,N)^2)}{4N}} S^{SU(\gcd(l,N))_{k\gcd(l,N)/N}}_{\tilde{\mu}\tilde{\nu}}. \tag{184}$$

Aside from the phase originating from the characters $\chi(l) = e^{2\pi ijl/N}$, the summands depend only on the value $\gcd(l,N)$. It generates itself a $\mathbb{Z}_{r(l)}$ subgroup, where $r(l) = N/\gcd(l,N)$. Using this, we obtain:

$$S^{PSU(N)_k}_{(\hat{\omega},\chi),(\hat{\omega}',\chi')} = \frac{N}{|\Gamma_{\hat{\omega}}||\Gamma_{\hat{\omega}'}|} \sum_{l \in \langle d_{\hat{\omega},\hat{\omega}'}\gamma_0 \rangle} e^{\frac{2\pi i(j-j')l}{N}} e^{-\frac{\pi ikN}{4}(1-\frac{1}{r(l)^2})} S^{SU(N/r(l))_{k/r(l)}}_{\tilde{\mu}\tilde{\nu}}. \tag{185}$$

This sum over $S$-matrices for theories of smaller rank $N/r(l)$ requires a map between integrable representations for $SU(N)$ and $SU(\frac{N}{d})$, for any divisor $d$ of $N$. This map can be constructed for instance using the observation [95] that the number of $\mathbb{Z}_d$ fixed points of the $SU(N)_k$ theory is equal to the number of vacua of the $SU(\frac{N}{d})_{\frac{k}{d}}$ theory. It is given explicitly in terms of Dynkin

labels by:[40]

$$\sim: \begin{cases} \mathfrak{R}[\mathrm{SU}(N)_k]^{\mathbb{Z}_d} \longrightarrow \mathfrak{R}\left[\mathrm{SU}(\tfrac{N}{d})_{\frac{k}{d}}\right], \\ (\lambda^1, \lambda^2, \dots, \lambda^{\frac{N}{d}}, \lambda^1, \lambda^2, \dots, \lambda^{\frac{N}{d}}, \dots, \lambda^1, \lambda^2, \dots, \lambda^{\frac{N}{d}-1}) \longmapsto (\lambda^1, \lambda^2, \dots, \lambda^{\frac{N}{d}-1}). \end{cases} \tag{186}$$

This isomorphism thus applies precisely to the form (185): If $\mu$ sits inside some orbit $\hat{\omega}$, it has stabiliser $\Gamma_{\hat{\omega}}$ under $\mathbb{Z}_N$, and is consequently a fixed point under $\Gamma_{\hat{\omega}}$. Since in (185) we sum over a subgroup of this stabiliser (the one that fixes $\nu$ as well), we can use (186) to reduce $\mu$ (as well as $\nu$) to the corresponding element $\tilde{\mu}$ (and $\tilde{\nu}$) in the $\mathrm{SU}(\tfrac{N}{d})_{\frac{k}{d}}$ theory, whose $S$-matrix is known.

The $S$-matrix for all the $(\mathrm{SU}(N)/\mathbb{Z}_r)_k$ theories can be obtained analogously, by adjusting the characters $\chi(l)$ to be characters for $\mathbb{Z}_r$ rather than for $\mathbb{Z}_N$, and the stabilisers $\Gamma_{\hat{\omega}}$ will be subgroups of $\mathbb{Z}_r$ rather than $\mathbb{Z}_N$.

## 3.2 One-form symmetries on supersymmetric Seifert manifolds

Let us now return our attention to the *A*-model. The supersymmetric Seifert manifolds (84) are half-BPS backgrounds obtained from $\Sigma_g \times S^1$ by the repeated application of Seifert fibering operators (43) adding in exceptional fibers $(q_i, p_i)$, $i = 0, \cdots, \mathbb{N}$. In the $\widetilde{G}_K$ $\mathcal{N} = 2$ supersymmetric CS theory (and more generally in $\mathcal{N} = 2$ CS-matter theories with gauge group $\widetilde{G}$), we define the *off-shell* Seifert fibering operator as in (103), as a sum over the orbifold flux lattice $\mathfrak{n} \in \Lambda_{\mathrm{mw}}^{\widetilde{G}}(q)$. Our main interest is thus in the object:

$$\mathcal{G}_{q,p}(u)_{\mathfrak{n}}, \qquad u \in \mathfrak{h}_{\mathbb{C}}/W_{\mathfrak{g}}, \qquad \mathfrak{n} \in \Lambda_{\mathrm{mw}}^{\widetilde{G}}(q), \tag{187}$$

whose explicit expression for $\widetilde{G}_k$ was given in the previous section. The $u$ parameters are subject to large gauge transformations, $u_a \sim u_a + 1$, which amount to quotienting $\mathfrak{h}_{\mathbb{C}}$ by the magnetic weight lattice:

$$u \sim u + \mathfrak{m}, \qquad \forall \mathfrak{m} \in \Lambda_{\mathrm{mw}}^{\widetilde{G}}, \tag{188}$$

thus obtaining a cylinder for each Cartan generator. We further need to quotient by the Weyl group action on $u$. The Bethe vacua are then specific solutions $\hat{u}$ on this classical Coulomb branch $(\mathbb{C}^*)^{\mathrm{rank}(\mathfrak{g})}/W_{\mathfrak{g}}$. We note the important identities [126]:

$$\mathcal{G}_{q,p}(u + \mathfrak{m})_{\mathfrak{n}+p\mathfrak{m}} = \mathcal{G}_{q,p}(u)_{\mathfrak{n}}, \qquad \mathcal{G}_{q,p}(u)_{\mathfrak{n}+q\mathfrak{m}} = \Pi(u)^{\mathfrak{m}} \mathcal{G}_{q,p}(u)_{\mathfrak{n}}, \tag{189}$$

which are consequences of gauge invariance and of the $\mathbb{Z}_q$ orbifold structure at the $(q, p)$ special fiber, respectively. Recall that the off-shell gauge flux operator (57) trivialises on-shell (that is, for $u = \hat{u}$). In the following, we explain how this structure generalises to $\mathcal{N} = 2$ CS theories $G_K$ for $G$ not simply-connected, under the assumption that $G_k$ is a bosonic CS theory as explained above.

The supersymmetric partition function on $\mathcal{M}$ for the $G_K$ theory can be obtained by gauging the non-anomalous one-form symmetry $\Gamma_{3\mathrm{d}}^{(1)} \cong \Gamma$ of the $\widetilde{G}_K$ theory, summing over all background gauge fields:

$$Z_{\mathcal{M}}[G_K] = |\Gamma|^{-m(\mathcal{M})} \sum_{B \in H^2(\mathcal{M}, \Gamma)} Z_{\mathcal{M}}[\widetilde{G}_K](B), \tag{190}$$

---

[40]In order to find fixed points under a $\mathbb{Z}_d$ subgroup, we require that $k$ is divisible by $d$, since are no fixed points otherwise [95]. Using the action (176) of the 0-form symmetry $[\lambda] \to \gamma \cdot [\lambda]$ with $\gamma = n\gamma_0$, we determine with $n = \frac{N}{d}$ that $\lambda^a = \lambda^b$ if $a = b \mod \frac{N}{d}$, and thus all fixed points are of the form (186). These are $d$ repeated tuples of length $\frac{N}{d}$, with the last entry of the last tuple missing. The second constraint comes from the component $\lambda^{\frac{N}{d}} = k - |\lambda|$. Using $k \geq |\lambda|$, this is equivalent to $\lambda^1 + \lambda^2 + \cdots + \lambda^{\frac{N}{d}-1} \leq \frac{k}{d}$, which are thus in one-to-one correspondence with the integrable representations of $\mathrm{SU}(\frac{N}{d})_{\frac{k}{d}}$.

with $m(\mathcal{M})$ some constant that we leave implicit. Equivalently, we sum over all possible insertions of abelian anyons $a_\gamma \cong \mathcal{U}^\gamma$ along one-cycles of $\mathcal{M}$:

$$Z_{\mathcal{M}}[G_K] = |\Gamma|^{-m(\mathcal{M})} \sum_{\gamma \in H_1(\mathcal{M}, \Gamma)} \langle \mathcal{U}^\gamma \rangle_{\mathcal{M}}^{\widetilde{G}_K}. \tag{191}$$

In the following, we compute this quantity using the $A$-model perspective, in which case we will not need to determine the normalisation factor $m(\mathcal{M})$ to obtain the complete answer.

### 3.2.1 Orbifold fluxes and the homology of the Seifert fibration

The first step is to expound on the structure of the homology group $H_1(\mathcal{M}, \Gamma)$, and on its relation to the structure of the orbifold base $\Sigma_{g,\mathrm{N}}$ on which the $A$-model lives. Consider the Seifert fibration

$$S^1 \longrightarrow \mathcal{M} \xrightarrow{\pi} \Sigma_{g,\mathrm{N}}. \tag{192}$$

At the location of any $(q, p)$ exceptional fibre on the orbifold base, we can have an orbifold flux. Consider a single $U(1)$ factor for simplicity, with a gauge field $A$. The general orbifold flux on $\Sigma_{g,\mathrm{N}}$ can be localised at a generic point $z_0$ and at the orbifold points $z_i$:

$$\frac{1}{2\pi} dA = \mathfrak{n}_0 \delta^2(z - z_0) + \sum_{i=1}^{\mathrm{N}} \frac{\mathfrak{n}_i}{q_i} \delta^2(z - z_i). \tag{193}$$

More invariantly, we have an orbifold line bundle $L$ with first Chern class

$$c_1(L) = \frac{1}{2\pi} \int_{\Sigma_{g,\mathrm{N}}} dA = \mathfrak{n}_0 + \sum_i \frac{\mathfrak{n}_i}{q_i}. \tag{194}$$

Topologically, any such $L$ is determined by the integers $\mathfrak{n}_0$ and $\mathfrak{n}_i$ modulo some relations, which we write as:

$$L \cong [\mathfrak{n}_0; g; (q_i, \mathfrak{m}_i)] \cong [\mathfrak{n}_0 - \sum_{i=1}^{\mathrm{N}} \mathfrak{n}_i; g; (q_i, \mathfrak{m}_i + \mathfrak{n}_i q_i)], \tag{195}$$

keeping track of the genus $g$ of the base, for $\mathfrak{m}_i \in \mathbb{Z}$. Here the second equivalence tells us that the orbifold flux at the orbifold point $z_i$ is $\mathbb{Z}_{q_i}$-valued – that is, we can always convert $q_i$ units of orbifold flux into one unit of ordinary flux. (This is anticipated in the second relation of (189).) The orbifold Picard group consists of all the topological classes of orbifold line bundles. It reads:

$$\mathrm{Pic}(\Sigma_{g,\mathrm{N}}) \cong \left\{ L_0 \cong [1; g; (q_i, 0)], \ L_j \cong [0; g; (q_i, \delta_{ij})] \ \middle| \ (L_j)^{q_j} = L_0, j = 1, \cdots, \mathrm{N} \right\}. \tag{196}$$

It is not freely generated, in general. Our Seifert manifold (83) is essentially the total space of some particular orbifold line bundle called the *defining line bundle*:[41]

$$\mathcal{M} \cong \mathcal{L}_0 \cong [\mathrm{d}; g; (q_i, p_i)]. \tag{197}$$

The so-called 3d Picard group $\widetilde{\mathrm{Pic}}(\mathcal{M})$ consists of all the topological classes of 3d line bundles that are obtained as pull-backs of orbifold line bundles, where $\mathcal{L}_0$ pulls back to the trivial line bundle [126]:

$$\widetilde{\mathrm{Pic}}(\mathcal{M}) \cong \mathrm{Pic}(\Sigma_{g,\mathrm{N}})/(\mathcal{L}_0). \tag{198}$$

For our purposes, the most useful presentation of this group is in terms of generators of the first homology of the trivial fibration $\Sigma_{g,\mathrm{N}} \times S_A^1$. Let $[\omega_A]$ denote the generator of $H_1(S_A^1)$, and

---

[41]More precisely, $\mathcal{M} \cong [\mathcal{L}_0]$ the circle fibration associated to the defining line bundle.

let $[\omega_i]$ correspond to a one-cycle $\omega_i$ on the Riemann surface that circles once around the orbifold point $z_i \in \Sigma_{g,N}$. We then have:

$$\widetilde{\text{Pic}}(\mathcal{M}) \cong \left\{ [\omega_A], [\omega_i] \,\middle|\, q_i[\omega_i] + p_i[\omega_A] = 0, \,\forall i, \, \sum_{i=1}^{N} [\omega_i] = d[\omega_A] \right\}. \qquad (199)$$

The generators of (196) pull back to ordinary line bundles on $\mathcal{M}$ as follows:

$$\pi^*(L_0) \cong -[\omega_A], \qquad \pi^*(L_j) \cong -s_j[\omega_A] + t_j[\omega_j]. \qquad (200)$$

In the present work, the abelian group (199) is important mostly because it gives us the non-trivial part of the first homology (and second cohomology) of the Seifert manifold:

$$H_1(\mathcal{M}, \mathbb{Z}) \cong H^2(\mathcal{M}, \mathbb{Z}) \cong \widetilde{\text{Pic}}(\mathcal{M}) \oplus \mathbb{Z}^{2g}, \qquad (201)$$

where the $\mathbb{Z}^{2g}$ factor comes from the ordinary $A$- and $B$-cycles of the genus-$g$ base. By the universal coefficient theorem, one also finds that, for $\Gamma$-valued one-cycles:

$$H_1(\mathcal{M}, \Gamma) \cong \widetilde{\text{Pic}}(\mathcal{M}, \Gamma) \oplus \Gamma^{2g}, \qquad (202)$$

where we defined the tensor product:

$$\widetilde{\text{Pic}}(\mathcal{M}, \Gamma) \equiv \widetilde{\text{Pic}}(\mathcal{M}) \otimes \Gamma. \qquad (203)$$

The gauging in (191) involves summing over the elements of the finite abelian group (202).

### 3.2.2 Topological lines in the 2d perspective

From the perspective of the $A$-model on the 2d orbifold $\Sigma_{g,N}$, the three-dimensional one-form symmetry appears as two-dimensional one-form and zero-form symmetries:

$$\Gamma_{3d}^{(1)} \quad \longrightarrow \quad \Gamma^{(1)} \oplus \Gamma^{(0)}, \qquad (204)$$

and its 't Hooft anomaly appears as a mixed anomaly between $\Gamma^{(1)}$ and $\Gamma^{(0)}$ [95]. Indeed, $\Gamma^{(1)}$ is generated by point-like topological operators corresponding to the 3d lines $\mathcal{U}^\gamma(S_A^1)$ wrapping the circle direction, while the 2d zero-form symmetry $\Gamma^{(0)}$ is generated by topological lines $\mathcal{U}^\gamma(\mathcal{C})$ wrapping homology 1-cycles

$$[\mathcal{C}] \in H_1(\Sigma_{g,N}, \mathbb{Z}) \cong \begin{cases} \mathbb{Z}^{2g}, & \text{if } N = 0, \\ \mathbb{Z}^{2g+N-1}, & \text{if } N > 0. \end{cases} \qquad (205)$$

In addition to the ordinary $A$- and $B$-cycles on $\Sigma_g$, we have the generators $[\omega_i]$ (for $N > 0$), as defined above, subject to the relation $\sum_{i=1}^{N}[\omega_i] = 1$. We also introduce a 'fake' orbifold point $(q_0, p_0) = (1, d)$ to carry the degree of the fibration as shown in (84), effectively replacing $N$ with $N+1$ in (205).

**The $\Gamma^{(1)}$ symmetry and its gauging.** The one-form symmetry in 2d is generated by:

$$\mathcal{U}^\gamma(S_A^1) \equiv \Pi^\gamma. \qquad (206)$$

It is a local operator diagonalised by the Bethe vacua,

$$\Pi^\gamma|\hat{u}\rangle = \Pi(\hat{u})^\gamma|\hat{u}\rangle, \qquad (207)$$

which can be written off-shell as [95]:

$$\Pi(u)^\gamma \equiv \exp\left(2\pi i \gamma \frac{\partial \mathcal{W}}{\partial u}\right) = e^{2\pi i K(\gamma, u)}, \qquad \gamma \in \Lambda_{\text{mw}}^{\widetilde{G}/\Gamma}/\Lambda_{\text{mw}}^{\widetilde{G}} \cong \Gamma, \tag{208}$$

up to a phase ambiguity discussed in [95]. Here we view $\gamma$ as an element of the magnetic flux lattice for $G = \widetilde{G}/\Gamma$, which is finer than the magnetic flux lattice for $\widetilde{G}$. While the ordinary gauge flux operator $\Pi^{\mathfrak{m}}$, $\mathfrak{m} \in \Lambda_{\text{mw}}^{\widetilde{G}}$, corresponds to the insertion of some $\widetilde{G}$ magnetic flux on $\Sigma$ (and is then trivial on-shell in the $\widetilde{G}_K$ theory), the insertion of the refined flux operator $\Pi^\gamma$ corresponds to the insertion of a $G$ bundle on $\Sigma$ which does not lift to a $\widetilde{G}$ bundle, and it is thus a non-trivial observable of the $\widetilde{G}_K$ theory.

Gauging $\Gamma^{(1)}$ amounts to summing over all possible insertions of the topological operator:

$$\langle \mathcal{L} \rangle_{\mathcal{M}} = \frac{1}{|\Gamma|} \sum_{\gamma \in \Gamma^{(1)}} \langle \mathcal{L} \Pi^\gamma \rangle_{\mathcal{M}}, \tag{209}$$

which simply restricts the sum over Bethe vacua to those that satisfy $\Pi(\hat{u})^\gamma = 1$. That is, we restrict ourselves to the $\vartheta = 1$ charge sector, in accordance with 'Step 1' of the anyon condensation process of subsection 3.1.2.

**The $\Gamma^{(0)}$ symmetry and its gauging.**    Next, we consider the insertion of arbitrary topological lines for $\Gamma^{(0)}$ on the orbifold. The insertion of lines on a smooth $\Sigma_g$ was discussed in detail in [95], and the relevant aspects will be reviewed below. By setting $g = 0$ for now, we can focus on the topological lines wrapping the orbifold points. Denoting by $\omega$ the small one-cycle wrapping the orbifold point at the base of a $(q, p)$ fiber, the line $\mathcal{U}^\gamma(\omega)$ acts on the ($\delta$-twisted sector) Bethe vacua as:

$$\mathcal{U}^\gamma(\omega)[\mathcal{G}_{q,p}]|\hat{u}_\mu; \delta\rangle = \mathcal{G}_{q,p}(\hat{u}_{\gamma(\mu)})|\hat{u}_\mu; \delta\rangle. \tag{210}$$

This is best explained in pictures:

$$\mathcal{U}^\gamma(\omega)[\mathcal{G}_{q,p}] = \quad \left( \begin{array}{c} \end{array} \right) \quad = \quad \left( \begin{array}{c} \end{array} \right), \tag{211}$$

where the crossed disk denotes the $\mathcal{G}_{q,p}$ operator. That is, using topological invariance and the trivial fusion $a_{-\gamma}a_\gamma = 1$, we obtain the Seifert fibering operator surrounded by a topological line $\gamma$ simply as the composition:

$$\mathcal{U}^\gamma(\omega)[\mathcal{G}_{q,p}] = \mathcal{U}^{-\gamma}(\mathcal{B})\,\mathcal{G}_{q,p}\,\mathcal{U}^\gamma(\mathcal{B}), \tag{212}$$

from which (210) directly follows. This can be written as an off-shell twisted chiral-ring operator as:

$$\mathcal{U}^\gamma(\omega)[\mathcal{G}_{q,p}(u)] = \mathcal{G}_{q,p}(u + \gamma). \tag{213}$$

The gauging of $\Gamma^{(0)}$ corresponds to summing over $\gamma \in \Gamma$, giving rise to a new Seifert fibering operator which we will discuss momentarily.

The consideration of a Riemann surface with $g > 0$, and the insertion of topological lines along its $2g$ $A$- and $B$-cycles, does not affect this discussion, since the Seifert fibering operators are local operators in 2d. As explained in [95], the insertion of topological lines

$$\mathcal{U}^\gamma = \prod_{l=1}^{2g} \mathcal{U}^{\gamma_l}(\mathcal{C}_i), \qquad \text{with} \quad \gamma \equiv \sum_{l=1}^{2g} \gamma_l[\mathcal{C}_l] \in \Gamma^{2g}, \tag{214}$$

where $\{[\mathcal{C}_l]\}$ form an integral basis for $H_1(\Sigma_g, \mathbb{Z}) \cong \mathbb{Z}^{2g}$, simply restricts the sum over Bethe vacua to those fixed by all the elements $\gamma_l \in \Gamma$. For any local operator $\mathcal{O}$, we have

$$\langle \mathcal{O}\mathcal{U}^\gamma \rangle_{\Sigma_g} = \sum_{\hat{u} \in \mathcal{S}_{\mathrm{BE}}^{(\gamma)}} \mathcal{O}(\hat{u}) \mathcal{H}(\hat{u})^{g-1}\,. \tag{215}$$

Here, the set of Bethe vacua we sum over consists of those Bethe vacua that are left invariant by the smallest subgroup $\mathrm{H}_\gamma^{(0)} \subseteq \Gamma^{(0)}$ that contains all the $\gamma_l$'s:

$$\mathcal{S}_{\mathrm{BE}}^{(\gamma)} \equiv \{\hat{u} \in \mathcal{S}_{\mathrm{BE}} \mid \hat{u} + \gamma^{(0)} \sim \hat{u}\,, \ \forall \gamma^{(0)} \in \mathrm{H}_\gamma^{(0)}\}\,. \tag{216}$$

Summing over all the insertions (214) accounts for the 'Steps 2 and 3' of the anyon condensation process on $\Sigma_g \times S^1$.

### 3.2.3   The Seifert fibering operator for $G_K$

Let us take another look at the off-shell fibering operator for the $\widetilde{G}_K$ theory,

$$\mathcal{G}_{q,p}(u) = q^{-\frac{\mathrm{rank}(\mathfrak{g})}{2}} \sum_{\mathfrak{n} \in \Lambda_{\mathrm{mw}}^{\widetilde{G}}(q)} \mathcal{G}_{q,p}(u)_{\mathfrak{n}}\,, \tag{217}$$

with the explicit expression for the summands given in (104). With some work, one can establish the identity:

$$\mathcal{G}_{q,p}(u+\gamma)_{\mathfrak{n}+p\gamma} = \mathcal{G}_{q,p}(u)_{\mathfrak{n}}\,, \tag{218}$$

using our stated assumptions about $\Gamma$.[42] One can show that this is consistent with the non-trivial homology relation $q[\omega] + p[\omega_A] = 0$ appearing in (199). Note also that (218) naturally generalises the first identity in (189). Indeed, it is the statement that the Seifert fibering operator of the $\widetilde{G}$ theory is consistent with gauge invariance under large gauge transformations along the maximal torus of $G = \widetilde{G}/\Gamma$ and not simply along the one of $\widetilde{G}$:

$$\mathcal{G}_{q,p}(u+\mathfrak{m})_{\mathfrak{n}+p\mathfrak{m}} = \mathcal{G}_{q,p}(u)_{\mathfrak{n}}\,, \qquad \forall \mathfrak{m} \in \Lambda_{\mathrm{mw}}^{G} \supset \Lambda_{\mathrm{mw}}^{\widetilde{G}}\,. \tag{219}$$

For our purposes, we can focus on the on-shell fibering operator, which is what should appear in the $A$-model formula (49). The action of $\mathcal{G}_{q,p}$ on the $G_K$ states (164) is easily worked out using the fact that $\widetilde{G}_K$ Seifert fibering operator is diagonalised by the Bethe states (even in the presence of the lines defining the $\delta$-twisted sectors), similarly to the $T$-matrix (167),

$$\mathcal{G}_{q,p}|\hat{u};\delta\rangle = \mathcal{G}_{q,p}(\hat{u})|\hat{u};\delta\rangle\,, \qquad \forall \delta \in \Gamma_{\hat{\omega}}\,. \tag{220}$$

We then see that:

$$\begin{aligned}\langle \hat{\omega}';\chi'|\mathcal{G}_{q,p}|\hat{\omega};\chi\rangle &= \frac{\delta_{\hat{\omega}\hat{\omega}'}}{|\Gamma|} \sum_{\delta' \in \Gamma_{\hat{\omega}}} \sum_{\delta \in \Gamma_{\hat{\omega}}} \chi'(\delta')^* \chi(\delta) \sum_{\hat{u}' \in \hat{\omega}} \sum_{\hat{u} \in \hat{\omega}} \langle \hat{u}';\delta'|\mathcal{G}_{q,p}|\hat{u};\delta\rangle \\ &= \frac{\delta_{\hat{\omega}\hat{\omega}'}\delta_{\chi\chi'}}{|\hat{\omega}|} \sum_{\hat{u} \in \hat{\omega}} \mathcal{G}_{q,p}(\hat{u})\,.\end{aligned} \tag{221}$$

Thus, the eigenvalues of the Seifert fibering operator $\mathcal{G}_{q,p}$ on the $G_K$ states are simply obtained by averaging the eigenvalues of the $\widetilde{G}_K$ operator over the $\Gamma^{(0)}$ orbits of vanishing $\Gamma^{(1)}$ charge:

$$\mathcal{G}_{q,p}|\hat{\omega};\chi\rangle = \mathcal{G}_{q,p}^{G}(\hat{\omega})|\hat{\omega};\chi\rangle\,, \qquad \text{with} \quad \mathcal{G}_{q,p}^{G}(\hat{\omega}) \equiv \frac{1}{|\hat{\omega}|} \sum_{\hat{u} \in \hat{\omega}} \mathcal{G}_{q,p}(\hat{u})\,, \tag{222}$$

---

[42]More precisely this holds on-shell in the $\vartheta = 1$ sector, which is what is needed for our purposes here.

for $\hat{\omega} = \hat{\omega}(\hat{u})$. The $G_K$ Seifert fibering operator can also be elegantly obtained by summing over the insertions $\mathcal{U}^\gamma[\omega]$ around the orbifold point. The action (213) give us:

$$\mathcal{G}_{q,p}^G(\hat{\omega}) = \frac{1}{|\Gamma|} \sum_{\gamma \in \Gamma} \mathcal{G}_{q,p}(\hat{u} + \gamma), \tag{223}$$

in perfect agreement with (222). Using the identity (218), we can further massage this into:

$$\mathcal{G}_{q,p}^G(\hat{\omega}) = \delta_{\vartheta,1} \sum_{\mathfrak{n} \in \Lambda_{\mathrm{mw}}^G(q)} \mathcal{G}_{q,p}(\hat{u})_{\mathfrak{n}}, \tag{224}$$

for any $\hat{u} \in \hat{\omega}$ and where the orbifold-flux sum is now over the mod-$q$ reduction of the magnetic weight lattice for $G = \widetilde{G}/\Gamma$. Moreover, while we can consider the sum over topological lines (223) for orbits $\hat{\omega}$ with generic one-form charge $\vartheta \in \hat{\Gamma}$, the result is non-vanishing if and only if $\vartheta = 1$.[43] These expressions for $\mathcal{G}_{q,p}^G(\hat{\omega})$ were first obtained by Brian Willett in [92] but our derivation clarifies a few subtle points, especially regarding the assumptions we needed to make about the Bethe vacua of the $G_K$ theory being bosonic.

**Comparison to the TQFT formulas.** For explicit computations, we most readily use the formula (222), namely

$$\mathcal{G}_{q,p}^G(\hat{\omega}) = \frac{1}{|\hat{\omega}|} \sum_{\mu \in \hat{\omega}} \mathcal{G}_{q,p}^{\widetilde{G}}(\hat{u}_\mu), \tag{225}$$

for the Seifert fibering operator of the $G_K = (\widetilde{G}/\Gamma)_K$ theory, with $\mathcal{G}_{q,p}(\hat{u}) \equiv \mathcal{G}_{q,p}^{\widetilde{G}}(\hat{u})$. The supersymmetric answer can be compared to the 3d TQFT formula, and the two should be related by the exact same counterterm as in (106):

$$\mathcal{G}_{q,p}^G(\hat{\omega}) = e^{-\frac{\pi i t}{12q} c(G_k)} \frac{\mathcal{U}_{(\hat{\omega},\chi),0}^{(q,p)}}{S_{0,(\hat{\omega},\chi)}}, \tag{226}$$

noting that $c(G_k) = c(\widetilde{G}_k) = c(\hat{\mathfrak{g}}_k)$, where the $\mathcal{U}$- and $S$-matrix elements appearing on the right-hand-side are those built out of the $S$ and $T$ matrices described in section 3.1.3. This relation is equivalent to the following non-trivial equality involving the modular matrices of the $G_K$ and $\widetilde{G}_K$ theories, respectively:

$$\frac{\mathcal{U}_{(\hat{\omega},\chi),0}^{(q,p)}}{S_{0,(\hat{\omega},\chi)}} = \frac{1}{|\hat{\omega}|} \sum_{\mu \in \hat{\omega}} \frac{\mathcal{U}_{\mu 0}^{(q,p)}}{S_{0\mu}}. \tag{227}$$

In the special case $(q,p) = (1,1)$, this is equivalent to the statement that the ordinary fibering operator satisfies $\mathcal{F}(\hat{u}_\mu) = \mathcal{F}(\hat{u}_\nu)$ if $\mu$ and $\nu$ are in the same orbit, while for generic $(q,p)$ this identity is quite more involved since it involves an 'averaging' over orbits. While we leave a mathematical proof of (227) to the interested reader, we verified this numerically in a number of non-trivial examples.[44]

### 3.2.4 Supersymmetric partition functions for $G_K$

The supersymmetric partition function of the $G_K$ theory on the Seifert three-manifold $\mathcal{M}$ takes the exact same form as in (108), except that we now trace over the Bethe states (164) of the $G_K$ theory:

$$Z_{\mathcal{M}}^{\mathrm{SUSY}}[G_K] = \left\langle \mathcal{G}_{\mathcal{M}}^G \right\rangle_{\Sigma_g} = \sum_{(\hat{\omega},\chi)} \mathcal{H}^G(\hat{\omega})^{g-1} \prod_{i=0}^{\mathrm{N}} \mathcal{G}_{q_i,p_i}^G(\hat{\omega}). \tag{228}$$

---

[43]In a sense, the non-trivial fibration combines 'Step 1' and 'Step 2' of the anyon condensation process.

[44]That is, for all the examples discussed in section 3.2.5 below.

Here the Seifert fibering operators are as defined in (225), and the eigenvalues of the handle-gluing operator are given by:

$$\mathcal{H}|\hat{\omega}; \chi\rangle = \mathcal{H}^G(\hat{\omega})|\hat{\omega}; \chi\rangle, \qquad \mathcal{H}^G(\hat{\omega}) = \frac{1}{|\hat{\omega}|^2}\mathcal{H}(\hat{u}_\mu), \tag{229}$$

as discussed in detail in [95]. Note that (229) also directly follows from (173) together with the general $A$-model relations $\mathcal{H}(\hat{u}_\mu) = S_{0\mu}^{-2}$ and $\mathcal{H}^G(\hat{\omega}) = S_{0,(\hat{\omega},\chi)}^{-2}$. The relation between the supersymmetric partition function (228) and the TQFT answer discussed in section 3.1.3 should remain exactly as in (110), with the same theory-independent counterterm multiplied by the WZW model central charge. Given our explicit proof of the relation (110) for the $\widetilde{G}_k$ theory in section 2.4, the proof of the same relation for the $G_k$ theory is equivalent to proving (227), which is a non-trivial identity formulated entirely in the 3d TQFT language — again, we leave the completion of this important missing step to the interested reader.

While (228) is our final and main result, it is interesting to further confirm how it arises as a sum over topological lines in the $A$-model. Inserting all possible topological symmetry operators for $\Gamma^{(1)} \times \Gamma^{(1)}$ on $\Sigma_{g,\mathbb{N}}$ gives us:

$$Z_{\mathcal{M}}^{\text{SUSY}}[G_K] = \frac{1}{|\Gamma|^{2g+\mathbb{N}+1}} \sum_{\delta \in \Gamma} \sum_{(\gamma,\zeta) \in \Gamma^{2g+\mathbb{N}+1}} \left\langle \Pi^\delta \mathcal{U}^{(\gamma,\zeta)} \right\rangle_{\mathcal{M}}^{\widetilde{G}_K}, \tag{230}$$

where $\Pi^\delta$ denote the $\Gamma^{(1)}$ symmetry operator and we use the shorthand:

$$\mathcal{U}^{(\gamma,\zeta)} \equiv \prod_{l=1}^{2g} \mathcal{U}^{\gamma_l}(\mathcal{C}_l)\prod_{i=0}^{\mathbb{N}} \mathcal{U}^{\zeta_i}(\omega_i), \quad \text{with} \quad (\gamma,\zeta) \equiv \sum_{l=1}^{2g}\gamma_l[\mathcal{C}_l] + \sum_{i=0}^{\mathbb{N}}\zeta_i[\omega_i] \in \Gamma^{2g+\mathbb{N}+1}, \tag{231}$$

generalising (214) to a Riemann surface with $\mathbb{N}+1$ marked points (including the point $z = z_0$ supporting the $(q_0, p_0) = (1, d)$ operator). Performing this sum as discussed above (and in more detail in [95] in the case of the sum over $\Gamma^{2g}$) gives us exactly the formula (228).

**Partition functions for the $(\text{SU}(N)/\mathbb{Z}_r)_K$ theories.** For $\widetilde{G} = \text{SU}(N)$, we can further massage the formula (230), similarly to the discussion in [95]. Since the insertion of $\mathcal{U}^\gamma$ implements a projection onto the space of Bethe vacua fixed simultaneously by all $\gamma$, the sum (230) can be simplified drastically by collecting the expectation values for all elements $\gamma$ generating a $\mathbb{Z}_d$ subgroup. Gauging the full $\mathbb{Z}_N$ symmetry, we may then express the $\text{PSU}(N)_K$ partition function on arbitrary Seifert manifolds $\mathcal{M}$ as:

$$Z_{\mathcal{M}}[\text{PSU}(N)_K] = \frac{1}{N^{2g-1}} \sum_{d|N} J_{2g}(d) \sum_{\hat{u} \in \mathcal{S}_{\text{BE}}^{\mathbb{Z}_d, \vartheta=1}} \mathcal{H}(\hat{u})^{g-1} \prod_{i=0}^{\mathbb{N}} \mathcal{G}_{q,p}^{\text{PSU}(N)}(\hat{\omega}), \tag{232}$$

where $\hat{\omega}$ is the $\mathbb{Z}_N$ orbit containing $\hat{u}$. Here, $\mathcal{S}_{\text{BE}}^{\mathbb{Z}_d, \vartheta=1}$ is the set of all Bethe vacua $\hat{u}$ in the $\mathbb{Z}_N$ charge sector $\vartheta = 1$ which are fixed under a $\mathbb{Z}_d$ subgroup. This can be easily generalised to all subgroups $\text{SU}(N)/\mathbb{Z}_r$, for any suitable[45] divisor $r$ of $N$:

$$Z_{\mathcal{M}}[(\text{SU}(N)/\mathbb{Z}_r)_K] = \frac{1}{r^{2g-1}} \sum_{d|\gcd(r,K)} J_{2g}(d) \sum_{\hat{u} \in \mathcal{S}_{\text{BE}}^{\mathbb{Z}_d, (\vartheta=1)^{(\mathbb{Z}_r)}}} \mathcal{H}(\hat{u})^{g-1} \prod_{i=0}^{\mathbb{N}} \mathcal{G}_{q,p}^{\text{PSU}(N)}(\hat{\omega}), \tag{233}$$

where we sum over $\mathbb{Z}_d$ fixed points with $\mathbb{Z}_r$ charges $\vartheta = 1$ instead. The $q$-reduced $\text{PSU}(N)$ fluxes $\widetilde{\mathfrak{n}} \in \Lambda_{\text{mw}}^{\text{PSU}(N)}(q)$ can be obtained form the $\text{SU}(N)$ fluxes $\mathfrak{n} \in \Lambda_{\text{mw}}^{\text{SU}(N)}(q)$ by $\widetilde{\mathfrak{n}} = A^{-1}\mathfrak{n}$,

---

[45]See section 3.1.4.

with $A$ the SU($N$) Cartan matrix (see Appendix B.1). This final result thus gives an efficient method to compute the partition function for all $(SU(N)/\mathbb{Z}_r)_K$ theories based on the fixed points under the $\mathbb{Z}_d$ subgroups. It generalises the Witten index (183) and the topologically twisted indices [95, Equation (3.108)] on $\Sigma_g \times S^1$ to arbitrary Seifert manifolds $\mathcal{M}$.

### 3.2.5 Examples and consistency checks

In this final section, we provide some evidence for the proposed formalism in the form of various consistency checks, as well as explicit results of partition functions on specific Seifert geometries that we demonstrate to match across distinct calculations.

$S^2 \times S^1$ **partition function.** When $\mathcal{M}$ contains an $S^1$ factor, we can interpret the partition function $Z_{\mathcal{M}}[G] = \eta_{00}$ as an index, and it should consequently be an integer for any 3d $\mathcal{N} = 2$ theory. For any 3d TQFT, the Hilbert space $\mathscr{H}_{S^2}$ is one-dimensional and therefore

$$Z_{S^2 \times S^1}[G] = 1 \,, \tag{234}$$

for all compact simple gauge groups $G$ [130, 164]. In particular, gauging a subgroup $\Gamma$ of the 3d centre symmetry leaves the partition function invariant. Since this particular three-manifold has nontrivial first homology, gauging is clearly a non-trivial operation. In the $A$-model language, (234) is equivalent to

$$\sum_{\hat{u} \in \mathcal{S}_{\mathrm{BE}}} \mathcal{H}(\hat{u})^{-1} = |\Gamma| \sum_{\hat{u} \in \mathcal{S}_{\mathrm{BE}}^{\vartheta=1}} \mathcal{H}(\hat{u})^{-1} \,, \tag{235}$$

which is consistent with the normalisation factor (230). Here $\mathcal{S}_{\mathrm{BE}}^{\vartheta=1}$ denotes the set of Bethe vacua in the $\vartheta = 1$ sector.

**Note on $\theta$-angles for $\Gamma^{(1)}$.** In [95], the gauging of the 2d one-form and zero-form symmetries were considered separately, and the insertion of a background gauge field for $\Gamma^{(1)}$ acted as a '$\theta$-angle' keeping track of the $\vartheta$-sectors (also called 'universes' [193]) – that is, one could consider the topologically twisted index of some $\vartheta$-sector for $\vartheta \neq 1$. Once we introduce exceptional fibers ($N > 0$), it is apparent from (224) that the naive analogue of the $\Gamma^{(0)}$ gauging on the orbifold already projects us onto the sector $\mathcal{S}_{\mathrm{BE}}^{\vartheta=1}$. This is immaterial provided that we gauge the $\Gamma^{(1)}$ symmetry simultaneously with vanishing $\theta$-angle, as in the above discussion, which projects onto the same universe. Due to the non-trivial geometry of the fibration, the effects of the one-form and zero-form gauging are not clearly separated, and it may thus not be meaningful to consider them separately. (Indeed, one can only introduce background gauge fields for $\Gamma_{3d}^{(1)}$, depending on the topology of $\mathcal{M}$.) Nonetheless, if we insisted on turning on $\theta \neq 0$ on a generic Seifert manifold, we would then find that the partition function vanishes:

$$Z_{\mathcal{M}}[(\widetilde{G}/\Gamma)_K^{\theta}] = \delta_{\vartheta,1} Z_{\mathcal{M}}[(\widetilde{G}/\Gamma)_K^{\theta=0}] \,. \tag{236}$$

Incidentally, even on $\mathcal{M} = T^3$ not every $\theta$-angle is 'allowed'. We have demonstrated this in [95, Equation (3.100)] for the case of pure $SU(N)_K$ CS theory. In general, the $\theta$-angle can furthermore interact nontrivially with the spin structures on $\mathcal{M}$.[46] We will discuss this in detail in future work [165].

---

[46]This was anticipated in [95], where it was found that the $T^3$ partition function for the pure $(SU(N)/\mathbb{Z}_r)_K$ Chern–Simons theory has a more intricate dependence on the $\theta$-angle whenever the non-supersymmetric $(SU(N)/\mathbb{Z}_r)_K$ theory is a spin-TQFT.

**Trivial homology.** In order to gauge the 3d one-form symmetry, we sum over all insertions of topological lines (230). Note that some or all of these lines might be trivial in homology on $\mathcal{M}$. We claim that the $A$-model calculation on the orbifold base $\hat{\Sigma}$ of the fibration encodes the homology group as relations among the fibering operators that 'construct' it—this is a claim we provide evidence for in the following.

Since the background gauge fields for the one-form symmetry $\Gamma$ of a 3d $\mathcal{N} = 2$ theory are valued in $H^2(\mathcal{M}, \Gamma)$, the 3d gauging (190) of $\Gamma$ is trivial if

$$H^2(\mathcal{M}, \Gamma) \cong H_1(\mathcal{M}, \Gamma) = 0. \tag{237}$$

As already alluded to in section 3.1.3, however in such cases the partition function of the $\widetilde{G}/\Gamma$ theory differs from that of the $\widetilde{G}$ theory by a simple overall factor of $|\Gamma|$:[47]

$$H_1(\mathcal{M}, \Gamma) = 0 \quad \Longrightarrow \quad Z_{\mathcal{M}}[\widetilde{G}/\Gamma] = |\Gamma| Z_{\mathcal{M}}[\widetilde{G}]. \tag{238}$$

Meanwhile, from the 3d $A$-model perspective, we still have distinct two-dimensional symmetries $\Gamma^{(1)}$ and $\Gamma^{(0)}$, and the discrete gauging in 2d is a non-trivial operation in general. Let us therefore study how the simple relation (238) arises in the $A$-model.

The condition (237) can be realised in two rather different ways. Either the integral homology $H_1(\mathcal{M}, \mathbb{Z})$ is trivial (that is, $\mathcal{M}$ is an integral homology three-sphere), and the gauging is trivial for any 3d centre symmetry. Or, more generally, the integral cohomology is nontrivial, *e.g.* $H_1(\mathcal{M}, \mathbb{Z}) \cong \mathbb{Z}_d$ for some integer $d$, while $\Gamma \cong \mathbb{Z}_N$, such that $H_1(\mathcal{M}, \Gamma) \cong \mathbb{Z}_{\gcd(d,N)}$.[48] Hence the discrete gauging is trivial only if $d$ and $N$ are coprime. While (237) can only occur for $g = 0$, two classes of examples we study in detail below are the cases $\mathtt{N} = 0$ and $\mathtt{d} = 0$. The former are the degree-d principal circle bundles $\mathcal{M}_{0,\mathrm{d}}$, while the latter include an infinite family of homology spheres, lens spaces, etc.

**The principal circle bundles.** Let us first consider the geometries $\mathcal{M}_{0,\mathrm{d}} \cong S^3/\mathbb{Z}_{\mathrm{d}}$, which have $H_1(\mathcal{M}, \mathbb{Z}) \cong \mathbb{Z}_{\mathrm{d}}$. These include in particular the three-sphere $S^3 = \mathcal{M}_{0,1}$. For a cyclic one-form symmetry $\Gamma \cong \mathbb{Z}_N$, the first homology is $H_1(\mathcal{M}, \Gamma) \cong \mathbb{Z}_{\gcd(\mathrm{d},N)}$, and we are here interested in the case $\gcd(\mathrm{d}, N) = 1$.

Focusing as before on the bosonic cases (142), we have $\mathcal{F}(\hat{u} + \zeta)^{\mathrm{d}} = \Pi(\hat{u})^{-\mathrm{d}\zeta} \mathcal{F}(\hat{u})^{\mathrm{d}}$. In the $A$-model gauging (230), we then consider sums of the form[49]

$$\sum_{\zeta \in \Gamma} \mathcal{F}(\hat{u} + \zeta)^{\mathrm{d}} = |\Gamma| \mathcal{F}(\hat{u})^{\mathrm{d}} \mathbf{1}_{\mathcal{S}_{\mathrm{BE}}^{\vartheta=1}}(\hat{u}). \tag{239}$$

We stress that $\gcd(\mathrm{d}, N) = 1$ is necessary for the projection map to $\mathcal{S}_{\mathrm{BE}}^{\vartheta=1}$ to work out precisely. Using (239) and $\sum_{\hat{u} \in \hat{\omega}} \mathcal{F}(\hat{u})^{\mathrm{d}} = |\mathrm{Stab}(\hat{\omega})|^{-1} \sum_{\zeta \in \Gamma} \mathcal{F}(\hat{u} + \zeta)^{\mathrm{d}}$, we can express the $\widetilde{G}$ partition function as

$$Z_{\mathcal{M}_{0,\mathrm{d}}}[\widetilde{G}] = \sum_{\hat{\omega} \in \mathcal{S}_{\mathrm{BE}}^{\vartheta=1}/\Gamma^{(0)}} |\hat{\omega}| \mathcal{H}(\hat{\omega})^{-1} \mathcal{F}(\hat{\omega})^{\mathrm{d}}. \tag{240}$$

This agrees precisely with $Z_{\mathcal{M}_{0,\mathrm{d}}}[G]$ (228) for $\mathtt{N} = 0$, up to the factor $|\Gamma|$. We have thus shown that

$$Z_{\mathcal{M}_{0,\mathrm{d}}}[\widetilde{G}/\Gamma] = |\Gamma| Z_{\mathcal{M}_{0,\mathrm{d}}}[\widetilde{G}], \qquad \text{if } H_1(\mathcal{M}_{0,\mathrm{d}}, \Gamma) = 0, \tag{241}$$

for the case of $\Gamma \cong \mathbb{Z}_N$. This includes the three-sphere result (174) for any group $\Gamma \cong \mathbb{Z}_N$, and it is a consistency check on the normalisation factor in (230).

---

[47]This overall factor of $|\Gamma|$ may be understood as the contributions from flat $\Gamma$-bundles [164].

[48]In the following, for simplicity we consider the case where $\Gamma$ is cyclic. The general case where $\Gamma$ is a product of cyclic groups follows analogously.

[49]For all $\hat{u} \in \mathcal{S}_{\mathrm{BE}}^{\vartheta=1}$, we have $\Pi^{\zeta_0}(\hat{u}) = 1$ with $\zeta_0$ a generator of $\Gamma \cong \mathbb{Z}_N$, and thus the sum is constant. When $\Pi^{\zeta_0}(\hat{u}) \neq 1$, it is a nontrivial $N$-th root of unity—this is of course because $\Pi^{N\zeta_0} \equiv \mathbb{1}$ is the identity. If d does not share any divisors with $N$, then $\Pi^{\mathrm{d}\zeta_0}(\hat{u}) \neq 1$, but $\Pi^{N\mathrm{d}\zeta_0}(\hat{u}) = 1$, and the geometric series vanishes.

**The case d = 0.** As a second class of examples, consider the manifolds $\mathcal{M} \cong [0;0;(q_i,p_i)]$ with $g = d = 0$. These have trivial homology if

$$\sum_{i=1}^{N} \frac{p_i}{q_i} = \pm \frac{1}{\prod_{i=1}^{N} q_i} \,, \tag{242}$$

which in particular implies that $\gcd(q_i,q_j) = 1$ for all $i \neq j$.[50] Many interesting cases can furthermore be generated by suitably adjusting $\Gamma$ such that $H_1(\mathcal{M},\Gamma) = 0$ even when $H_1(\mathcal{M},\mathbb{Z})$ is nontrivial. We can write the $\widetilde{G}$ partition function as

$$Z_{\mathcal{M}}[\widetilde{G}] = \sum_{\hat{\omega} \in \mathcal{S}_{\text{BE}}/\Gamma^{(0)}} \mathcal{H}(\hat{\omega})^{-1} \sum_{\hat{u} \in \hat{\omega}} \prod_{i=1}^{N} \mathcal{G}_{q_i,p_i}(\hat{u}) \,. \tag{244}$$

In order to relate the partition functions for $\widetilde{G}$ and $G$, we postulate the 'orthogonality' relation:

$$\prod_{i=1}^{N} \sum_{\hat{u} \in \hat{\omega}} \mathcal{G}_{q_i,p_i}(\hat{u}) = |\hat{\omega}|^{N-1} \sum_{\hat{u} \in \hat{\omega}} \prod_{i=1}^{N} \mathcal{G}_{q_i,p_i}(\hat{u}) \,, \qquad \forall \hat{\omega} \in \mathcal{S}_{\text{BE}}/\Gamma^{(0)} \,. \tag{245}$$

Assuming this relation, (244) becomes:

$$Z_{\mathcal{M}}[\widetilde{G}] = \sum_{\hat{\omega} \in \mathcal{S}_{\text{BE}}/\Gamma^{(0)}} \mathcal{H}(\hat{\omega})^{-1} |\hat{\omega}| \prod_{i=1}^{N} \mathcal{G}^{G}_{q_i,p_i}(\hat{\omega}) \,. \tag{246}$$

Due to (224), the $G_K$ Seifert fibering operator $\mathcal{G}^{G}_{q,p}(\hat{\omega})$ vanishes if $\hat{\omega} \notin \mathcal{S}^{\vartheta=1}_{\text{BE}}/\Gamma^{(0)}$. As a consequence, only the sector $\vartheta = 1$ contributes to this partition function, which precisely gives us the gauged partition function $Z_{\mathcal{M}}[G]$ (228) (with d = 0), up to the expected factor of $|\Gamma|$. We have thus shown that

$$(245) \quad \implies \quad Z_{\mathcal{M}}[\widetilde{G}/\Gamma] = |\Gamma| Z_{\mathcal{M}}[\widetilde{G}] \,, \tag{247}$$

matching the expectation (238). Of course, this discussion hinges crucially on the very nontrivial identity (245), which we did not prove for $N > 1$. (Note that (245) is clearly true for $N = 1$.) We conjecture that $H_1(\mathcal{M},\Gamma) = 0$ always implies the relation (245). For $SU(N)_K$ Chern–Simons theory, we have checked this numerically in a number of cases.[51]

**Example: $SU(2)_K$ Chern–Simons theory.** Before giving some more explicit results for Seifert manifolds, let us make the formalism concrete on the simple example of $\mathcal{N} = 2$ supersymmetric $SU(2)_K$ CS theory, with $K$ even but $\frac{K}{2}$ odd (recall that here $k = K - 2$ is the bosonic level). We have $K - 1$ states are indexed by $\alpha = 0, 1, \cdots, K - 2$, and the $T$ and $S$-matrices read,

$$T_{\alpha\beta} = e^{2\pi i(h_\alpha - \frac{c}{24})} \delta_{\alpha\beta} \,,$$
$$S_{\alpha\beta} = \sqrt{\frac{2}{K}} \sin\left(\frac{\pi(\alpha+1)(\beta+1)}{K}\right), \tag{248}$$

---

[50]Particularly interesting examples are the Poincaré homology sphere $\mathcal{S}^3[E_8]$, and the manifold $\mathcal{S}^3[E_{10}]$ which has $SL(2,\mathbb{Z})$ Thurston geometry, where

$$\mathcal{S}^3[E_{m+3}] \cong [0 \,;\, 0 \,;\, (2,-1)\,,\, (3,1)\,,\, (m,1)]\,. \tag{243}$$

[51]In particular for the spherical manifolds $\mathcal{S}^3[E_{m+3}]$ ($m = 3, 4, 5$) (243), we checked (245) numerically up to $N \leq 7$ and $K \leq 14$, where the identity holds if and only if $H_1(\mathcal{S}^3[E_{m+3}],\mathbb{Z}_N) \cong \mathbb{Z}_{\gcd(6-m,N)}$ is trivial.

where

$$h_\alpha = \frac{\frac{\alpha}{2}\left(\frac{\alpha}{2}+1\right)}{K}, \qquad c = \frac{3(K-2)}{K}. \tag{249}$$

In order to write down the $S$ and $T$-matrices of the $PSU(2)_K$ theory, we need to determine the states in the gauged theory first. In order to index the states, we use the isospin $j$, with $\alpha = 2j$. For $\frac{K}{2}$ odd, the states in the $PSU(2)_K$ theory are then labelled by $(j,s)$ with $j = 0, \ldots, \frac{k}{4}$, where the twisted sector $s$ is trivial, unless $j = \frac{k}{4}$, where $s$ labels a $\mathbb{Z}_2$ stabiliser. Using the formalism of Section 3.1.4, in particular (185), we then find

$$S_{(j,s),(l,s')} = \begin{cases} 2S_{j,l}^{(0)}, & \text{if } j, l \neq \frac{k}{2}, \\ S_{j,l}^{(0)}, & \text{if } j, \text{ or } l = \frac{k}{2}, \text{ and } j \neq l, \\ \frac{1}{2}(S_{\frac{k}{2},\frac{k}{2}}^{(0)} + S_{\frac{k}{2},\frac{k}{2}}^{(1)}), & \text{if } j = l = \frac{k}{2}, \text{ and } s = s' \\ \frac{1}{2}(S_{\frac{k}{2},\frac{k}{2}}^{(0)} - S_{\frac{k}{2},\frac{k}{2}}^{(1)}), & \text{if } j = l = \frac{k}{2}, \text{ and } s \neq s'. \end{cases} \tag{250}$$

Here, $S^{(0)}$ denotes the original $SU(2)_k$ entries $S_{\alpha,\beta}$ with $\alpha = 2j$, $\beta = 2l$, with $j, l = 0, \cdots, \frac{k}{2}$. The state $j = \frac{k}{2}$ resolves into two states ($s = 0, 1$) with the $2 \times 2$ matrix as shown above, with $S_{\frac{k}{2},\frac{k}{2}}^{(1)} = i^{\frac{k}{4}}$. The $T$-matrix is trivially obtained from the one of the $SU(2)_K$ theory.

**Torus bundles.** With these, we can study another strong consistency check of our formalism which comes from the gauging on torus bundles over a circle. As described in section 2.4.2, the TQFT partition function should coincide with the trace of the matrix that represents the torus bundle monodromy on the torus Hilbert space,

$$Z_{\mathcal{M}^A}^{\text{TQFT}}[G] = \text{Tr}_{\mathcal{H}_{T2}}(\mathcal{A}^G), \tag{251}$$

extending (130) to the non-simply connected case. Together with the 3d TQFT calculation described in section 3.1.3, this gives us three separate calculations for the partition functions $Z_{\mathcal{M}^A}[G_k]$ on the torus bundles that admit a Seifert fibration. Let us check this explicitly in some simple examples.

For the $SU(2)_k$ theory, the $\mathbb{Z}_2$ gauging results in a bosonic $PSU(2)_k$ theory if $k$ is a multiple of 4. In those cases, we determine (compare with (135)):

$$Z_{\mathcal{M}^C}[PSU(2)_k] = \frac{k}{4} + 2\delta_{\frac{k}{4} \bmod 2, 0},$$

$$Z_{\mathcal{M}^S}[PSU(2)_k] = 2\delta_{k \bmod 16, 0} - i\delta_{k \bmod 16, 4} + i\delta_{k \bmod 16, 12},$$

$$Z_{\mathcal{M}^{T^{-1}S}}[PSU(2)_k] = 2\delta_{k \bmod 12, 0} + \sqrt{3}e^{-\frac{\pi i}{6}}\delta_{k \bmod 12, 4} + \delta_{k \bmod 12, 8}, \tag{252}$$

$$Z_{\mathcal{M}^{ST}}[PSU(2)_k] = 2\delta_{k \bmod 24, 0} + e^{\frac{2\pi i}{3}}\delta_{k \bmod 24, 4} + \delta_{k \bmod 24, 8}$$
$$+ \sqrt{3}e^{\frac{\pi i}{6}}\delta_{k \bmod 24, 16} - \delta_{k \bmod 24, 20}.$$

For larger rank, the calculations get more involved. For $k$ any multiple of 3, we find (compare with (136)):

$$Z_{\mathcal{M}^C}[PSU(3)_k] = \frac{1}{2}\left(5 + k + \delta_{k \bmod 6, 0}\right),$$

$$Z_{\mathcal{M}^S}[PSU(3)_k] = 3\delta_{k \bmod 12, 0} + 2\delta_{k \bmod 12, 3} + 2\delta_{k \bmod 12, 6} + (2 - i)\delta_{k \bmod 12, 9},$$

$$Z_{\mathcal{M}^{T^{-1}S}}[PSU(3)_k] = \frac{1}{3}e^{\frac{\pi i}{3}}k - 1 - 2\sqrt{3}i\left\lfloor\frac{k+3}{9}\right\rfloor + 4\delta_{k \bmod 9, 0}, \tag{253}$$

$$Z_{\mathcal{M}^{ST}}[PSU(3)_k] = 3\delta_{k \bmod 18, 0} + e^{-\frac{2\pi i}{3}}\delta_{k \bmod 18, 3} + \sqrt{3}i\delta_{k \bmod 18, 6},$$
$$+ (2 + e^{\frac{\pi i}{3}})\delta_{k \bmod 18, 9} - \sqrt{3}i\delta_{k \bmod 18, 12} + (\sqrt{3}i + e^{\frac{2\pi i}{3}})\delta_{k \bmod 18, 15}.$$

Moreover, we have checked the relation (110) numerically for various $(SU(N)/\mathbb{Z}_r)_k$ theories on numerous other geometries, including principal bundles, lens spaces, spherical manifolds, homology spheres, and more.

## Acknowledgments

We are grateful to Lea Bottini, Sakura Schäfer-Nameki, and Shu-Heng Shao for correspondence and discussions. CC also acknowledges crucial discussions with Heeyeon Kim, Victor Mikhaylov and Brian Willett on the subject of the 3d *A*-model around 2017–2018. CC is a Royal Society University Research Fellow.

**Funding information** EF is supported by the EPSRC grant "Local Mirror Symmetry and Five-dimensional Field Theory". The work of AK and of OK is supported by the School of Mathematics at the University of Birmingham.

## A  Lie algebra and Lie group conventions

In this appendix, we gather our Lie algebra and Lie group conventions and recall some useful formulas. All the material in this appendix (and of the next one) is textbook material, hence we will be brief – see *e.g.* [172].

### A.1  Lie algebra and Killing form

Consider $\mathfrak{g}$ a compact (semi-)simple Lie algebra. Its complexification $\mathfrak{g}_{\mathbb{C}}$ admits a decomposition:

$$\mathfrak{g}_{\mathbb{C}} = \mathfrak{h}_{\mathbb{C}} \oplus \bigoplus_{\alpha \in \Delta} V_\alpha, \tag{A.1}$$

with $\mathfrak{h}_{\mathbb{C}}$ being the Cartan subalgebra and $V_\alpha := \{X \in \mathfrak{g} \mid [H, X] = \alpha(H)X, \; \forall H \in \mathfrak{h}_{\mathbb{C}}\} \subset \mathfrak{g}_{\mathbb{C}}$ are the root spaces indexed by the roots $\alpha \in \Delta \subseteq \mathfrak{h}_{\mathbb{C}}^*$. The integral span of the roots $\alpha \in \Delta$ gives us the *root lattice* $\Lambda_{\mathrm{r}} \subset \mathfrak{h}_{\mathbb{C}}^*$ of the Lie algebra $\mathfrak{g}$. The *simple roots* are the $\mathrm{rank}(\mathfrak{g})$ roots such that any root $\alpha \in \Delta$ can be written as a linear combination of simple roots, with integral coefficients which are either all positive or all negative. In particular, the simple roots form a basis of the root lattice. The set of all positive (negative) roots is denoted by $\Delta^\pm$, with $\Delta = \Delta^+ \oplus \Delta^-$.

Another lattice directly associated with the algebra $\mathfrak{g}$ is the weight lattice $\Lambda_{\mathrm{w}} \subset \mathfrak{h}^*$. For each root $\alpha \in \Delta$, there is a Cartan element $H_\alpha \in \mathfrak{h}_{\mathbb{C}}$ satisfying the requirement that $H_\alpha \in [V_\alpha, V_{-\alpha}]$ and $\alpha(H_\alpha) = 2$. The weight lattice $\Lambda_{\mathrm{w}}$ is generated by $\beta \in \mathfrak{h}^*$ such that $\beta(H_\alpha) \in \mathbb{Z}$. The roots $\alpha$ are weight for the adjoint representation, and thus we have the embedding $\Lambda_{\mathrm{r}} \subseteq \Lambda_{\mathrm{w}}$. The quotient of these two lattices gives us a finite abelian group:

$$\Lambda_{\mathrm{w}}/\Lambda_{\mathrm{r}} \cong Z(\widetilde{G}) \equiv \widetilde{\Gamma}, \tag{A.2}$$

which is isomorphic to the centre of $\widetilde{G}$, the unique simply-connected Lie group with Lie algebra $\mathfrak{g}$.

**The Cartan–Killing form.** We denote by $(\rho, \lambda)$ the Killing form on weight space $\Lambda_{\mathrm{w}}$, and similarly by $(u, v)$ the Killing form on $\mathfrak{g}_{\mathbb{C}}$ itself. We denote by $\|\alpha\|^2 = (\alpha, \alpha)$ the length squared of the root $\alpha$, with the normalisation that gives $\|\alpha\|^2 = 2$ to the longer simple roots.[52] Let $\alpha^{(a)}$

---

[52]Except for $\mathfrak{g}_2$ where the roots have squared lengths 2 and $\frac{2}{3}$. What really matters is that $2/\|\alpha\|^2 \in \mathbb{Z}$ for all simple roots.

denote the simple roots, with $a = 1, \cdots, \mathrm{rank}(\mathfrak{g})$. The Cartan matrix of $\mathfrak{g}$ is defined by:

$$A^{ab} = 2\frac{(\alpha^{(a)}, \alpha^{(b)})}{(\alpha^{(b)}, \alpha^{(b)})}. \tag{A.3}$$

Note that this is not symmetric unless $\mathfrak{g}$ is simply-laced. The *fundamental weights* $\{e_a\}$ are defined through the relation:

$$(e_a, (\alpha^{(b)})^\vee) = \delta_a{}^b, \qquad \text{with} \quad (\alpha^{(a)})^\vee \equiv 2\frac{\alpha^{(a)}}{(\alpha^{(a)}, \alpha^{(a)})}, \tag{A.4}$$

where the coroots $\alpha^\vee \equiv 2\alpha/(\alpha, \alpha)$ satisfy $(\alpha^\vee, \lambda) \in \mathbb{Z}$ for any $\lambda \in \Lambda_{\mathrm{w}}$. Using the Killing form to define the elements $(\alpha^\vee, -) \in \mathfrak{h}$, one also defines the the *coroot lattice*

$$\Lambda_{\mathrm{cr}} \cong \Lambda_{\mathrm{w}}^*, \tag{A.5}$$

as the lattice spanned by the coroots $(\alpha^\vee, -)$. Hence by 'a coroot' one can mean either a weight $\alpha^\vee \in \Lambda_{\mathrm{w}}$ or an element of the dual weight lattice (A.5). Here we choose to view the coroots as weights, and thus mostly avoid the notation (A.5). Instead, when thinking of the simply-connected group $\widetilde{G}$, we shall view

$$\Lambda_{\mathrm{cr}} \cong \Lambda_{\mathrm{w}}^* \cong \Lambda_{\mathrm{mw}}^{\widetilde{G}}, \tag{A.6}$$

as the magnetic weight lattice of the compact group $\widetilde{G}$. (We review our notation for magnetic and electric weight lattices in subsection A.2 below.)

The fundamental weights form an integral basis of $\Lambda_{\mathrm{w}}$, hence any weight $\lambda$ is expanded as:

$$\lambda = \lambda^a e_a. \tag{A.7}$$

For $\lambda$ the highest weight of a representation $\mathfrak{R}_\lambda$, the coefficients $\lambda^a$ are called the Dynkin labels of the representation (they are then non-negative). The simple roots themselves are expanded as:

$$\alpha^{(a)} = A^{ab} e_b, \tag{A.8}$$

hence the Cartan matrix encodes the Dynkin labels of the simple roots. We denote by $\kappa^{-1}$ the matrix for the symmetric quadratic form $(-, -)$ in the fundamental-weight basis. It is given in terms of the inverse Cartan matrix as:

$$\kappa_{ab}^{-1} = (e_a, e_b) = (A^{-1})_{ab}\frac{\|\alpha^{(b)}\|^2}{2}. \tag{A.9}$$

Similarly, we denote the dual fundamental basis on $\mathfrak{h}$ by $\{e^a\}$, so that $e_a(e^b) = \delta_a{}^b$, and the Killing form on $\mathfrak{g}$ is then given explicitly by:

$$\kappa^{ab} = \frac{2}{\|\alpha^{(a)}\|^2}A^{ab} = ((\alpha^{(a)})^\vee, (\alpha^{(b)})^\vee), \tag{A.10}$$

which is clearly symmetric. We also recall the useful relations:

$$\det(\kappa) = \prod_a \frac{2}{\|\alpha^{(a)}\|^2}\det(A), \qquad \det(A) = |\Lambda_{\mathrm{w}}/\Lambda_{\mathrm{r}}| = |Z(\widetilde{G})|, \tag{A.11}$$

as well as:

$$\left|\frac{\Lambda_{\mathrm{w}}}{K\Lambda_{\mathrm{cr}}}\right| = K^{\mathrm{rank}(\mathfrak{g})}\det(\kappa), \tag{A.12}$$

where here $\Lambda_{\mathrm{w}}/K\Lambda_{\mathrm{cr}}$ denotes the quotient of the weight lattice by the equivalence relation $\lambda \sim \lambda + K\alpha^\vee$, for any coroot $\alpha^\vee$.

## A.2 Compact groups and their electric and magnetic weight lattices

Given any Lie group $G$ with Lie algebra is $\mathfrak{g}$, we consider its (electric) weight lattice $\Lambda_{\mathrm{w}}^G \subseteq \Lambda_{\mathrm{w}}$, which contains all possible weights for representations of $G$, and its magnetic-weight lattice $\Lambda_{\mathrm{mw}}^G \subseteq \Lambda_{\mathrm{r}}^*$, which is the lattice of the GNO-quantised magnetic fluxes [194],

$$(\Lambda_{\mathrm{mw}}^G) \cong (\Lambda_{\mathrm{w}}^G)^* . \tag{A.13}$$

Denoting by $\Lambda_{\mathrm{mw}} \equiv \Lambda_r^*$ the dual lattice to the root lattice, which is the largest possible magnetic-weight lattice, we note that:

$$\Lambda_{\mathrm{w}}^{\widetilde{G}} = \Lambda_{\mathrm{w}} , \qquad \Lambda_{\mathrm{mw}}^{\widetilde{G}} = \Lambda_{\mathrm{cr}} , \qquad (\Lambda_{\mathrm{mw}}^{\widetilde{G}/Z(\widetilde{G})}) = \Lambda_{\mathrm{mw}} , \tag{A.14}$$

for $\widetilde{G}$ the universal cover of $G$ and $\widetilde{G}/Z(\widetilde{G}) \cong G/Z(G)$ the centreless version of the Lie group. We then have the inclusions:

$$
\begin{array}{ccccccc}
\mathfrak{h}^* : & \Lambda_{\mathrm{r}} & \overset{Z(G)}{\subseteq} & \Lambda_{\mathrm{w}}^G & \overset{\pi_1(G)}{\subseteq} & \Lambda_{\mathrm{w}} & \\
& \updownarrow {\scriptstyle *} & & \updownarrow {\scriptstyle *} & & \updownarrow {\scriptstyle *} & \\
\mathfrak{h} : & \Lambda_{\mathrm{mw}} & \overset{Z(G)}{\supseteq} & \Lambda_{\mathrm{mw}}^G & \overset{\pi_1(G)}{\supseteq} & \Lambda_{\mathrm{cr}} ,
\end{array}
\tag{A.15}
$$

where here $A \overset{\mathcal{G}}{\subseteq} B$ stands for relation $\mathcal{G} \cong B/A$. For $\Gamma \subseteq \widetilde{\Gamma}$ any subgroup of the centre $\widetilde{\Gamma} = Z(\widetilde{G})$ given as in in (A.2), we have a group $G = \widetilde{G}/\Gamma$, so that:

$$\pi_1(G) \cong \Gamma , \qquad Z(G) \cong \widetilde{\Gamma}/\Gamma . \tag{A.16}$$

Conversely, given a Lie algebra $\mathfrak{g}$, a choice of sub-lattices

$$\Lambda_{\mathrm{w}}^G \times \Lambda_{\mathrm{mw}}^G \subseteq \Lambda_{\mathrm{w}} \times \Lambda_{\mathrm{mw}} , \quad \text{such that} \quad \Lambda_{\mathrm{mw}}^G \cong (\Lambda_{\mathrm{w}}^G)^* , \tag{A.17}$$

determines a compact Lie group $G$.

## A.3 Weyl group and Weyl character formula

The Weyl group $W_{\mathfrak{g}} = W_G$ is generated by $s_\alpha$, the reflections along the roots. The action of these reflections on the weights $\lambda \in \Lambda_{\mathrm{w}}$ is given by:

$$s_\alpha(\lambda) = \lambda - (\alpha^\vee, \lambda)\alpha . \tag{A.18}$$

Any element $w \in W_{\mathfrak{g}}$ can be written as a word in these simple reflections. We denote the action of $w$ on a weight by $w(\lambda)$. Of particular interest to us will be the Weyl character formula:

$$\sum_{w \in W_{\mathfrak{g}}} \epsilon(w) e^{-2\pi i w(\rho_{\mathrm{w}} + \lambda)(u)} = \mathrm{ch}_\lambda(e^{-2\pi i u}) \, e^{-2\pi i \rho_{\mathrm{w}}(u)} \prod_{\alpha \in \Delta^+} (1 - e^{2\pi i \alpha(u)}) , \tag{A.19}$$

which holds for an irreducible representation of highest weight $\lambda$ with character:

$$\mathrm{ch}_\lambda(e^{-2\pi i u}) = \sum_{\rho \in \mathfrak{R}_\lambda} e^{-2\pi i \rho(u)} . \tag{A.20}$$

In particular, for $\lambda = 0$ (the trivial representation) we have the Weyl determinant formula:

$$\sum_{w \in W_{\mathfrak{g}}} \epsilon(w) e^{-2\pi i w \rho_{\mathrm{w}}(u)} = e^{-2\pi i \rho_{\mathrm{w}}(u)} \prod_{\alpha \in \Delta^+} (1 - e^{2\pi i \alpha(u)}) . \tag{A.21}$$

Table 2: Simple Lie algebras classification — a short fact sheet.

| $\mathfrak{g}$ | $\widetilde{G}$ | $Z(\widetilde{G})$ | $h^\vee(\mathfrak{g})$ | $\dim \mathfrak{g}$ |
|---|---|---|---|---|
| $\mathfrak{a}_n$ | SU$(n+1)$ | $\mathbb{Z}_{n+1}$ | $n+1$ | $n(2+n)$ |
| $\mathfrak{b}_n$ | Spin$(2n+1)$ | $\mathbb{Z}_2$ | $2n-1$ | $n(2n+1)$ |
| $\mathfrak{c}_n$ | Sp$(2n)$ | $\mathbb{Z}_2$ | $n+1$ | $n(2n+1)$ |
| $\mathfrak{d}_n$ | Spin$(2n)$ | $\mathbb{Z}_4$ for $n$ odd | $2n-2$ | $n(2n-1)$ |
|  |  | $\mathbb{Z}_2 \times \mathbb{Z}_2$ for $n$ even |  |  |
| $\mathfrak{e}_6$ | E$_6$ | $\mathbb{Z}_3$ | 12 | 78 |
| $\mathfrak{e}_7$ | E$_7$ | $\mathbb{Z}_2$ | 18 | 133 |
| $\mathfrak{e}_8$ | E$_8$ | $\emptyset$ | 30 | 248 |
| $\mathfrak{f}_4$ | F$_4$ | $\emptyset$ | 4 | 52 |
| $\mathfrak{g}_2$ | G$_2$ | $\emptyset$ | 9 | 14 |

# B  Simple Lie groups and Chern–Simons TQFTs

In this appendix, for each simple Lie algebra $\mathfrak{g}$, we list our conventions for the Cartan matrix and the Killing form, we study the one-form symmetry of the simply-connected group $\widetilde{G}$, and we classify the possible $\mathcal{N}=2$ supersymmetric Chern–Simons theories $G_K$ obtained as quotients

$$G = \widetilde{G}/\Gamma\,, \qquad \Gamma \subseteq \widetilde{\Gamma} \cong Z(\widetilde{G})\,. \tag{B.1}$$

For each $\widetilde{G}$, we write down the abelian anyons generating the one-form symmetry. These are the Wilson lines $a_\gamma = W_{\lambda_\gamma}$, for some integrable representations $\lambda_\gamma$ associated to the group elements $\gamma \in \Gamma$. Their conformal spin is given by:

$$h[a_\gamma] = \frac{(\lambda_\gamma\,,\,2\rho_W + \lambda_\gamma)}{2K} \mod 1\,. \tag{B.2}$$

Recall that $K = k + h^\vee$, where $k$ is the bosonic Chern–Simons level and $h^\vee$ the dual Coxeter number of $\mathfrak{g}$. The abelian one-form symmetry $\Gamma_\gamma$ generated by $a_\gamma$ is non-anomalous if $h[a_\gamma] \in \frac{1}{2}\mathbb{Z}$, and anomalous otherwise. Furthermore, in the non-anomalous case, the resulting quotient theory $(\widetilde{G}/\Gamma_\gamma)_K$ is a bosonic Chern–Simons theory if $h[a_\gamma] \in \mathbb{Z}$, and it is a spin-TQFT if $h[a_\gamma] + \frac{1}{2} \in \mathbb{Z}$.

Some relevant quantities for all simple Lie algebras are recalled in table 2. In the following, $n = \text{rank}(\mathfrak{g})$.

## B.1  The $\mathfrak{a}_n$ series

Consider $\mathfrak{g} = \mathfrak{a}_n = \mathfrak{su}(n+1)$, for $n \geq 1$. The Cartan matrix is given by:

$$\mathfrak{a}_n: \qquad A = \begin{pmatrix} 2 & -1 & 0 & \cdots & 0 & 0 \\ -1 & 2 & -1 & \cdots & 0 & 0 \\ 0 & -1 & 2 & \cdots & 0 & 0 \\ \vdots & \vdots & \vdots & \ddots & \vdots & \vdots \\ 0 & 0 & 0 & \cdots & 2 & -1 \\ 0 & 0 & 0 & \cdots & -1 & 2 \end{pmatrix}, \qquad \det(A) = n+1\,. \tag{B.3}$$

Let us use the notation $N = n + 1$. The simply-connected group is $\widetilde{G} = \mathrm{SU}(N)$, with centre $Z(\widetilde{G}) = \mathbb{Z}_N$. This is a simply-laced Lie algebra, hence all simple roots have length squared $\|\alpha^{(a)}\|^2 = 2$ and the Killing form reads:

$$\kappa_{ab} = A^{ab}, \qquad \kappa_{ab}^{-1} = (A^{-1})_{ab}. \tag{B.4}$$

In particular, we have:

$$\kappa_{ab}^{-1} = \frac{\min(a, b)(n+1) - ab}{n+1}, \qquad a, b = 1, \ldots, n, \tag{B.5}$$

which gives us the matrix:

$$\kappa^{-1} = \frac{1}{n+1} \begin{pmatrix} n & n-1 & n-2 & \cdots & 2 & 1 \\ n-1 & 2(n-1) & 2(n-2) & \cdots & 4 & 2 \\ n-2 & 2(n-2) & 3(n-2) & \cdots & 6 & 3 \\ \vdots & \vdots & \vdots & \ddots & \vdots & \vdots \\ 2 & 4 & 6 & \cdots & 2(n-1) & n-1 \\ 1 & 2 & 3 & \cdots & n-1 & n \end{pmatrix}. \tag{B.6}$$

**One-form symmetry and Chern–Simons theories.** The $\mathbb{Z}_N^{(1)}$ one-form symmetry of the $\mathrm{SU}(N)_k$ CS theory is generated by the abelian anyon $a = a_{\gamma_0}$ with

$$\lambda_{\gamma_0} = [k, 0, 0, \cdots, 0]. \tag{B.7}$$

More generally, the element $\gamma = s\gamma_0$ (for $s \in \mathbb{Z}_N$) corresponds to the abelian anyon $a^s = a_{s\gamma_0}$ and to the integrable representation:

$$[\lambda_{s\gamma_0}^a] = [k\delta^{a,s}]. \tag{B.8}$$

We have:

$$h[a^s] = \frac{k(N-s)s}{2N} \pmod 1. \tag{B.9}$$

Without loss of generality, assume that $s$ divides $N$ and define $r = \frac{N}{s}$, so that $a^s$ generates the one-form symmetry $\mathbb{Z}_r^{(1)} \subseteq \mathbb{Z}_N^{(1)}$. We then have

$$h[a^s] = \frac{kN(r-1)}{2r^2} \pmod 1. \tag{B.10}$$

The $\mathbb{Z}_r^{(1)}$ symmetry is non-anomalous if and only if $\frac{kN}{r^2} \in \mathbb{Z}$. In this case, we have:

$$h[a^s] = \begin{cases} \frac{1}{2}, & \text{if } r \text{ is even and } \frac{kN}{r^2} \text{ is odd}, \\ 0, & \text{otherwise} \end{cases} \qquad \left(\text{assuming } \frac{kN}{r^2} \in \mathbb{Z}\right). \tag{B.11}$$

Therefore, gauging $\mathbb{Z}_r^{(1)}$ to obtain $(\mathrm{SU}(N)/\mathbb{Z}_r)_k$ (in the non-supersymmetric notation) gives us a spin-TQFT in the first case, while it gives us a bosonic CS theory in the second case.

## B.2 The $\mathfrak{b}_n$ series

Consider $\mathfrak{g} = \mathfrak{b}_n = \mathfrak{so}(2n+1)$, for $n \geq 2$. The Cartan matrix is given by:

$$\mathfrak{b}_n: \qquad A = \begin{pmatrix} 2 & -1 & 0 & \cdots & 0 & 0 \\ -1 & 2 & -1 & \cdots & 0 & 0 \\ 0 & -1 & 2 & \cdots & 0 & 0 \\ \vdots & \vdots & \vdots & \ddots & \vdots & \vdots \\ 0 & 0 & 0 & \cdots & 2 & -2 \\ 0 & 0 & 0 & \cdots & -1 & 2 \end{pmatrix}, \qquad \det(A) = 2. \tag{B.12}$$

The simply-connected group is $\widetilde{G} = \mathrm{Spin}(2n+1)$, with centre $Z(\widetilde{G}) = \mathbb{Z}_2$. The roots have squared lengths:

$$(\|\alpha^{(a)}\|^2) = (2, 2, \cdots, 2, 1). \tag{B.13}$$

The Killing form and its inverse are:

$$\kappa = \begin{pmatrix} 2 & -1 & 0 & \cdots & 0 & 0 \\ -1 & 2 & -1 & \cdots & 0 & 0 \\ 0 & -1 & 2 & \cdots & 0 & 0 \\ \vdots & \vdots & \vdots & \ddots & \vdots & \vdots \\ 0 & 0 & 0 & \cdots & 2 & -2 \\ 0 & 0 & 0 & \cdots & -2 & 4 \end{pmatrix}, \quad \kappa^{-1} = \frac{1}{2}\begin{pmatrix} 2 & 2 & 2 & \cdots & 2 & 1 \\ 2 & 4 & 4 & \cdots & 4 & 2 \\ 2 & 4 & 6 & \cdots & 6 & 3 \\ \vdots & \vdots & \vdots & \ddots & \vdots & \vdots \\ 2 & 4 & 6 & \cdots & 2(n-1) & n-1 \\ 1 & 2 & 3 & \cdots & n-1 & \frac{n}{2} \end{pmatrix}. \tag{B.14}$$

**One-form symmetry and Chern–Simons theories.** The $\mathbb{Z}_2^{(1)}$ one-form symmetry of the $\mathrm{Spin}(2n+1)_k$ CS theory is generated by an abelian anyon $a_{\gamma_0}$ with:

$$\lambda_{\gamma_0} = [k, 0, \cdots, 0], \qquad h[a_{\gamma_0}] = \frac{k}{2}. \tag{B.15}$$

Therefore the $\mathbb{Z}_2^{(1)}$ symmetry is never anomalous. Upon gauging, we get

$$\mathrm{SO}(2n+1) = \mathrm{Spin}(2n+1)/\mathbb{Z}_2,$$

and the Chern–Simons theory $\mathrm{SO}(2n+1)_k$ is bosonic for $k$ even and a spin-TQFT for $k$ odd.

## B.3  The $\mathfrak{c}_n$ series

Consider $\mathfrak{g} = \mathfrak{c}_n = \mathfrak{sp}(2n)$, for $n \geq 2$. The Cartan matrix is given by:

$$\mathfrak{c}_n: \qquad A = \begin{pmatrix} 2 & -1 & 0 & \cdots & 0 & 0 \\ -1 & 2 & -1 & \cdots & 0 & 0 \\ 0 & -1 & 2 & \cdots & 0 & 0 \\ \vdots & \vdots & \vdots & \ddots & \vdots & \vdots \\ 0 & 0 & 0 & \cdots & 2 & -1 \\ 0 & 0 & 0 & \cdots & -2 & 2 \end{pmatrix}, \qquad \det(A) = 2. \tag{B.16}$$

The simply-connected group is $\widetilde{G} = \mathrm{Sp}(2n)$, with centre $Z(\widetilde{G}) = \mathbb{Z}_2$. The roots have squared lengths:

$$(\|\alpha^{(a)}\|^2) = (1, 1, \cdots, 1, 2). \tag{B.17}$$

The Killing form and its inverse are:

$$\kappa = \begin{pmatrix} 4 & -2 & 0 & \cdots & 0 & 0 \\ -2 & 4 & -2 & \cdots & 0 & 0 \\ 0 & -2 & 4 & \cdots & 0 & 0 \\ \vdots & \vdots & \vdots & \ddots & \vdots & \vdots \\ 0 & 0 & 0 & \cdots & 4 & -2 \\ 0 & 0 & 0 & \cdots & -2 & 2 \end{pmatrix}, \quad \kappa^{-1} = \frac{1}{2}\begin{pmatrix} 1 & 1 & 1 & \cdots & 1 & 1 \\ 1 & 2 & 2 & \cdots & 2 & 2 \\ 1 & 2 & 3 & \cdots & 3 & 3 \\ \vdots & \vdots & \vdots & \ddots & \vdots & \vdots \\ 1 & 2 & 3 & \cdots & n-1 & n-1 \\ 1 & 2 & .3 & \cdots & n-1 & n \end{pmatrix}. \tag{B.18}$$

**One-form symmetry and Chern–Simons theories.** The $\mathbb{Z}_2^{(1)}$ one-form symmetry of the $\mathrm{Sp}(2n)_k$ CS theory is generated by an abelian anyon $a_{\gamma_0}$ with:

$$\lambda_{\gamma_0} = [0, \cdots, 0, k], \qquad h[a_{\gamma_0}] = \frac{kn}{4}. \tag{B.19}$$

Therefore the $\mathbb{Z}_2^{(1)}$ symmetry is non-anomalous if and only if $\frac{kn}{2} \in \mathbb{Z}$. In the non-anomalous case, we then have

$$h[a_{\gamma_0}] = \begin{cases} \frac{1}{2}, & \text{if } \frac{kn}{2} \text{ is odd}, \\ 0, & \text{if } \frac{kn}{2} \text{ is even} \end{cases} \qquad \left(\text{assuming } \frac{kn}{2} \in \mathbb{Z}\right). \tag{B.20}$$

Upon gauging, we have $\mathrm{PSp}(2n) \equiv \mathrm{Sp}(2n)/\mathbb{Z}_2$ and the Chern–Simons theory $\mathrm{PSp}(2n)_k$ is a spin-TQFT in the first case and a bosonic theory in the second case.

### B.4 The $\mathfrak{d}_n$ series

Consider $\mathfrak{g} = \mathfrak{d}_n = \mathfrak{so}(2n)$, for $n \geq 2$. The Cartan matrix is given by:

$$\mathfrak{d}_n: \qquad A = \begin{pmatrix} 2 & -1 & 0 & \cdots & 0 & 0 & 0 \\ -1 & 2 & -1 & \cdots & 0 & 0 & 0 \\ 0 & -1 & 2 & \cdots & 0 & 0 & 0 \\ \vdots & \vdots & \vdots & \ddots & \vdots & \vdots & \vdots \\ 0 & 0 & 0 & \cdots & 2 & -1 & -1 \\ 0 & 0 & 0 & \cdots & -1 & 2 & 0 \\ 0 & 0 & 0 & \cdots & -1 & 0 & 2 \end{pmatrix}, \qquad \det(A) = 4. \tag{B.21}$$

The simply-connected group is $\widetilde{G} = \mathrm{Spin}(2n)$, with centre $Z(\widetilde{G}) = \mathbb{Z}_4$ if $n$ is odd and $Z(\widetilde{G}) = \mathbb{Z}_2 \times \mathbb{Z}_2$ if $n$ is even. The Lie algebra is simply-laced hence the Killing form is given as in (B.4), and we have:

$$\kappa_{ab}^{-1} = \begin{cases} \min(a, b), & \text{if } a, b \leq n-2, \\ \frac{a}{2}, & \text{if } a \leq n-2 \text{ and } b = n-1 \text{ or } n, \\ \frac{b}{2}, & \text{if } b \leq n-2 \text{ and } a = n-1 \text{ or } n, \\ \frac{n}{4}, & \text{if } a = b = n-1 \text{ or } a = b = n, \\ \frac{n-2}{4}, & \text{if } a = n-1 \text{ and } b = n \text{ or vice versa}, \end{cases} \tag{B.22}$$

that is:

$$\kappa^{-1} = \begin{pmatrix} 1 & 1 & 1 & \cdots & 1 & \frac{1}{2} & \frac{1}{2} \\ 1 & 2 & 2 & \cdots & 2 & 1 & 1 \\ 1 & 2 & 3 & \cdots & 3 & \frac{3}{2} & \frac{3}{2} \\ \vdots & \vdots & \vdots & \ddots & \vdots & \vdots & \vdots \\ 1 & 2 & 3 & \cdots & n-2 & \frac{n-2}{2} & \frac{n-2}{2} \\ \frac{1}{2} & 1 & \frac{3}{2} & \cdots & \frac{n-2}{2} & \frac{n}{4} & \frac{n-2}{4} \\ \frac{1}{2} & 1 & \frac{3}{2} & \cdots & \frac{n-2}{2} & \frac{n-2}{4} & \frac{n}{4} \end{pmatrix}. \tag{B.23}$$

**One-form symmetry and Chern–Simons theories for $n = 2l + 1$.** For $n = 2l + 1$ ($n$ odd), the $\mathrm{Spin}(4l + 2)_k$ CS theory has a one-form symmetry $\mathbb{Z}_4^{(1)}$ which is generated by an abelian anyon $a_{\gamma_0}$ with:

$$\lambda_{\gamma_0} = [0, \cdots, 0, k], \qquad h[a_{\gamma_0}] = \frac{kn}{8}. \tag{B.24}$$

Therefore the full $\mathbb{Z}_4^{(1)}$ symmetry is non-anomalous if and only if $\frac{k(2l+1)}{4} \in \mathbb{Z}$. In the non-anomalous case, we then have

$$
h[a_{\gamma_0}] = \begin{cases} \frac{1}{2}, & \text{if } \frac{k(2l+1)}{4} \text{ is odd}, \\ 0, & \text{if } \frac{k(2l+1)}{4} \text{ is even} \end{cases} \qquad \left(\text{assuming } \frac{k(2l+1)}{4} \in \mathbb{Z}\right). \tag{B.25}
$$

Upon gauging, we have $\mathrm{PSO}(4l+2) \equiv \mathrm{Spin}(4l+2)/\mathbb{Z}_4$ and the Chern–Simons theory $\mathrm{PSO}(4l+2)_k$ is a spin-TQFT in the first case and a bosonic theory in the second case.

We can also consider gauging the $\mathbb{Z}_2^{(1)} \subset \mathbb{Z}_4^{(1)}$ subgroup generated by:

$$
\lambda_{2\gamma_0} = [k, 0, \cdots, 0], \qquad h[a_{\gamma_0}^2] = \frac{k}{2}. \tag{B.26}
$$

This symmetry is never anomalous. Upon gauging, we get $\mathrm{SO}(4l+2) \equiv \mathrm{Spin}(4l+2)/\mathbb{Z}_2$ and the Chern–Simons theory $\mathrm{SO}(4l+2)_k$ is a spin-TQFT if $k$ is odd and a bosonic CS theory if $k$ is even.

**One-form symmetry and Chern–Simons theories for $n = 2l$.** For $n = 2l$ ($n$ even), the $\mathrm{Spin}(4l)_k$ CS theory has a one-form symmetry $\mathbb{Z}_2^{(1)} \times \widetilde{\mathbb{Z}}_2^{(1)}$ which is generated by two abelian anyons $a_{\gamma_0}$ and $a_{\widetilde{\gamma}_0}$ with:

$$
\begin{aligned}
\lambda_{\gamma_0} &= [0, \cdots, 0, k, 0], & h[a_{\gamma_0}] &= \frac{kn}{8}, \\
\lambda_{\widetilde{\gamma}_0} &= [0, \cdots, 0, 0, k], & h[a_{\widetilde{\gamma}_0}] &= \frac{kn}{8}.
\end{aligned} \tag{B.27}
$$

We also consider the diagonal $\mathbb{Z}_2$, denoted by $\mathbb{Z}_2^{\mathrm{diag}}$, generated by $a_{\gamma_0} a_{\widetilde{\gamma}_0} = a_{\gamma_0 + \widetilde{\gamma}_0}$:

$$
\lambda_{\gamma_0 + \gamma_0} = [k, 0, \cdots, 0, 0, 0], \qquad h[a_{\gamma_0}] = \frac{k}{2}. \tag{B.28}
$$

In the special case $k = 1$, the abelian anyons $a_{\gamma_0}$ and $a_{\widetilde{\gamma}_0}$ are the two Wilson lines in the spinor representations $S^{\pm}$, while $a_{\gamma_0 + \widetilde{\gamma}_0}$ is the Wilson line in the vector representation (which is always a subrepresentation of $S^+ \otimes S^-$).

The full one-form symmetry is non-anomalous if and only if $\frac{kl}{2} \in \mathbb{Z}$. Then, we obtain the $\mathrm{PSO}(4l)_k$ theory, which is bosonic if $k$ is even and spin if $k$ is odd. If we only quotient by one of the three $\mathbb{Z}_2$ subgroups we obtain:

$$
\begin{aligned}
\mathrm{SO}_+(4l)_k &\equiv (\mathrm{Spin}(4l)/\mathbb{Z}_2)_k, \\
\mathrm{SO}_-(4l)_k &\equiv (\mathrm{Spin}(4l)/\widetilde{\mathbb{Z}}_2)_k, \\
\mathrm{SO}(4l)_k &\equiv (\mathrm{Spin}(4l)/\mathbb{Z}_2^{\mathrm{diag}})_k.
\end{aligned} \tag{B.29}
$$

Here, $\mathrm{SO}_{\pm}(4l)$ denote the semi-spin groups, which admit only one of the two spinor representations, while $\mathrm{SO}(4l)$ is the ordinary special orthogonal group which does not admit spinors. Note that, for all $\mathrm{SO}(m)_k$ Chern–Simons theories ($m \in \mathbb{Z}$, from either the $\mathfrak{b}_n$ or $\mathfrak{d}_n$ series), we have a spin-TQFT for $k$ odd and a bosonic theory for $k$ even.

## B.5 The $\mathfrak{e}_n$ series

The exceptional algebras $\mathfrak{e}_n$ for $n = 6, 7, 8$ are simply laced. Let us consider each in turn.

### B.5.1 The $\mathfrak{e}_6$ algebra and groups

The $\mathfrak{e}_6$ Lie algebra has the Cartan matrix and quadratic form:

$$A = \kappa = \begin{pmatrix} 2 & -1 & 0 & 0 & 0 & 0 \\ -1 & 2 & -1 & 0 & 0 & 0 \\ 0 & -1 & 2 & -1 & 0 & -1 \\ 0 & 0 & -1 & 2 & -1 & 0 \\ 0 & 0 & 0 & -1 & 2 & 0 \\ 0 & 0 & -1 & 0 & 0 & 2 \end{pmatrix}, \qquad \kappa^{-1} = \frac{1}{3}\begin{pmatrix} 4 & 5 & 6 & 4 & 2 & 3 \\ 5 & 10 & 12 & 8 & 4 & 6 \\ 6 & 12 & 18 & 12 & 6 & 9 \\ 4 & 8 & 12 & 10 & 5 & 6 \\ 2 & 4 & 6 & 5 & 4 & 3 \\ 3 & 6 & 9 & 6 & 3 & 6 \end{pmatrix}. \tag{B.30}$$

The simply-connected group is $\widetilde{G} = \mathrm{E}_6$, with centre $Z(\widetilde{G}) = \mathbb{Z}_3$.

**One-form symmetry and Chern–Simons theories.** The $(\mathrm{E}_6)_k$ CS theory has a one-form symmetry $\mathbb{Z}_3^{(1)}$ generated by an abelian anyon $a_{\gamma_0}$ with:

$$\lambda_{\gamma_0} = [0,0,0,0,k,0], \qquad h[a_{\gamma_0}] = \frac{2k}{3}. \tag{B.31}$$

This means that the symmetry is non-anomalous if and only if $k \in 3\mathbb{Z}$. In this case, the CS theory $(\mathrm{E}_6/\mathbb{Z}_3)_k$ is bosonic.

### B.5.2 The $\mathfrak{e}_7$ algebra and groups

The $\mathfrak{e}_7$ Lie algebra has the Cartan matrix and quadratic form:

$$A = \kappa = \begin{pmatrix} 2 & -1 & 0 & 0 & 0 & 0 & 0 \\ -1 & 2 & -1 & 0 & 0 & 0 & 0 \\ 0 & -1 & 2 & -1 & 0 & 0 & 0 \\ 0 & 0 & -1 & 2 & -1 & 0 & -1 \\ 0 & 0 & 0 & -1 & 2 & -1 & 0 \\ 0 & 0 & 0 & 0 & -1 & 2 & 0 \\ 0 & 0 & 0 & -1 & 0 & 0 & 2 \end{pmatrix}, \qquad \kappa^{-1} = \frac{1}{2}\begin{pmatrix} 4 & 6 & 8 & 6 & 4 & 2 & 4 \\ 6 & 12 & 16 & 12 & 8 & 4 & 8 \\ 8 & 16 & 24 & 18 & 12 & 6 & 12 \\ 6 & 12 & 18 & 15 & 10 & 5 & 9 \\ 4 & 8 & 12 & 10 & 8 & 4 & 6 \\ 2 & 4 & 6 & 5 & 4 & 3 & 3 \\ 4 & 8 & 12 & 9 & 6 & 3 & 7 \end{pmatrix}.$$
$$\tag{B.32}$$

The simply-connected group is $\widetilde{G} = \mathrm{E}_7$, with centre $Z(\widetilde{G}) = \mathbb{Z}_2$.

**One-form symmetry and Chern–Simons theories.** The $(\mathrm{E}_7)_k$ CS theory has a one-form symmetry $\mathbb{Z}_2^{(1)}$ generated by an abelian anyon $a_{\gamma_0}$ with:

$$\lambda_{\gamma_0} = [0,0,0,0,0,k,0], \qquad h[a_{\gamma_0}] = \frac{3k}{4}. \tag{B.33}$$

This means that the symmetry is non-anomalous if and only if $k \in 2\mathbb{Z}$. In this case, the CS theory $(\mathrm{E}_7/\mathbb{Z}_2)_k$ is bosonic if $\frac{k}{2}$ is even and it is a spin-TQFT if $\frac{k}{2}$ is odd.

### B.5.3 The $\mathfrak{e}_8$ algebra and groups

The $\mathfrak{e}_8$ Lie algebra has the Cartan matrix and quadratic form:

$$A = \kappa = \begin{pmatrix} 2 & -1 & 0 & 0 & 0 & 0 & 0 & 0 \\ -1 & 2 & -1 & 0 & 0 & 0 & 0 & 0 \\ 0 & -1 & 2 & -1 & 0 & 0 & 0 & 0 \\ 0 & 0 & -1 & 2 & -1 & 0 & 0 & 0 \\ 0 & 0 & 0 & -1 & 2 & -1 & 0 & -1 \\ 0 & 0 & 0 & 0 & -1 & 2 & -1 & 0 \\ 0 & 0 & 0 & 0 & 0 & -1 & 2 & 0 \\ 0 & 0 & 0 & 0 & -1 & 0 & 0 & 2 \end{pmatrix},$$

$$\kappa^{-1} = \frac{1}{2} \begin{pmatrix} 2 & 3 & 4 & 5 & 6 & 4 & 2 & 3 \\ 3 & 6 & 8 & 10 & 12 & 8 & 4 & 6 \\ 4 & 8 & 12 & 15 & 18 & 12 & 6 & 9 \\ 5 & 10 & 15 & 20 & 24 & 16 & 8 & 12 \\ 6 & 12 & 18 & 24 & 30 & 20 & 10 & 15 \\ 4 & 8 & 12 & 16 & 20 & 14 & 7 & 10 \\ 2 & 4 & 6 & 8 & 10 & 7 & 4 & 5 \\ 3 & 6 & 9 & 12 & 15 & 10 & 5 & 8 \end{pmatrix}. \tag{B.34}$$

The simply-connected group $E_8$ has a trivial centre, so the bosonic CS theory $(E_8)_k$ does not have any one-form symmetry.

### B.6 The $\mathfrak{f}_4$ and $\mathfrak{g}_2$ algebras

For completeness, let us list the same basic quantities for the $\mathfrak{f}_4$ and $\mathfrak{g}_2$ algebras. The corresponding simply-connected group is centreless, hence the Chern–Simons theories for these groups are uniquely determined by the level and have a trivial one-form symmetry.

#### B.6.1 The $\mathfrak{f}_4$ algebra and group

The $\mathfrak{f}_4$ Lie algebra has a Cartan matrix:

$$A = \begin{pmatrix} 2 & -1 & 0 & 0 \\ -1 & 2 & -2 & 0 \\ 0 & -1 & 2 & -1 \\ 0 & 0 & -1 & 2 \end{pmatrix}. \tag{B.35}$$

The squared lengths of the simple roots are:

$$(\|\alpha^{(a)}\|^2) = (2, 2, 1, 1). \tag{B.36}$$

The Killing form and its inverse read:

$$\kappa = \begin{pmatrix} 2 & -1 & 0 & 0 \\ -1 & 2 & -2 & 0 \\ 0 & -2 & 4 & -2 \\ 0 & 0 & -2 & 4 \end{pmatrix}, \qquad \kappa^{-1} = \begin{pmatrix} 2 & 3 & 2 & 1 \\ 3 & 6 & 4 & 2 \\ 2 & 4 & 3 & \frac{3}{2} \\ 1 & 2 & \frac{3}{2} & 1 \end{pmatrix}. \tag{B.37}$$

#### B.6.2 The $\mathfrak{g}_2$ algebra and group

The $\mathfrak{g}_2$ Lie algebra has a Cartan matrix:

$$A = \begin{pmatrix} 2 & -3 \\ -1 & 2 \end{pmatrix}. \tag{B.38}$$

The squared lengths of the simple roots are:

$$(\|\alpha^{(a)}\|^2) = \left(2, \frac{2}{3}\right). \tag{B.39}$$

The Killing form and its inverse read:

$$\kappa = \begin{pmatrix} 2 & -3 \\ -3 & 6 \end{pmatrix}, \qquad \kappa^{-1} = \begin{pmatrix} 2 & 1 \\ 1 & \frac{2}{3} \end{pmatrix}. \tag{B.40}$$

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
