# Peer review of "The 3d $A$-model and generalised symmetries, Part I: bosonic Chern-Simons theories"

_SciPost Physics, doi:SciPost Phys. 19, 106 (2025)_

## Round 2 · Referee Report · Anonymous (Referee 2) · 2025-5-18

Strengths

The paper extends the application of the 3d $A$-model to a broader class of gauge groups, specifically those obtained by gauging discrete one-form symmetries. The authors provide a meticulous comparison between the supersymmetric and TQFT approaches, identifying and accounting for subtle counterterms.

Weaknesses

There are some statements that should be clarified.

Report

This paper makes a significant contribution to the understanding of the 3d $A$-model and its relation to bosonic Chern-Simons theories. The extension to more general gauge groups that are not simply connected and the detailed analysis of the correspondence between different approaches are commendable. With minor revisions, the paper would be a valuable addition to the literature on topological defects in supersymmetric gauge theories.

Requested changes

1) Questions about the presentation (these are almost trivial. I point them out because the paper, especially section 2, has an intent of acting as a reference source) - In (2.14) the notation for the Bethe equations is $\Pi_a(\hat{u}, \nu) =1$, but it's different from that used in (2.12), and the subscript $a$ has not been defined earlier. Could the authors please clarify? - p. 11 +1 "Bethe vacua of the $\tilde{G}_k$ (?) theory are (?) the solutions" - is the color-coding used in figure (2.31) consistent with that of, e.g., (2.58)? Should the circle in (2.31) intended to be blue? - p. 17 - 6 $\tilde{G}_k$ (?) - p. 18 - 10. Hasn't the modular group already been "alluded to" also in section 2.2? - p. 21 -7 "The class of (?) geometries" - p. 36 footnote 34: could the authors add a couple of words around the equation $\mathcal{N}{\mu\nu\delta} = C$ to explain why the $\delta$ index disappeared?

2) Clarifications - What is the status of (3.91)? It is said to be equivalent to (3.90), which is one of the comparison equations between supersymmetric and TQFT approach. Has it been proved? Could the authors please clarify what they mean by "verified numerically in a number of non-trivial examples"? For instance, something along the lines of footnote 50 (p. 55). How relevant is it to the comparison of (3.92) with the TQFT answer? - p. 52 below (3.99): could the authors please clarify what they mean? Is this a check or a rewriting? What are they matching to?

Recommendation

Publish (meets expectations and criteria for this Journal)

  • validity: -
  • significance: -
  • originality: -
  • clarity: -
  • formatting: -
  • grammar: -

Author:  Cyril Closset  on 2025-09-05  [id 5789]

(in reply to Report 1 on 2025-05-18)
Category:
correction

Thank you very much to the referee for the careful reading. All these points will be clarified and better explained in the next version. Let me answer here the most salient points:

  • Footnote 34 as written was wrong: we needed the extra assumption of the line a_\delta being a condensable boson, i.e. h[\delta]=0 and B(\delta, \mu)=1 for all \mu. Thank you for bringing this to our attention.
  • Equation (3.91) is equivalent to the statement (2.100) for G_k given that we have proven (2.100) for \tilde G_k. We have not proven (3.91), but we point out that it is a purely 3d TQFT identity, which could be of independent interest. (We checked it numerically in all geometries discussed later in the section.)

---

## Round 2 · Referee Report · Anonymous (Referee 1) · 2025-5-18

Strengths

1. The paper presents a complete and well-motivated formalism for implementing 1-form symmetry gauging in 3D \mathcal{N}=2 (bosonic) Chern–Simons theories via the A-model.

2. It carefully matches SUSY partition functions with TQFT computations, including explicit modular data and Seifert fibering operators.

3. The exposition is technically thorough and precise, providing a solid foundation for future work on generalised symmetries in the context of SUSY QFTs.

Weaknesses

1. The conceptual ideas behind 1-form symmetry gauging are not new, and the main novelty lies in the detailed realization within a supersymmetric setting.

2. The restriction to bosonic CS theories with simple simply-connected group without matter limits the richness of the symmetry structure, although this is acknowledged as a first step.

3. Some derivations, especially in the later sections, could benefit from more pedagogical examples to guide readers through the formalism.

Report

This is a carefully constructed and technically detailed paper that provides a systematic study of 1-form symmetry gauging in 3d \mathcal{N}=2 supersymmetric (pure) Chern–Simons theories, using the formalism of the 3d A-model.

While the general idea of gauging one-form symmetries in 3d is well-established, the present work stands out by formulating and executing this procedure in a supersymmetric context. It fills a gap in the literature and sets the stage for further developments involving matter fields and spin TQFTs.

The paper is clearly written, self-contained, and of high technical quality. I believe it meets the editorial criteria of SciPost and recommend publication after a few minor revisions to improve accessibility and clarity.

(Optional) While clearly beyond the scope of the current paper, it might be interesting in future work to clarify whether the gauging procedure considered here leads to codimension-1 defects associated with generalized (possibly non-invertible) 0-form symmetries, and how such structures manifest in the 3d A-model framework. Also, It may be interesting in future work to comment on potential mathematical connections to quantum K-theory.

Requested changes

  1. Clarify the off-shell versus on-shell distinction for the Seifert fibering operators in Sections 2.3–2.4. A brief comment on their significance beyond the matching with TQFT would be helpful.

  2. Add more concrete examples in Section 3.2.5 to illustrate the gauging procedure in familiar low-rank cases (e.g., SU(2), SU(3)). This could be helpful for the reader not familiar with the context.

  3. Further comments on the counterterm (2.101) would be appreciated. Specifically, can the cancellation of the framing anomaly be understood in terms of an anomaly polynomial or a supersymmetric completion of a gravitational Chern–Simons term involving the background supergravity fields?

  4. (Optional) A brief remark in the conclusion about future directions—such as matter fields or spin TQFTs—would help contextualize this work as part of a broader program.

Recommendation

Ask for minor revision

  • validity: -
  • significance: -
  • originality: -
  • clarity: -
  • formatting: -
  • grammar: -

Author:  Cyril Closset  on 2025-09-06  [id 5791]

(in reply to Report 2 on 2025-05-18)
Category:
answer to question

Thank you for the referee for the detailed report. We amended the text (just resubmitted to the arxiv) to address the requested points:

  1. We added the following sentence at the end of the paragraph below (2.37): "In any 3d $\CN=2$ gauge theory, the off-shell Seifert operator, like the off-shell handle-gluing operator, is a defect line operator wrapping the $S^1_A$ factor, which is realised on the 2d Coulomb branch as a particular holomorphic function $\CG_{q,p}(u)$ of the Coulomb-branch variables $u$; the on-shell operator is simply the value of that function on the Bethe vacua, which are located at particular points $u=\h u_\alpha$ on the Coulomb branch --- here, we use the shorthand notation $\CG_{q,p}(\alpha)\equiv \CG_{q,p}(\h u_\alpha)$ for the off-shell operator in order to match our 2d TQFT notation."

  2. We added explicit formulas for SU(2)_K in section 3.2.5, see the new equations (3.112)--(3.114) and comments there.

  3. There were already some comment about this, but we clarified it as follows now: "We know that this functional cannot be supersymmetric, since it allows us to go from a supersymmetric to a TQFT scheme, which makes it much harder to pin it down explicitly. We leave realising $S_{\rm ct}$ as an explicit local functional in the background fields as a challenge for future work."

  4. Indeed both a Part II and Part III paper are in preparation, and Part III will start addressing non-invertibles. We added a sentence at the end of the intro: "Finally, one can also consider non-invertible symmetries and their explicit realisation in the 3d $A$-model; we will study some instances of categorical symmetries in future work as well~\cite{CFKK-24-III}."

---

## Round 3 · Author Response

List of changes
-Added comments to paragraph after (2.37).
-Clarified the comment after (2.101) .
-Rewrote footnote 34 to correct a mistake.
-Clarified status of equation (3.91).
-Added SU(2) example in section 3.2.5.
-Added comment and reference to further work in progress at the end of section 1.

---

## Round 3 · List of Changes

-Added comments to paragraph after (2.37).
-Clarified the comment after (2.101) .
-Rewrote footnote 34 to correct a mistake.
-Clarified status of equation (3.91).
-Added SU(2) example in section 3.2.5.
-Added comment and reference to further work in progress at the end of section 1.

---

## Editorial Decision

published